# The injured sciatic nerve atlas (iSNAT), insights into the cellular and molecular basis of neural tissue degeneration and regeneration

Xiao-Feng Zhao[1†], Lucas D Huffman[1,2†], Hannah Hafner[1†], Mitre Athaiya[1,2], Matthew C Finneran[1,2], Ashley L Kalinski[1], Rafi Kohen[1,2], Corey Flynn[1], Ryan Passino[1], Craig N Johnson[1], David Kohrman[3], Riki Kawaguchi[4], Lynda JS Yang[5‡], Jeffery L Twiss[6], Daniel H Geschwind[7,8,9], Gabriel Corfas[2,3,10], Roman J Giger[1,2,10]*

[1]Department of Cell and Developmental Biology, University of Michigan-Ann Arbor, Ann Arbor, United States; [2]Neuroscience Graduate Program, University of Michigan–Ann Arbor, Ann Arbor, United States; [3]Kresge Hearing Institute, University of Michigan–Ann Arbor, Ann Arbor, United States; [4]Departments of Psychiatry and Neurology, University of California, Los Angeles, Los Angeles, United States; [5]Department of Neurosurgery, University of Michigan-Ann Arbor, Ann Arbor, United States; [6]Department of Biological Sciences, University of South Carolina, Columbia, United States; [7]Department of Neurology, Program in Neurogenetics, David Geffen School of Medicine, University of California, Los Angeles, Los Angeles, United States; [8]Department of Human Genetics,David Geffen School of Medicine, University of California, Los Angeles, Los Angeles, United States; [9]Institute of Precision Health, University of California, Los Angeles, Los Angeles, United States; [10]Department of Neurology, University of Michigan–Ann Arbor, Ann Arbor, United States

*For correspondence:
rgiger@umich.edu

[†]These authors contributed equally to this work

[‡]Deceased

**Abstract** Upon trauma, the adult murine peripheral nervous system (PNS) displays a remarkable degree of spontaneous anatomical and functional regeneration. To explore extrinsic mechanisms of neural repair, we carried out single-cell analysis of naïve mouse sciatic nerve, peripheral blood mono-nuclear cells, and crushed sciatic nerves at 1 day, 3 days, and 7 days following injury. During the first week, monocytes and macrophages (Mo/Mac) rapidly accumulate in the injured nerve and undergo extensive metabolic reprogramming. Proinflammatory Mo/Mac with a high glycolytic flux dominate the early injury response and rapidly give way to inflammation resolving Mac, programmed toward oxidative phosphorylation. Nerve crush injury causes partial leakiness of the blood–nerve barrier, proliferation of endoneurial and perineurial stromal cells, and entry of opsonizing serum proteins. Micro-dissection of the nerve injury site and distal nerve, followed by single-cell RNA-sequencing, identified distinct immune compartments, triggered by mechanical nerve wounding and Wallerian degeneration, respectively. This finding was independently confirmed with *Sarm1[-/-]* mice, in which Wallerian degeneration is greatly delayed. Experiments with chimeric mice showed that wildtype immune cells readily enter the injury site in *Sarm1[-/-]* mice, but are sparse in the distal nerve, except for Mo. We used CellChat to explore intercellular communications in the naïve and injured PNS and report on hundreds of ligand–receptor interactions. Our longitudinal analysis represents a new resource for neural tissue regeneration, reveals location-specific immune microenvironments, and reports on large intercellular communication networks. To facilitate mining of scRNAseq datasets, we generated the injured sciatic nerve atlas (iSNAT): https://cdb-rshiny.med.umich.edu/Giger_iSNAT/.

## Editor's evaluation

Zhao et al. provide an extensive transcriptomic analysis, using single-cell RNAseq, of the injured sciatic nerve, assessing both temporal and spatial differences as well as active communication networks between cells. They focus particularly on immune cells in the nerve and demonstrate shifts in the various populations and metabolic state over time and position relative to the injury. Collectively their findings suggest that Wallerian degeneration is critical for recruiting and maturation of immune cells distal to the injury but not at the injury site itself. These results provide a valuable resource and important insights for understanding inflammation, regeneration, and pain responses following nerve injury.

## Introduction

Axonal injury caused by trauma or metabolic imbalances triggers a biochemical program that results in axon self-destruction, a process known as Wallerian degeneration (WD). In the peripheral nervous system (PNS), WD is associated with nerve fiber disintegration, accumulation of myelin ovoids, reprogramming of Schwann cells (SC) into repair (rSC), and massive nerve inflammation. Fragmented axons and myelin ovoids are rapidly cleared by rSC and professional phagocytes, including neutrophils and macrophages (Mac) (*Jang et al., 2016*; *Klein and Martini, 2016*; *Perry and Brown, 1992*; *Rotshenker, 2011*). In the mammalian PNS, timely clearance of fiber debris during WD stands in stark contrast to the central nervous system (CNS), where upon injury, clearance of degenerated axons and myelin is protracted and accompanied by prolonged inflammation (*Bastien and Lacroix, 2014*; *Vargas and Barres, 2007*).

Successful PNS regeneration depends upon the coordinated action and communication among diverse cell types, including fibroblast-like structural cells, vascular cells, SC, and different types of immune cells (*Cattin et al., 2015*; *Chen et al., 2021*; *Girouard et al., 2018*; *Pan et al., 2020*; *Stratton et al., 2018*). Nerve trauma not only results in fiber transection, but also causes necrotic cell death and release of intracellular content at the site of nerve injury. Depending on severity, physical nerve trauma results in vasculature damage, endoneurial bleeding, breakdown of the blood–nerve barrier (BNB), and tissue hypoxia (*Cattin et al., 2015*; *Rotshenker, 2011*). Distal to the injury site, where mechanical damage is not directly experienced, severed axons undergo WD. The vast majority of immune cells in the injured PNS are blood-borne myeloid cells, including neutrophils, monocytes (Mo), and Mac (*Kalinski et al., 2020*; *Ydens et al., 2020*). Entry into the injured nerve causes rapid activation and acquisition of specific phenotypes. A comparative analysis between circulating immune cells in peripheral blood, and their descends in the injured nerve, however, has not yet been carried out.

It is well established that the immune system plays a key role in the tissue repair response (*Bouchery and Harris, 2019*). Following PNS injury, the complement system (*Ramaglia et al., 2007*) and natural killer cells (NK) promote WD of damaged axons (*Davies et al., 2019*). Mac and neutrophils phagocytose nerve debris, including degenerating myelin and axon remnants (*Chen et al., 2015*; *Kuhlmann et al., 2002*; *Lindborg et al., 2017*; *Rosenberg et al., 2012*). Mac protect the injured tissue from secondary necrosis by clearing apoptotic cells through phagocytosis, a process called efferocytosis (*Boada-Romero et al., 2020*; *Greenlee-Wacker, 2016*; *Lantz et al., 2020*). In the injured sciatic nerve, efferocytosis readily takes place and is associated with inflammation resolution (*Kalinski et al., 2020*). The highly dynamic nature of intercellular communications, coupled with spatial differences in the nerve microenvironment, make it difficult to untangle the immune response to nerve trauma from the immune response to WD. While previous studies have employed single-cell RNA sequencing (scRNAseq) to describe naïve or injured PNS tissue (*Carr et al., 2019*; *Kalinski et al., 2020*; *Wang et al., 2020*; *Wolbert et al., 2020*; *Ydens et al., 2020*), a longitudinal study to investigate transcriptomic changes and cell–cell communication networks has not yet been carried out. Moreover, a comparative analysis of cell types and transcriptional states at the nerve injury site versus the distal nerve, does not yet exist.

To better understand cellular functions, immune cell trafficking, injury-induced changes in gene expression, and intercellular communication, we carried out a longitudinal scRNAseq study, including three different post-injury time points. Changes in cell numbers and gene expression were independently validated by flow cytometry, ELISA, in situ hybridization with RNAscope, and immunofluorescence labeling of nerve tissue sections. Peripheral blood mononuclear cells (PBMCs) from naïve

mice were analyzed by scRNAseq to identify leukocytes that enter the injured nerve. Differential gene expression analysis was used to determine injury-regulated genes, signaling pathways, metabolic states, and cellular functions. To distinguish between the immune response to nerve trauma and WD, we employed *Sarm1* null (*Sarm1*⁻/⁻) mice in which WD is greatly delayed (*Osterloh et al., 2012*). We show that in injured *Sarm1*⁻/⁻ mice blood-borne Mac rapidly accumulate at the nerve crush site. However, in the absence of WD, inflammation in the distal nerve is greatly reduced. While monocytes are found in the distal nerve of *Sarm1*⁻/⁻ mice at 7 days following injury, they fail to mature into Mac. In sum, we report on temporal changes in the cellular and metabolic landscape of the injured mammalian PNS, identify spatially distinct, yet overlapping immune responses to nerve wounding and WD, and provide insights into WD-elicited nerve inflammation.

## Results

### The cellular and molecular landscape of naïve sciatic nerve

To gain insights into extrinsic mechanisms associated with neural tissue degeneration and regeneration, we carried out a longitudinal analysis of injured mouse sciatic nerves using single-cell transcriptomics. We first captured CD45⁺ immune cells from sciatic nerve trunk, and in parallel, CD45⁻ nonimmune cells for scRNAseq (*Figure 1A*). The resulting datasets were combined, and high-quality cells identified (*Supplementary file 1*) and subjected to dimensional reduction using the top 30 principal components. Uniform manifold approximation and projection (UMAP) was used for visualization of cell clusters and clusters determined using the Louvain algorithm with a resolution of 0.5, revealing 24 clusters (*Figure 1B*). Marker gene expression analysis was used for cell-type identification (*Figure 1C*). Stromal cells, also referred to as structural cells, are abundant in the naïve nerve. They include epineurial fibroblasts (Fb1-Fb3, clusters 8–10), identified by strong expression of *Pcolce2*/procollagen C-endopeptidase enhance 2 (*Figure 1D*), endoneurial mesenchymal cells (eMES1-eMES3, clusters 11–13), strongly expressing the tumor suppressor *Cdkn2a*/cyclin-dependent kinase inhibitor 2A (*Figure 1E*), and perineurial (p)MES in clusters 14 and 15, expressing the tight junction molecule *Cldn1*/claudin 1 (*Figure 1F*).

Cells that belong to the nerve vasculature include *Pecam1*/CD31⁺ endothelial cells (EC1-EC3, clusters 17–19) that can be subdivided into venous EC (clusters 17 and 18) marked by *Car4*⁺/carbonic anhydrase 4 expression and arterial EC (cluster 19), expressing *Fbln5*⁺/fibulin 5. EC express high levels of adherent junction (*Cldn5*/claudin 5), tight junction (*Tjp1*), and junctional adhesion molecules (*Jam2*), key components of the BNB (*Figure 1—figure supplement 1A*; *Maiuolo et al., 2019*). Clusters 21 and 22 are connected and harbor *Notch3*⁺ mural cells (*Figure 1H*) that can be further subdivided into vascular smooth muscle cells (vSMC, *Acta2*⁺) and pericytes (PC, *Pdgfrb*⁺) (*Figure 1—figure supplement 1A*). A small island (cluster 20) with lymphatic endothelial cells (LyEC, *Prox1*) was detected (*Figure 1—figure supplement 1A*). Schwann cells (SC, *Sox10*) are mainly comprised of non-myelinating (nm)SC (cluster 23) and few myelinating (m)SC (cluster 24) (*Figure 1I*; *Gerber et al., 2021*). The low number of mSC indicates that many are lost during the cell isolation process from naïve nerve.

Cells in clusters 0, 1, and 2 are tissue-resident Mac (*Figure 1B*). They are the most abundant immune cell type in the naïve nerve, representing 83% of all leukocytes. Smaller clusters with immune cells include cluster 3 with dendritic cells (DC, 4%), cluster 4 with mast cells (Mast, 6%), and cluster 5 with T cells and natural killer cells (TC and NK, 6%) (*Figure 1B*, *Figure 1—figure supplement 1B*). Less than 1% of nerve resident immune cells express markers for granulocytes (GC, *Cxcr2*/C-X-C motif chemokine receptor 2), monocytes (Mo, *Chil3*/chitinase-like protein 3), or B cells (BC, *Cd79a*/B cell antigen receptor-associated protein), indicating these cell types are sparse in the naïve PNS of healthy mice. Tissue resident Mac in the naïve PNS strongly express the scavenger receptor cysteine-rich type 1 (SCARI1), encoded by *Cd163*. A small group of *Fn1* (fibronectin) expressing Mac (cluster 0) was detected, possibly representing leukocytes that recently entered the nerve (*Figure 1—figure supplement 2A and B*).

Previous scRNAseq studies employed cell sorting to isolate Mac from naïve PNS by CD45⁺CD64⁺F4/80⁺ (*Ydens et al., 2020*) or CD45^int CD64⁺ antibody sorting (*Wang et al., 2020*), and reported transcriptionally distinct epineurial (epi) Mac and endoneurial (endo) Mac. When compared to our datasets, we find that cells in cluster 1 express *Fcna*/Ficolin-A and *Ccl8*/monocyte chemotactic protein 2, and thus, represent epiMac (*Figure 1J*, *Figure 1—figure supplement 2C*). In addition,

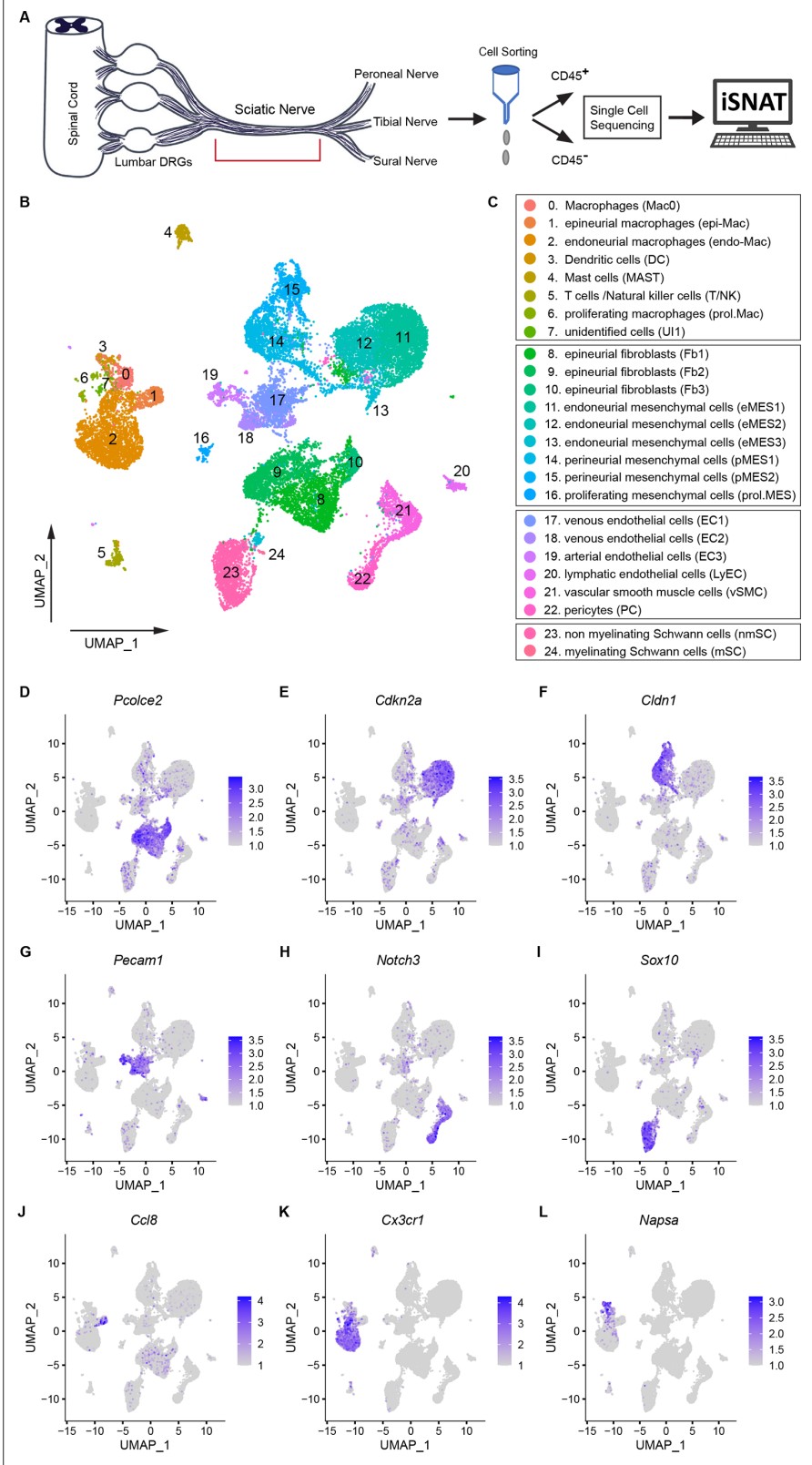

**Figure 1.** The cellular and molecular landscape of naïve mouse sciatic nerve. (**A**) Schematic of workflow for peripheral nervous tissue analysis. Cartoon of a mouse spinal cord, lumbar dorsal root ganglia (DRGs), and sciatic nerve trunk with main branches. The nerve segment marked with a red bracket was harvested for further analysis. Immune cells were captured with anti-CD45 conjugated magnetic beads. The flow through, containing

*Figure 1 continued on next page*

*Figure 1 continued*

non-immune (CD45⁻) cells, was collected as well. In separate scRNAseq runs, CD45⁺ and CD45⁻ single-cell transcriptomes were determined. A total of 21,973 high-quality transcriptomes, including 4539 CD45⁺ cells and 17,434 CD45⁻ cells were generated and used for downstream analysis. (**B**) UMAP embedding of naïve sciatic nerve cells. Unsupervised Seurat-based clustering identified 24 clusters. (**C**) List of cell types identified in the naïve sciatic nerve, grouped into immune cells (clusters 0–7), structural/stromal cells (clusters 8–16), cells associated with the nerve vasculature (clusters 17–22), and Schwann cells (clusters 23 and 24). (**D–L**) Feature plots of scRNAseq data showing expression of canonical markers used for assignment of major cell types, including epineurial fibroblasts (*Pcolce2/*procollagen C-endopeptidase enhancer 2), endoneurial MES (*Cdkn2a/*cyclin dependent kinase inhibitor 2A), perineurial MES (*Cldn1/*claudin-1), endothelial cells (*Pecam1/*CD31), vascular smooth muscle cells and pericytes (*Notch3/*notch receptor 3), Schwann cells (*Sox10/*SRY-box transcription factor 10), epineurial Mac (*Ccl8/*C-C motif chemokine ligand 8), endoneurial Mac (*Cx3cr1/*C-X3-C motif chemokine receptor 1), and dendritic cells (*Napsa/*napsin A, aspartic peptidase). Expression levels are color coded and calibrated to average gene expression.

The online version of this article includes the following source data and figure supplement(s) for figure 1:

**Figure supplement 1.** Cell-type-enriched gene products in the naïve mouse sciatic nerve.

**Figure supplement 1—source data 1.** X-ray films of cytokine ELISA membranes probed with serum or sicatic nerve lysate of naive mice.

**Figure supplement 2.** Markers for nerve resident macrophages.

**Figure supplement 3.** Validation of naïve nerve scRNAseq dataset by ELISA.

---

epiMac are strongly enriched for the C-type lectin members *Cd209f, Cd209g, Cd209d.* The expression of *Ccl8* by epiMac was validated by in situ hybridization using RNAscope. *Ccl8* labeling was highest in Mac located in the epineurium (***Figure 1—figure supplement 2G and G'***). Cells represented by cluster 2 preferentially express fractalkine receptor (*Cx3cr1),* triggering receptor expressed on myeloid cell 2 (*Trem2*), and leukocyte Ig-like receptor A5 (*Lilra5*) and represent endoMac. The toll-like receptor signaling regulator *Unc93b1* (Unc-93 homolog B1) is a pan-myeloid cell marker for naive sciatic nerve (***Figure 1K***, ***Figure 1—figure supplement 2D–F***). Longitudinal sciatic nerve sections of *Cx3cr1-GFP* reporter mice confirmed labeling of endoMac (***Figure 1—figure supplement 2H and H'***).

Mac are educated by the local nerve microenvironment and acquire niche-specific phenotypes. In the naïve PNS, epiMac (*Lyve1ʰⁱ, Cd209ʰⁱ, Cx3cr1ˡᵒ*) are located in the heavily vascularized epineurium and resemble perivascular Mac (***Chakarov et al., 2019***), while endoMac exhibit neural niche gene signatures reminiscent of Mac in the enteric plexus (***De Schepper et al., 2019***) and microglia (***Wang et al., 2020***). There are few DC (*Napsa/*aspartic peptidase) in the naïve nerve, likely functioning as sentinels, cells specialized in uptake and presentation of antigens to facilitate adaptive immunity (***Figure 1L***). Mast cells in the naïve nerve are readily identified by their preferential expression of *Fcer1a* (immunoglobulin epsilon receptor for IgE) and *Cpa3* (carboxypeptidase A) (***Figure 1—figure supplement 1B***).

## Identification of serum proteins that enter the naïve PNS

To independently validate naïve nerve scRNAseq datasets, we examined the presence of some of the corresponding proteins in nerve lysate by ELISA. We used an ELISA kit that allows simultaneous profiling of 111 extracellular proteins, including cytokines, chemokines, growth factors, proteases, and protease inhibitors (***Figure 1—figure supplement 3A***). Proteins abundantly detected include, FGF1 (encoded by *Fgf1*), IGF binding protein 6 (*Igfbp6*), coagulation factor III (*F3*), endostatin (*Col18a1*), interleukin 33 (*Il33*), fetuin A/alpha 2-HS glycoprotein (*Ahsg*), osteopontin/OSP (*Spp1*), cystatin C (*Cst3*), matrix metallopeptidase 2 (*Mmp2*), resistin/adipose tissue-specific secretory factor (*Retn*), retinoic acid binding protein 4 (*Rbp4*), CCL11 (*Ccl11*), low-density lipoprotein receptor (*Ldlr*), adiponectin (*Adipoq*), intercellular adhesion molecule 1 (*Icam1*), and serpin F1/neurotrophic protein (*Serpinf1*) (***Figure 1—figure supplement 3B***).

For many of the proteins detected by ELISA, the corresponding transcripts are found in the nerve scRNAseq dataset (***Figure 1—figure supplement 3C***). However, for some abundantly detected proteins, including fetuin A, resistin, and adiponectin, the corresponding transcripts (*Ahsg, Retn, Adipoq*) were not detected by scRNAseq (***Figure 1—figure supplement 3C***). Resistin and adiponectin are both secreted proteins produced by adipocytes, and fetuin A is secreted by hepatocytes. All three

proteins are known serum components, and thus, may enter the nerve via the circulatory system. Indeed, analysis of serum revealed high levels of fetuin A, resistin, and adiponectin (*Figure 1—figure supplement 3D and E*). Importantly, some of the most abundant serum proteins, proprotein convertase 9 (encoded by *Pcsk1*), selectin E (*Sele*), IGFBPs (*Igfbp2, 3, and 5*), soluble C1qR1 (*Cd93*), angiopoietins (*Angpt2, Angpt1*), C-C motif chemokine ligand 21 (*Ccl21*), and CRP/C-reactive protein (*Crp*) are not present in naïve nerve lysates, providing confidence that mice were perfused properly, and nerve specimens not contaminated with serum (*Figure 1—figure supplement 3E*). Thus, our experiments identify select serum proteins that can cross the BNB. Collectively, this work establishes a baseline of the cellular landscape and molecular milieu of naïve PNS. These data will be used as reference to determine how a crush injury alters the nerve microenvironment.

## The injured sciatic nerve atlas (iSNAT)

To capture the dynamic nature of cellular and molecular changes in the injured PNS tissue, we subjected adult mice to sciatic nerve crush injury (SNC) and carried out scRNAseq at different time points, including 1 day post-SNC (1dpc), 3dpc, and 7dpc (*Figure 2A*). *Supplementary file 1* shows the number of scRNAseq runs and cells analyzed at each time point. UMAP embedding was used for visualization of cell clusters and clusters determined using the Louvain algorithm with a resolution of 0.5 (*Figure 2*). Marker gene expression analysis was used for cell-type identification and revealed gene products that allow reliable identification of major cell types in naïve nerve and at different post injury time points (*Figure 2—figure supplement 1A–D*). SNC triggers massive nerve infiltration of blood-borne immune cells (*Kalinski et al., 2020*; *Ydens et al., 2020*). In the 1dpc UMAP plot, we identified granulocytes (GC, *Cxcr2*) in cluster 11 (*Figure 2D*), Mo (*Ly6c2*/lymphocyte antigen 6C2 and *Chil3*/Ym1) in cluster 0 (*Figure 2E*, *Figure 2—figure supplement 1B*), and five Mac subpopulations (Ma-I to Mac-V) in clusters 1–5, expressing *Adgre1*(F4/80) (*Figure 2G*). GC are primarily composed of neutrophils (*Retnlg*/resistin-like gamma, *Grina*/NMDA receptor-associated protein 1), intermingled with a small population of eosinophils (*Siglecf*/sialic acid binding Ig-like lectin F) (*Figure 2F*). Mo that recently entered the nerve can be identified by high levels of *Cd177*, a known surface molecule that interacts with the EC adhesion molecule *Pecam1*/CD31 and promotes transmigration. Mo strongly express *Ly6c2* and *Ccr2*, suggesting that they represent 'classical' Ly6C$^{hi}$ proinflammatory cells; however, they do not express inducible nitric oxide synthase (*Nos2*). In the UMAP plot of 1dpc nerve, cluster 0 (Mo) has a volcano-like shape connected at the top to cluster 1, harboring Mac-I (*Ccr2$^{hi}$, Ly6c2$^{low}$, Cx3cr1$^-$*), indicating these are maturing monocyte-derived Mac (*Figure 2B*). Four smaller Mac clusters include Mac-II (*C1qa*, complement factor C1q), Mac-III (*Ltc4s*/leukotriene C4 synthase, *Mmp19*/matrix metallopeptidase 19), Mac-IV expressing MHCII genes (*H2-Aa, H2-Eb1, H2-Ab1, Cd74*), and Mac-V (*Il1rn*/IL1 receptor antagonist, *Cd36*/ECM receptor, coagulation factors *F7* and *F10*), suggesting roles in opsonization, matrix remodeling, antigen presentation, and coagulation (*Figure 2B*, and data not shown). Cluster 6 harbors monocyte-derived dendritic cells (MoDC: *Cd209a*/DC-SIGN, *H2-Aa*/MHC class II antigen A, alpha), professional antigen-presenting cells (*Figure 2H and I*). A small island (cluster 13) harbors a mixture of TC and NK (*Figure 2B*). The 1dpc dataset contains five clusters with structural cells (Fb, differentiating (d)MES, eMES, and pMES), including a small number of proliferating (*Mki67$^+$*/Ki67) cells, vasculature cells, EC (cluster 20), and vSMC and PC (cluster 21) (*Figure 2B*, *Figure 2—figure supplement 2B*).

UMAP embedding of 3dpc nerve cells (*Figure 2J*) indicates that there are fewer GC (*Cxcr2*) (*Figure 2L*) and Mo (*Ly6c2*)(*Figure 2M*) than in the 1dpc nerve (*Figure 2B*). *Adgre1$^+$* Mac are the most dominant immune cell type at 3dpc, comprised of five subpopulations (clusters 1–5) designated Mac1-Mac4, and prol. Mac (*Figure 2O*). *Ccr2* labels most Mac and DC, except for Mac4 and a subset of Mac1 cells (*Figure 2N*). As discussed below, there is no one-to-one match of Mac subclusters identified in 1dpc and 3dpc nerves, likely owing to their high degree of transcriptional plasticity. To emphasize this observation, we labeled Mac in the 1dpc nerve as MacI-MacV and Mac in the 3dpc nerve as Mac1-Mac4 and prol.Mac (*Figure 2C and K*). MoDC (*Cd209a*) are readily detected in the 3dpc nerve (*Figure 2P*). Compared to the 1dpc nerve, the fraction of MHCII$^+$ (*H2-Aa*) cells is increased, including four subpopulations of dendritic cells (MoDC, cDC, DCx, and pDC, see below for details), as well as subpopulations of MHCII$^+$ Mac (*Figure 2Q*). Structural cells begin to proliferate heavily at 3dpc, suggesting that a nerve crush injury causes substantial damage to the epineurium, perineurium, and endoneurium (*Figure 2—figure supplement 2C*). SNC is known to inflict vascular

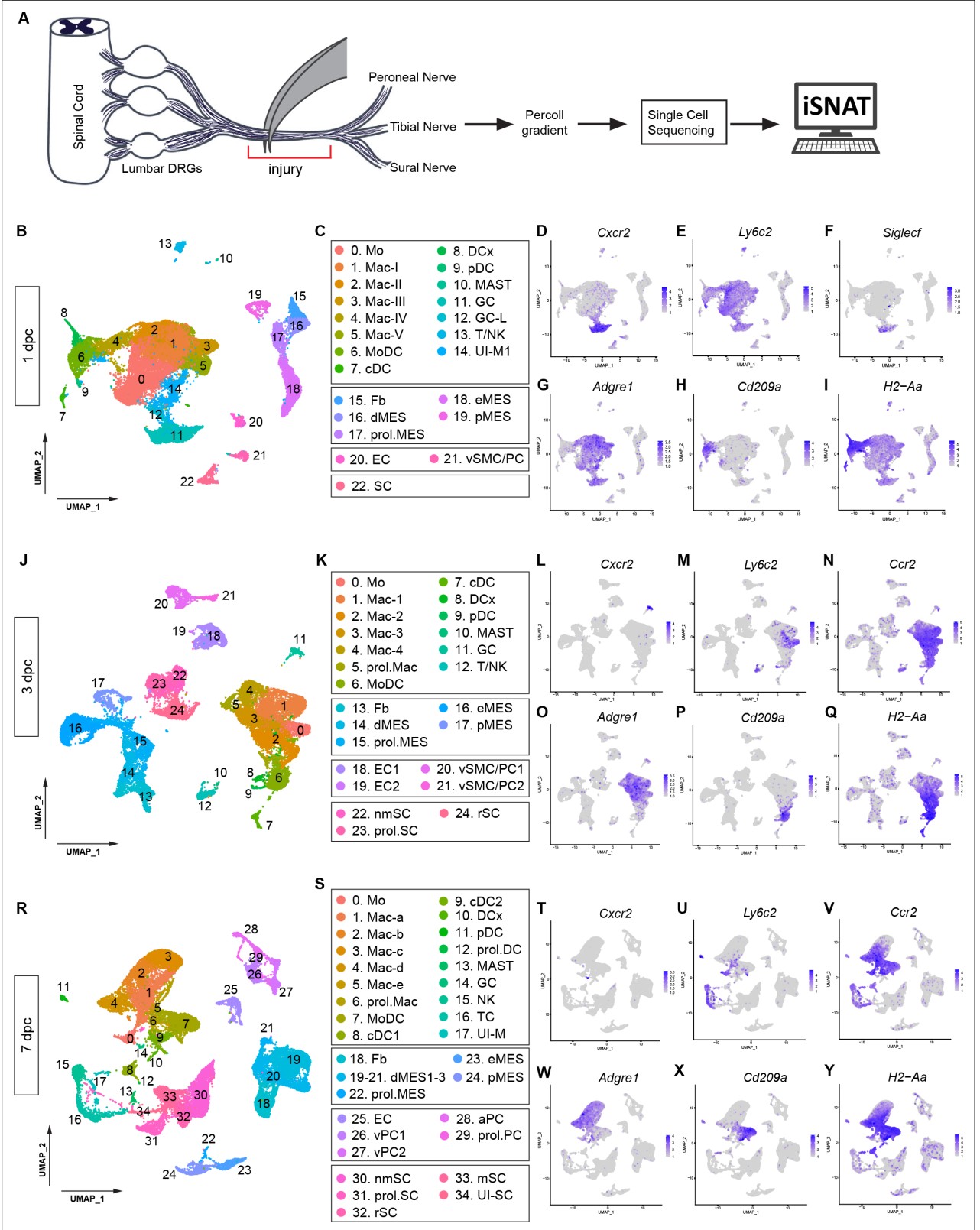

**Figure 2.** Longitudinal analysis of single-cell transcriptomes of injured peripheral nervous system (PNS). (**A**) Schematic of workflow for single-cell analysis of injured mouse sciatic nerve trunk. Cartoon of lumbar spinal cord with dorsal root ganglia (DRGs) and sciatic nerve. The nerve injury site is shown, and the segment marked with the red bracket harvested at different post-injury time points for analysis by scRNAseq. (**B**) UMAP plot embedding of sciatic nerve cells at 1dpc. A total of 29,070 high-quality cells (n = 5 replates) were subjected to unsupervised Seurat-based clustering resulting in

*Figure 2 continued on next page*

*Figure 2 continued*

22 clusters. (**C**) List of cell types identified, grouped into immune cells (clusters 0–14), structural cells (clusters 15–19), cells associated with the nerve vasculature (clusters 20 and 21), and Schwann cells (cluster 22). (**D–I**) Feature plots of canonical immune cell markers reveals clusters that harbor GC (*Cxcr2*), including a subset of eosinophils (*Siglecf*/sialic acid binding Ig-like lectin F), Mo (*Ly6c2*/Ly6C), Mac (*Adgre1*/F4/80), MoDC (*Cd209a*/DC-SIGN), and antigen-presenting myeloid cells (*H2-Aa*/histocompatibility 2, class II antigen A, alpha). (**J**) UMAP plot embedding of sciatic nerve cells at 3dpc. A total of 24,672 high-quality cells (n = 6 replates) were subjected to unsupervised Seurat-based clustering, resulting in 24 cell clusters. (**K**) List of cell types in the 3-day injured nerve, grouped into immune cells (clusters 0–12), structural cells (clusters 13–17), cells associated with the nerve vasculature (clusters 18–21), and Schwann cells (clusters 22–24). (**L–Q**) Feature plots of canonical markers for immune cells to identify clusters with GC (*Cxcr2*), Mo (*Ly6c2*), Mo/Mac (*Ccr2*), Mac (*Adgre1*), MoDC (*Cd209a*), and antigen-presenting cells (H2–Aa). (**R**) UMAP plot embedding of sciatic nerve cells at 7dpc. A total of 32,976 high-quality cells (n = 8 replates) were subjected to unsupervised Seurat-based clustering resulting in 34 cell clusters. (**S**) List of cell types in the nerve at 7dpc, grouped into immune cells (clusters 0–17), structural cells (clusters 18–24), cells associated with the nerve vasculature (clusters 25–29), and Schwann cells (clusters 30–34). (**T–Y**) Feature plots of canonical markers for immune cells revealed GC (*Cxcr2*), Mo (*Ly6c2*), Mo/Mac (*Ccr2*), Mac (*Adgre1*), MoDC (*Cd209a*), and antigen-presenting cells (H2–Aa). Expression levels are projected onto UMAP with a minimum expression cutoff of 1. Abbreviations: Mo, monocytes; Mac, macrophages; prol.Mac, proliferating macrophages; MoDC, monocyte-derived dendritic cells; cDC, conventional dendritic cells; DCx mature/migrating dendritic cells; pDC, plasmacytoid dendritic cells; MAST, mast cells; GC, granulocytes (including neutrophils and eosinophils), GC-L, granule cell-like; TC, T cells; NK, natural killer cells. Abbreviations for stromal cells: Fb, fibroblast; dMES, differentiating mesenchymal cells; prol.MES, proliferating mesenchymal cells; eMES, endoneurial mesenchymal cells; pMES, perineurial mesenchymal cells. Abbreviations for vascular cells: EC, endothelial cells, vSMC, vascular smooth muscle cells; PC, pericytes; vPC, venous pericytes; aPC, arterial pericytes; prol.PC, proliferating pericytes. Abbreviations for Schwann cells, nmSC, non-myelinating Schwann cells; mSC myelinating Schwann cells; rSC, repair Schwann cells; prol.SC, proliferating Schwann cells. UI, unidentified cells.

The online version of this article includes the following figure supplement(s) for figure 2:

**Figure supplement 1.** Identification of marker genes for cell type identification in the naïve and injured peripheral nervous system (PNS).

**Figure supplement 2.** Quality test of scRNAseq datasets and identification of proliferating cells in the naïve and injured sciatic nerve.

**Figure supplement 3.** Cell composition on naïve and injured sciatic nerve.

---

damage and breach of the BNB. This is underscored by the strong proliferative response of vascular cells, including EC, vSMC, and PC (*Figure 2—figure supplement 2C*).

The UMAP plot generated from 7dpc nerve cells (*Figure 2R*) reveals that GC and Mo have further declined (*Figure 2T and U*), while *Adgre1*⁺ remain prevalent (*Figure 2W*). The 7dpc Mac subpopulations form a connected continuum of five clusters, harboring Mac-a to Mac-e (*Figure 2S*). *Ccr2* remains high in most Mac subpopulations, except for Mac-c and is also observed in Mo and DC (*Figure 2V*). Similar to earlier time points, there is no clear one-to-one match of Mac subpopulations in 7dpc and 3dpc nerves, and thus, we labeled 7dpc Mac populations as Mac-a to Mac-e (*Figure 2R*). At 7dpc, the number of *Cd209a*⁺DC has increased compared to earlier time points (*Figure 2X*). MHCII (*H2-Aa*) expressing Mac and DC are readily detected (*Figure 2Y*). Lymphoid cells, identified by *Il2rb*/IL2 receptor subunit β expression, include NK (cluster 15) and TC (cluster 16) (*Figure 2R*). At 7dpc, the proliferation of structural and vascular cells is reduced compared to 3dpc (*Figure 2—figure supplement 2C and D*), suggesting that cells required for the repair process and wound healing are in place. The dynamic nature of cell types that participate in the repair process, and their proportion in naïve, 1dpc, 3dpc, and 7dpc nerves, is demonstrated in a composition plot (*Figure 2—figure supplement 3*).

The sequelae of successful tissue repair require extensive communication between nerve resident cells and hematogenous immune cells. For a detailed description of the cellular and molecular changes that occur during the first week following PNS injury, we generated and analyzed more than 150,000 high-quality single-cell transcriptomes (*Supplementary file 1*). These datasets provide a new resource for understanding cell function and molecules in the degenerating and regenerating mammalian PNS. For widespread distribution of scRNAseq data, we created the iSNAT, a user-friendly and freely available web-based resource. The *Expression Analysis* function in iSNAT provides a readily accessible platform for comparative analysis of gene expression in naïve and injured nerves at single-cell resolution (https://cdb-rshiny.med.umich.edu/Giger_iSNAT/).

## Insights into PNS injury-induced gene products and signaling pathway

To demonstrate the application of iSNAT toward understanding how cellular function may change during the repair process, we carried out differential gene expression (DEG) analysis by comparing cell-type-specific transcriptomes in the naïve nerve with the corresponding transcriptomes of injured nerve. Because of their profound increase upon nerve injury, we first focused on Mac by comparing

naïve nerve Mac (endoMac and epiMac) with Mac accumulating in the injured nerve. The top SNC upregulated gene products are shown in *Figure 3A*, and violin plots are shown for the chemokines *Ppbp*/Cxcl7, *Cxcl3*, *Ccl2*, *Ccl7*, *Ccl24*, and *Il1rn* (*Figure 3B–G*). Next, top regulated gene products in Mac were subjected to pathway analysis (*Figure 3H*). Significantly upregulated pathways include *glycolysis*, *pathogen-induced chemokine storm signaling pathway*, *phagosome formation*, *tumor microenvironment*, *Hif1α signaling,* and *Fcγ receptor-mediated phagocytosis*.

Because stromal cells exhibit a dynamic expression of gene products that control tissue inflammation (*Figure 3—figure supplement 1*) and are thought to play important roles in shaping the nerve immune microenvironment (*Kalinski et al., 2020*; *Li et al., 2022*), we next focused on injury-regulated DEGs in eMES (*Figure 3—figure supplement 2*), pMES (*Figure 3—figure supplement 3*), and Fb/dMES (*Figure 3—figure supplement 4*). For eMES, top injury-induced gene products include *Mt2*/metallothionein-2, *Ptx3*/pentraxin 3, *Rdh10*/retinol dehydrogenase, and *Il1r1*/IL-1 receptor type 1. Pathway analysis identified an increase in *Hif1α signaling*, *pathogen induced chemokine storm signaling pathway*, *NAD signaling pathway*, *glycoprotein VI (GP6) signaling pathway,* and *wound healing signaling pathway* (*Figure 3—figure supplement 2*). For pMES, top injury-induced gene products include *Mgp*/matrix gla protein, *Clu*/clusterin, and *Msln*/mesothelin. Pathway analysis identified an increase in *unfolded protein response*, *endoplasmic reticulum stress pathways*, *immunogenic cell death signaling pathways*, and *pathogen-induced cytokine storm signaling pathway* (*Figure 3—figure supplement 3*). For Fb/dMES, top injury-induced gene products include *Cxcl14*, *Mmp3*/matrix metallopeptidase 3, *Serpina3n*/antitrypsin, and *Slc39a14*/zinc, iron, and cadmium transporter. Pathway analysis identified *unfolded protein response*, *role of chondrocytes in rheumatoid arthritis signaling pathway*, and *S100 signaling pathway* (*Figure 3—figure supplement 4*). Many of the top DEGs following SNC encode secreted gene products, or membrane-bound proteins, indicative of highly dynamic cell–cell communication networks in the injured nerve.

## CellChat identifies large cell–cell communication networks activated during nerve repair

To better understand cell–cell communication networks during the repair process, we interrogated cell surface protein interactions using CellChat (*Jin et al., 2021*). The output of CellChat is the probability for cell–cell communications to occur via specific ligand–receptor systems. In naïve and injured nerve, CellChat identified hundreds of ligand–receptor pairs among different cell groups, which are further categorized into 64 major signaling networks. To facilitate mining of the predicted protein–protein interactions, we added CellChat as a feature to iSNAT. Prominent examples of injury-regulated networks include CXCL (*Figure 4A–H*), CCL (*Figure 4I–P*), SPP1 (osteopontin/secreted phosphoprotein 1, *Spp1 – Cd44, Spp1 – Itga4 + Itgb1, Spp1 – Itga5 + Itgb1, Spp1 - Itgav + Itgb1*), IL1 (*Il1b – Il1r1 + Ilrap*), OSM (oncostatin, *Osm – Osmr + Il6st and Osm – Lifr + Il6*st), GRN (progranulin, *Grn – Sort1*), and TGFb (*Tgfb1 – Tgfbr1 + Tgfbr2, Tgfb1 – Acvr1 +Tgfbr1, Tgfb1 – Acvr1b + Tgfbr2*).

Many of the ligand–receptor pairs identified by CellChat function in leukocyte chemotaxis (*SenGupta et al., 2019*), and thus, can be mined to identify ligand sources (sender cells) and cells expressing the corresponding receptors (receiver cells). Mechanisms that promote GC and Mo infiltration into the 1dpc nerve involve Mac-II and Mac-III as sender cells, expressing *Cxcl2, Cxcl3, Cxcl5*, and *Ppbp*/Cxcl7, activators and chemoattractants for *Cxcr2*⁺ GC. Additional senders include Mac-I expressing *Ccl2*, Mac-II expressing *Pf4*/CXCL4, *Grn*/progranulin, and Mac-V expressing *Ccl24, Spp1*, and *Il1rn*. Analysis of chemotactic receptor expression identified GC expressing *Cxcr2*, the formyl peptide receptors *Fpr1* and *Fpr2*, *Tnfrsf1a*/TNF receptor 1, *Ltb4r1*/leukotriene B4 receptor 1, and *Ccr1*. Mo express the chemotactic receptors *Ccr1, Ccr2, Ccr5,* and *Ltb4r1*, indicating that GC and Mo use overlapping, yet distinct chemotactic ligand–receptor pairs for entering the injured nerve. Many chemokines with chemotactic activity, including *Ccl2, Ccl7, Ccl9*, and *Ccl12*, show highest expression at 1dpc. At 3dpc, chemokine expression is reduced compared to peak levels, and by 7dpc have declined further, approaching steady-state levels comparable to naïve nerves (*Figure 4A–P*).

In addition to immune cells, CellChat identifies structural cells as a major hub for chemotactic factors, providing evidence for prominent immune-stroma crosstalk. For example, eMES rapidly increase the production of *Ccl2, Ccl7, Il6, Il11, Cxcl1, Cxcl5, Cxcl12, Cx3cl1, Spp1,* and *Lif*, genes identified in the DEG analysis (*Figure 3—figure supplement 1*). In addition, injured eMES show elevated expression of serum amyloid A apolipoproteins (*Saa1, Saa2, Saa3*), chemotactic molecules for GC and Mo/Mac.

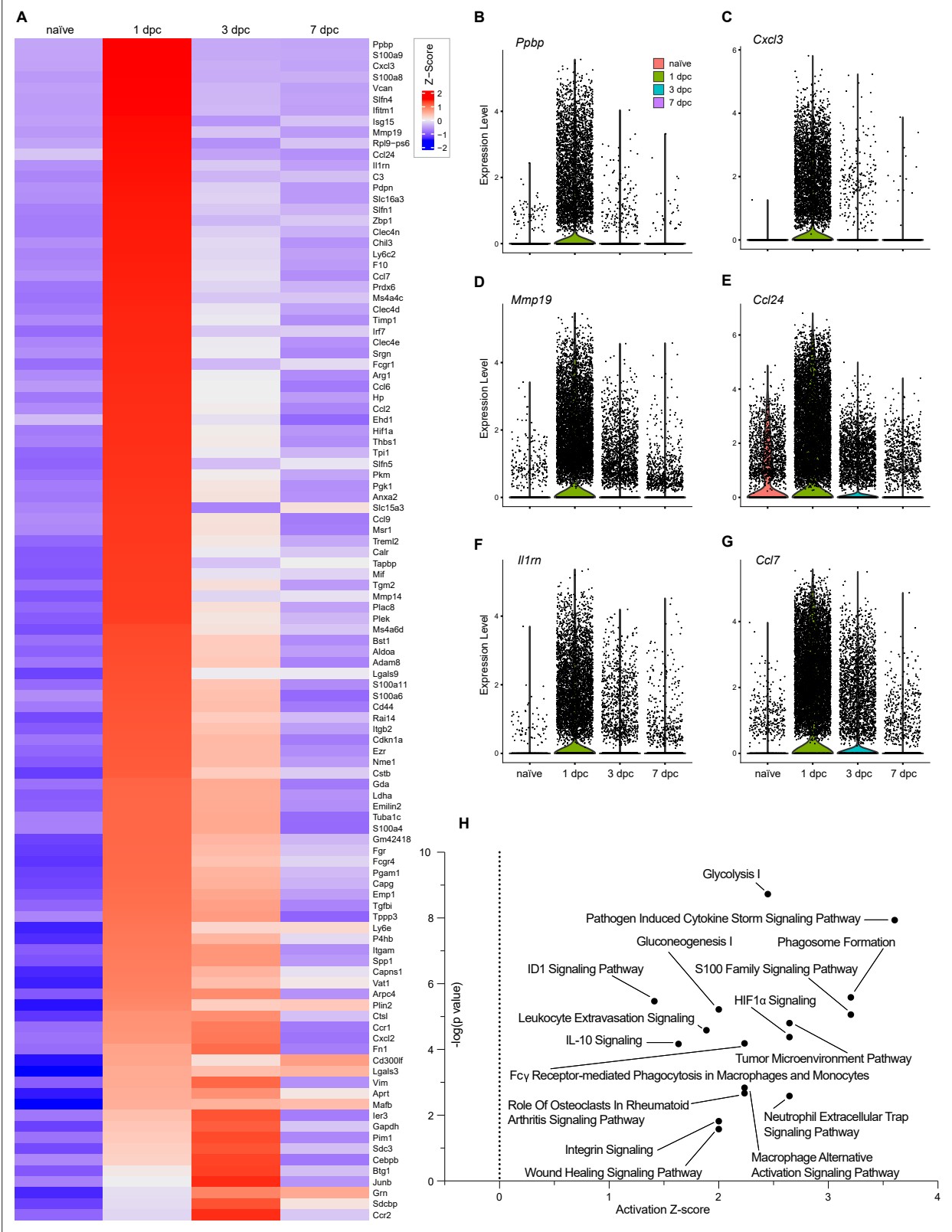

**Figure 3.** Top injury-regulated Mac gene products. (**A**) Heatmap of top 107 injury-upregulated gene products from a list of significant genes, filtered for upregulation in Mac at 1dpc and expression in at least 25% of cells. Shown are z-scores for average expression levels in naïve nerve Mac, and Mac at 1dpc, 3dpc, and 7dpc. (**B–G**) Examples of injury-induced gene products, violin plots are shown for naïve and injured nerves. (**H**) Ingenuity pathway

*Figure 3 continued on next page*

*Figure 3 continued*

analysis for injury-induced gene products in Mac utilizing the top 107 differentially expressed genes determined by Seurat's FindAllMarkers function and the Wilcoxon rank-sum test. The top-scoring enriched canonical pathways are represented through activation z-scores and p-values.

The online version of this article includes the following figure supplement(s) for figure 3:

**Figure supplement 1.** Structural cells in the injured nerve shape the local immune microenvironment.

**Figure supplement 2.** Top injury-regulated gene products in endoneurial mesenchymal cell (eMES).

**Figure supplement 3.** Top injury-regulated gene products in perineurial mesenchymal cells (pMES).

**Figure supplement 4.** Top injury-regulated gene products in fibroblasts/differentiating mesenchymal cells (Fb/dMES).

*Ptx3*/pentraxin 3, a factor implicated in wound healing (*Erreni et al., 2017*), is regulated by injury and rapidly increases in eMES and dMES at 1dpc and 3dpc (*Figure 3—figure supplements 1 and 2*). Distinct immune molecules are expressed by pMES, including *Serping1* (complement C1 inhibiting factor), *Cfh* (complement factor H), and *Lbp* (lipopolysaccharide binding protein) (*Figure 3—figure supplement 3*). Upon injury, dMES upregulate *Ccl2, Ccl7, Cxcl1, Cxcl5, Cxcl12,* underscoring the importance of diverse stromal cells in orchestrating the immune response (*Figure 3—figure supplement 1* and *Figure 3—figure supplement 4*).

Because SNC inflicts vascular damage, we searched CellChat for injury-induced angiogenic signaling pathways; top hits include VEGF, ANGPT (angiopoietin), ANGPTL (angiopoietin-like proteins), SEMA3 (semaphorin 3), EPHA, and EPHB (ephrin receptors). Independent evidence for angiogenic sprouting is the upregulation of endothelial NO synthase (*Nos3*) in EC of the injured nerve (*Smith et al., 2021*). While the importance of angiogenesis and angiogenic factors in PNS repair has been appreciated, CellChat informs on specific ligand–receptor systems and shows which cell types produce these factors and how expression is regulated upon nerve injury.

## Validation of sciatic nerve injury-regulated extracellular gene products

CellChat predicts a large number of ligand–receptor interactions to occur in the injured nerve. We used ELISA to probe nerve lysates to ask whether some of the corresponding extracellular proteins are present at 1, 3, and 7dpc (*Figure 4—figure supplement 1A–C*). ELISA data from naïve nerve tissue and serum were used for comparison (*Figure 1—figure supplement 3*). Proteins strongly upregulated by SNC include chemokines (CCL6, CCL11, CCL21, CXCL2, LIX, and osteopontin), serpin family members (Serpin E1/PAI-1, Serpin F1/PEDF), ECM degrading matrix metallopeptidases (MMP9, MMP3, MMP2), IGF binding proteins (IGFBP2, IGFBP3, IGFBP5, IGFBP6), and proteins that regulate angiogenesis (Col18a1/Endostatin, Eng/endoglin, ADIPOQ/adiponectin, FGF1), opsonization and phagocytosis, including pentraxins (NPTX2, PTX3, CRP), AHSG/Fetuin-A, and LDLR/low-density lipoprotein receptor (*Figure 4—figure supplement 1D–F*).

To examine whether the corresponding transcripts are expressed in the injured nerve, and to identify the cellular sources, we generated dotplots from scRNAseq datasets. For the top 25 proteins detected by ELISA, there is a close correlation at the transcriptional level with many gene products strongly expressed by structural cells (*Figure 4—figure supplement 1G–I*). Some of the top gene products detected by ELISA but not by scRNAseq are serum proteins (*Figure 1—figure supplement 3D*). They include Nptx2 (pentraxin), AHSG (fetuin A), Mpo (myeloperoxidase), CRP (C reactive protein), IGFBP-2, Reg3g (regenerating family member 3γ), Adipoq (adiponectin), and CFD (complement factor D). CRP is of interest because it binds to the surface of dead cells and activates complement C1q (*Alnaas et al., 2017*). Collaboration between CRP and C1q is supported by the strong expression of C1q components (*C1qa, C1qb,* and *C1qc*) in most Mac subpopulations in the injured nerve. CFD is a serine protease required for the formation of C3 convertase. Soluble C1qR1/CD93 functions as a bridging molecule that aids apoptotic cell binding to professional phagocytes for removal by efferocytosis (*Blackburn et al., 2019*). Taken together, we provide validation for many injury-regulated gene products identified by scRNA-seq and show that serum proteins that function in opsonization feature prominently in the injured nerve. Interestingly, some proteins abundant in serum (Angpt1, Angpt2, Angptl3, Areg, IGFBP-1) are sparce in the injured nerve parenchyma (*Figure 4—figure supplement 2*). This suggests that these proteins are either rapidly degraded or that SNC causes a partial breakdown of the BNB, allowing only some serum proteins to enter injured nerve tissue.

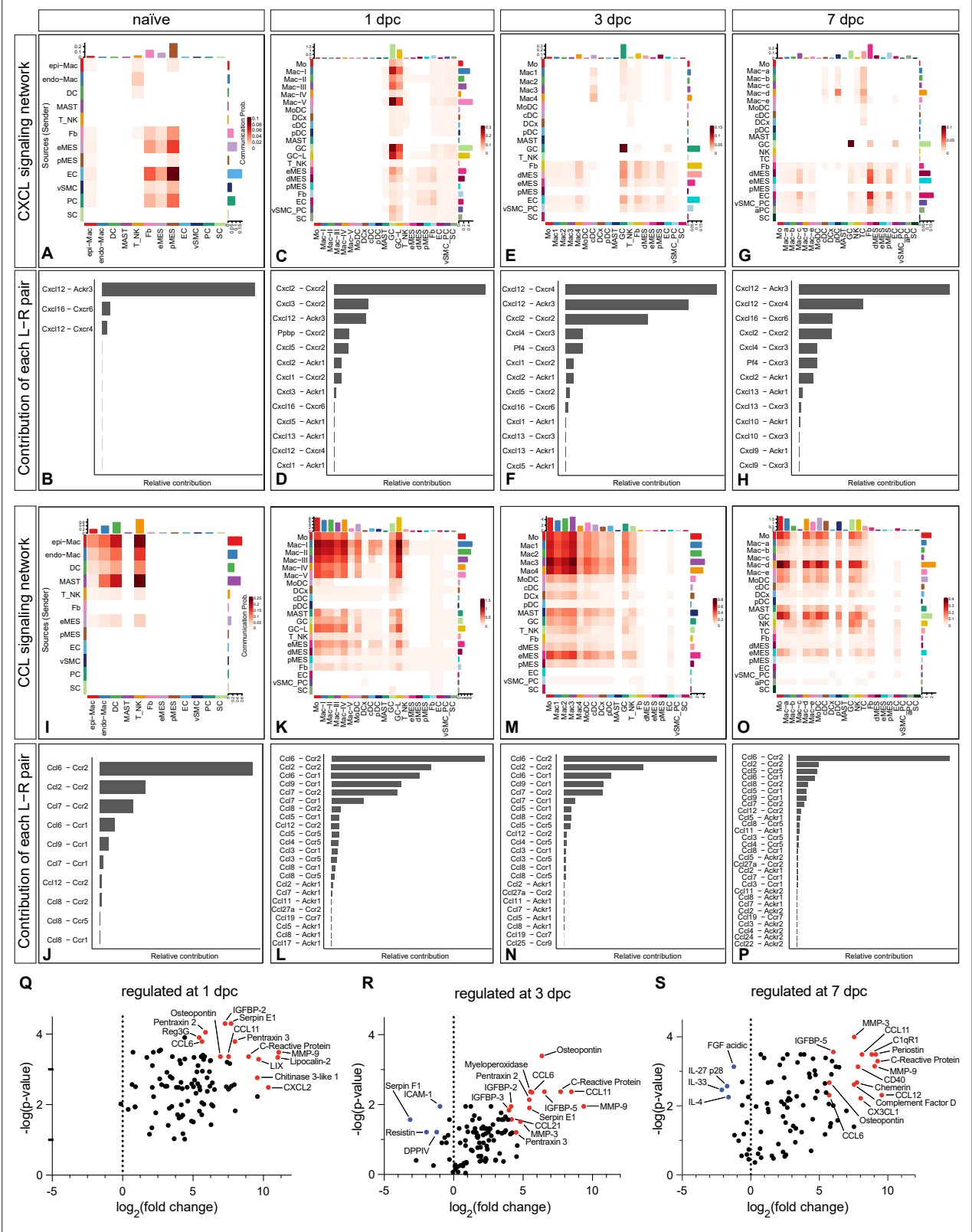

**Figure 4.** CellChat reveals chemotactic cell–cell communication networks in the injured peripheral nervous system (PNS). Hierarchical plots of CellChat analysis showing the inferred intercellular communication networks for (**A–H**) CXCL-chemokines and (**I–P**) CCL-cytokines in naïve nerve and during the first week following injury. The sender cells (ligand sources) are shown on the y-axis and receiving cells (receptor expression) on the x-axis. The probabilities for cells to communicate with each other are indicated. (**B, D ,F, H**) The bar graphs show the contributions of CXCL ligand–receptor pairs

*Figure 4 continued on next page*

*Figure 4 continued*

for each time point. (**J, L, N, P**) The bar graphs show the relative contributions of CCL ligand–receptor pairs for each time point. (**Q–S**) Volcano plot of extracellular proteins detected by ELISA compared to naïve nerve. The most abundant and strongly upregulated proteins in the 1dpc nerve (**Q**), the 3dpc nerve (**R**), and the 7dpc nerve (**S**) are labeled in red. The normalized signal on the x-axis shows the log2 fold-change and the y-axis shows the -log(p-value), normalized to naïve nerve.

The online version of this article includes the following source data and figure supplement(s) for figure 4:

**Figure supplement 1.** Identification of extracellular proteins in the injured sciatic nerve by ELISA.

**Figure supplement 1—source data 1.** X-Ray films of ELISA membranes probed with injured sciatic nerve lysates.

**Figure supplement 2.** Identification of extracellular proteins in the injured sciatic nerve.

**Figure supplement 3.** Identification of multiple dendritic cell (DC) populations in the injured peripheral nervous system (PNS).

**Figure supplement 4.** Identification of specific T cell (TC) and natural killer cell (NK) populations in the injured peripheral nervous system (PNS).

## The injured sciatic nerve harbors several dendritic cell populations

Only few DC are present in the naïve nerve, and they can be distinguished from resident Mac by their prominent expression of *Flt3* (FMS-like tyrosine kinase 3) and *Napsa* (**Figure 1L**). During the first week following nerve injury, MoDC, identified by *Cd209a* expression, gradually increase (**Figure 2H, P and X**). Gene expression analysis in iSNAT identifies additional DC subpopulations, including conventional dendritic cells (cDC, *Sept3/*septin 3 GTPase, *Clec9a/*C-type lectin domain containing 9A), plasmacytoid dendritic cells (pDC, *Ccr9* and *Siglech*), and DCx (*Ccr7/*chemokine receptor 7, *Ccl17*, *Ccl22*, *Fscn1/*fascin an actin-binding protein) (**Figure 4—figure supplement 3A**). DCx are reminiscent of Langerhans cells and express *Ccr7*, suggesting that they are mature cells destined for homing by exiting the injured nerve and migration to draining lymph nodes (**Bros et al., 2011**; **Liu et al., 2021**). At 7dpc, two subpopulations of conventional DC (cDC1 and cDC2) are detected. Previous work has established that cDC1 are specialized in cross-presentation and activation of cytotoxic TC, and cDC2 are specialized in driving helper TC responses (**Steinman, 2012**). The two populations can be distinguished based on preferential expression of marker genes, cDC1 (*Itgam⁻/*CD11b, *Itgae⁺/*integrin αE, *Clec9a⁺*, *Tlr3⁺/*toll-like receptor 3, *Xcr1⁺/*X-C motif chemokine receptor 1⁺) and cDC2 (*Itgam⁺*, *Clec9a⁻*, *Xcr1⁻*, *Irf4⁺/*interferon regulatory factor 4, *Slamf7/*CD2-like receptor-activating cytotoxic cells).

CellChat identifies an FLT3 signaling network in the injured nerve and predicts communication between TC, NK, and all four DC populations (data not shown). Nerve injury triggers strong expression of interferon-inducible genes in MoDC (*Ifi30, Ifitm1, Ifitm6*), cDC1 (*Ifi30, Ifi205, Irf8*), and cDC2 (*Ifi30*). MoDC express high levels of macrophage galactose-type lectin, encoded by the C-type lectin receptor *Clec10a*, typically found on tolerogenic antigen-presenting cells (**van Kooyk et al., 2015**). The *Clec9a* gene product is expressed by cDC and binds to filamentous actin exposed by damaged or ruptured cells (**Zhang et al., 2012**). The strong expression of *Clec9a* by cDC1 at 7dpc suggests a role in antigen uptake and presentation. While DC are superior to Mac in presenting antigen and express high levels of *MHC class II* components (*H2-Aa, H2-Ab1, H2-DMa, H2-Eb1, Cd74*), most Mac in the naïve nerve, and some Mac in the injured nerve, express *MHC class II* and *Cd74*, suggesting that they are endowed with antigen-presenting capabilities to naïve CD4⁺ TC.

To infer potential functions for DC in the injured nerve, we carried out pathway analysis (**Figure 4—figure supplement 3B–E**). At 7dpc, the top positively regulated pathways include *Th1 pathway* for MoDC and *Th2 pathway* for cDC1, suggestive of proinflammatory and inflammation resolving functions, respectively. The top pathways for pDC are *NK cell signaling* and *Fcγ receptor-mediated phagocytosis*. Top pathways for DCx include *neuroinflammation signaling* and *crosstalk between DC and NK cells*. Taken together, we identify different DC populations in the injured PNS and pathway analysis predicts extensive crosstalk between DC-NK and DC-TC.

## PNS injury triggers a gradual T cell response

For TC classification, we used the *Expression Analysis* function in iSNAT. Most TC express the T cell receptor (TCR) α-chain constant (*Trac*) and β-chains (*Trbc1* and *Trbc2*), suggesting that they are αβ TC (**Figure 4—figure supplement 4A–D**). While only a few TC are present in the naïve sciatic (**Figure 1B**), they gradually increase upon injury, and by 7dpc make up ~10% of all immune cells. Few *Mki67/*Ki67⁺ TC are observed in the injured nerve and the majority are *Ms4a4b^{hi}*, a negative regulator of cell proliferation (**Figure 2—figure supplement 2A–D**). This suggests that the observed increase in

TC is primarily due to nerve infiltration rather than local proliferation. Commensurate with this, TC in the injured nerve strongly express *Cxcr6*, a chemotactic receptor for *Cxcl16*. CellChat identifies Mac and DCx as sender cells expressing *Cxcl16*, suggesting that the increase in TC is due to Cxcr6-Cxcl16-mediated chemoattraction. Nearly all TC express CD3 (*Cd3g, Cd3d, Cd3e,* and *Cd3z/Cd247*), a key component of the TCR-CD3 complex (*Figure 4—figure supplement 4E–H*). At 7dpc, the TC population is comprised of CD4+ T helper cells (Th), expressing *Cd4,* (*Figure 4—figure supplement 4I–L*), and CD8 cells expressing *Cd8a* and *Cd8b1,* suggesting that they are CD8αβ+ cells. Differentiation of CD4+ Th into pro-inflammatory Th1 or anti-inflammatory Th2 effector cells is controlled by the transcription regulators T-bet (*Tbx21*) and GATA3 (*Gata3*), respectively (*Jenner et al., 2009*). *Gata3+* Th2 cells are observed in naïve and injured nerves, while *Tbx21+* are largely absent from naïve nerve but increase following injury and express the killer cell lectin-like receptors *Klrd1* and *Klrc1*, markers of an activated pro-inflammatory state. Pathway analysis of TC at 7dpc identified *Th2 pathway* and *Th1 pathway* as top hits (*Figure 4—figure supplement 4Q*). It is well established that the Th1 response plays a key role in neuropathic pain development and persistence (*Davies et al., 2020*; *Moalem and Tracey, 2006*). At 7dpc, there is a small population of γδ TC (*Trdc*), and some T regulatory cells (Tregs), a specialized subpopulation of TC that acts to suppress the activation and expansion of autoreactive T cells. In the PNS, Tregs (*Cd4, Il2ra*/CD25, *Foxp3*) are of interest because they may function in self-tolerance and pain mitigation (*Davoli-Ferreira et al., 2020*).

## Natural killer cells increase following nerve injury

In contrast to TC, NK do not express CD3 (*Cd3g, Cd3d, Cd3e*), but strongly express natural cytotoxicity receptor (*Ncr1*), killer cell lectin-like receptor subfamily B member 1C (*Klrb1c*/NK1.1), the pore-forming glycoprotein perforin (*Prf1*), and the granzyme family serine proteases (*Gzma, Gzmb*) (*Figure 4—figure supplement 4M–P*). Granzymes are delivered into target cells through the immunological synapse to cause cell death. While the full spectrum of cells targeted by NK has yet to be determined, NK cytotoxic factors have been shown to accelerate degeneration of damaged axons (*Davies et al., 2019*). Compared to NK, CD8+ TC express low levels of granulysin (*Gnly*), *Prf1, Gzma,* and *Gzma*, suggesting limited cytotoxic activity. However, similar to NK, many TC express *Nkg7*, a natural killer cell granule protein, and the killer cell lectin-like receptor K1 (*Klrk1*/NKG2D), indicative of cytotoxic abilities. NK (and some TC) strongly express *Klrc1*/NKG2A (CD94), an immune inhibitory checkpoint gene. The ligand for NKG2A is the nonclassical MHC1 molecule Qa-2 (encoded by *H2-q7*), expressed by NK and TC. CellChat predicts a high probability for an MHC-I signaling network between CD8+ TC, NK, and pDC, and an MHC-II signaling network between CD4+ TC and with DC and Mac in the injured nerve. Pathway analysis for NK at 7dpc identifies *NK signaling*, *Th2 pathway*, and *crosstalk between DC and NK* as top pathways (*Figure 4—figure supplement 4R*). The majority of NK and TC in the injured nerve produce interferon γ (*Ifng*), some TC produce TNFα (*Tnf*), and few produce GM-CSF (*Csf2*). In addition, NK and TC produce chemoattractants and survival signals for DC, including *Flt3l, Ccl5, Xcl1*/lymphotactin, suggesting that they directly regulate DC migration and function.

## Myeloid cells in the injured nerve undergo rapid metabolic reprogramming

Tissue repair is an energetically demanding process, suggesting that cells must efficiently compete for limited resources, at the same time, the repair process requires highly coordinated action among diverse cell types. Pathway analysis predicts that Mo/Mac in the 1-day injured nerve upregulate glycolysis to cover their bioenergetic needs (*Figure 5*). Nerve injury results in strong upregulation of the transcription factor *Hif1a*/hypoxia-induced factor 1α (Hif1α) in Mo, maturing Mac, and GC (*Figure 5—figure supplement 1A–D*). After a sharp increase at 1dpc, *Hif1a* levels remain elevated at 3dpc, before declining to pre-injury levels at 7dpc. Hif1α is a master regulator of cellular metabolism and the molecular machinery for glycolytic energy production (*Nagao et al., 2019*; *Pearce and Pearce, 2013*; *Schuster et al., 2021*). A metabolic shift, away from oxidative phosphorylation (OXPHOS) and toward aerobic glycolysis for the conversion of glucose into lactate, is known as Warburg effect (*Figure 5A*). In immune cells, the Warburg effect is of interest because it not only regulates metabolic pathways for energy production, but also gene expression to drive Mac toward a proinflammatory, glycolytic state (*McGettrick and O'Neill, 2020*; *Palsson-McDermott et al., 2015*).

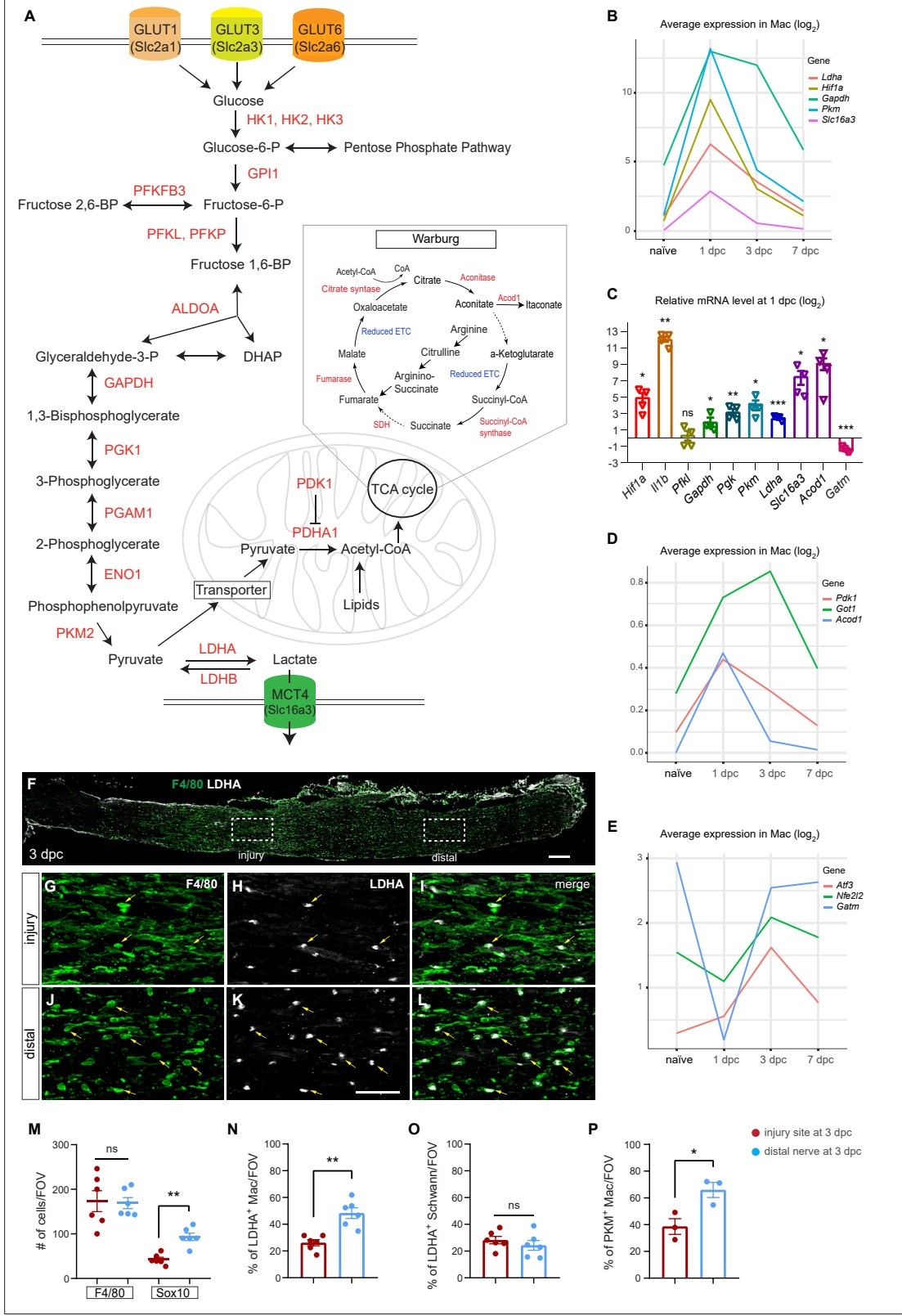

**Figure 5.** Application of injured sciatic nerve atlas (iSNAT) reveals metabolic reprogramming of immune cells. Metabolic pathways regulated by peripheral nervous system (PNS) injury. (**A**) Glycolysis: metabolites and enzymes for the catabolism of glucose into pyruvate and pyruvate to lactate. Glucose-6-phospate is required for nucleotide synthesis through the pentose phosphate pathway. Glycolysis and the pentose phosphate pathway occur in the cytosol. Pyruvate can be metabolized into Acetyl-CoA and enter the tricarboxylic acid (TCA) cycle. The TCA takes place in mitochondria. The

*Figure 5 continued on next page*

*Figure 5 continued*

Warburg effect allows for rapid ATP production through aerobic glycolysis and fermentation of pyruvate to lactate. The Warburg effect is characterized by limited mitochondrial ATP production because of TCA cycle fragmentation at the conversion of aconitate to α-ketoglutarate and succinate to fumerate, steps marked with dotted arrows. (**B**) Injury-regulated gene products associated with glycolysis, as inferred by scRNAseq data. Log2 average expression of genes for cells classified as Mac. (**C**) Quantification of gene expression by qRT-PCR in the 1dpc nerve relative to naïve nerve. Log2-fold changes relative to naïve nerve are shown. Per gene product, n = 4 biological replicates. p-values, *<0.05, **<0.001, ***<0.0001, Student's *t* test. ns, not significant. (**D, E**) PNS injury-regulated gene products associated with inhibition of mitochondrial energy synthesis (**D**) and inflammation resolution (**E**), as inferred by scRNAseq data. Log2 average expression of genes for cells classified as Mac. (**F**) Longitudinal section of 3dpc injured sciatic nerve stained for Mo/Mac (anti-F4/80, green) and LDHA (white), proximal is to the left. Dotted boxes mark regions at the injury site and the distal nerve, respectively. Scale bar, 100 µm. (**G–I**) Higher magnification view of the dotted box at the injury site. Some F4/80 and LDHA double-positive cells are labeled with arrows (yellow). (**J–L**) Higher magnification view of distal nerve, dotted box showin in F. box in. Some F4/80 and LDHA double-positive cells are labeled with arrows (yellow). Scale bar (**G–L**), 50 µm. (**M**) Quantification of Mo/Mac (F4/80$^+$) and SC (Sox10$^+$) per field of view (FOV) at the 3dpc injury site (red dots) and distal nerve (blue dots). (**N–P**) Quantification of double labeled cells per FOV at the injury site and distal nerve for (**N**) LDHA$^+$ Mac (**O**) LDHA$^+$ SC, and (**P**) PKM$^+$ Mac. Biological replicates n = 3 with two technical replicates each. Nonparametric independent Student's *t* tests. *<0.05, **<0.01, GraphPad Prism 9.

The online version of this article includes the following figure supplement(s) for figure 5:

**Figure supplement 1.** Longitudinal analysis of gene products that regulate glycolysis.

**Figure supplement 2.** Peripheral nervous system (PNS) injury induces glycolytic macrophages and Schwann cells.

Analysis of gene products implicated in glucose metabolism revealed that during the early injury response, GC and Mo/Mac express transporters for glucose (*Slc2a1/*GLUT1, *Slc2a3/*GLUT3) and hexose (*Slc2a6/*GLUT6) to import carbohydrates as a means of energy production. Moreover, there is rapid, injury-induced upregulation of most glycolytic enzymes, including hexokinases (*Hk1, Hk2, Hk3*), phospho-fructokinases (*Pfkp, Pfkl*), glyceraldehyde-3-phosphate dehydrogenase (*Gapdh*), phosphoglycerate kinase (*Pgk1*), and pyruvate kinase (*Pkm*) (***Figure 5B***, ***Figure 5—figure supplement 1E–T***). Injury-regulated expression of *Hif1a* and key glycolytic enzymes was validated by qRT-PCR (***Figure 5C***). The *Pkm* gene products, PKM1 and PKM2, convert phosphophenol pyruvate into pyruvate and are the rate-limiting enzymes of glycolysis (***Figure 5A***). PKM2 is of interest because of its nuclear role and interaction with Hif1α to promote expression of glycolytic enzymes and proinflammatory cytokines, including *Il1b* (***Palsson-McDermott et al., 2015***). Analysis of 1dpc nerves by qRT-PCR revealed a strong upregulation of *Il1b* when compared to naïve nerves (***Figure 5C***). Mac in injured nerves express elevated levels of *Ldha*, the enzyme that converts pyruvate to lactate (***Figure 5B and C***). Evidence for intracellular lactate build-up is the sharp increase in *Slc16a3*, encoding the monocarboxylate transporter 4 (MCT4) for shuttling lactate out of cells (***Figure 5B and C***). To validate the increase in glycolytic enzymes in injured nerve tissue, longitudinal sections of naïve, 1, 3, and 7dpc mice were stained with anti-PKM2, anti-LDHA, and F4/80 (***Figure 5—figure supplement 2A–H***). In naïve nerve, very few cells stain with anti-PKM and anti-LDHA; however, there is an increase in F4/80$^+$ Mac labeled with anti-PKM and anti-LDHA at 1, 3, and 7dpc (***Figure 5—figure supplement 2A–H***). Next, we assessed the abundance of glycolytic F4/80$^+$ Mac at the 3dpc injury site and the distal nerve (***Figure 5F***). The density of F4/80$^+$ cells per field of view (FOV) is comparable between injury site and distal nerve (***Figure 5M***). At 3dpc, PKM2$^+$ and LDHA$^+$ Mac are found at the nerve injury site as well as in the distal nerve (***Figure 5F–L***). Quantification of LDHA$^+$F4/80$^+$ double-labeled and PKM$^+$F4/80$^+$ double-labeled cells revealed higher density in the distal nerve than at the injury site (***Figure 5N and P***). The number of Sox10$^+$ SC is reduced at the nerve injury site compared to distal nerve (***Figure 5M***); however, the density of LDHA$^+$Sox10$^+$ SC is comparable between injury site and distal nerve (***Figure 5O*** and ***Figure 5—figure supplement 2I–N***).

Lactate is far more than a metabolic waste product since extracellular lactate has been shown to exert immunosuppressive functions, promote angiogenesis, axonal growth, and neuronal health (***Chen et al., 2018***; ***Fünfschilling et al., 2012***; ***Hayes et al., 2021***; ***Kes et al., 2020***). Moreover, lactate released by SC has axon protective effects and elevated lactate may be particularly important during the early injury response (***Babetto et al., 2020***). Evidence for inhibition of OXPHOS in the nerve during the early injury response is the upregulation of *Acod1* (***Figure 5C and D***), an enzyme that converts aconitate into itaconate, thereby disrupting the TCA cycle (***Figure 5A***). Moreover, itaconate functions as an inhibitor of succinate dehydrogenase (SDH), leading to further inhibition of the TCA cycle (***Lampropoulou et al., 2016***). Similarly, *Pdk1* (pyruvate dehydrogenase kinase 1) and

*Got1* (glutamic-oxaloacetic transaminase), inhibitors of OXPHOS, are high at 1 and 3dpc, but low at 7dpc (*Figure 5D*). Fragmentation of the TCA cycle at the level of citrate and succinate is a key feature of pro-inflammatory Mac and a hallmark of the Warburg effect (*Figure 5A*; *Eming et al., 2021*). During the inflammation resolution phase, Mac undergo metabolic reprogramming away from glycolysis toward OXPHOS, a switch that coincides with upregulation of the anti-inflammatory transcription factors *Atf3*/activating transcription factor 3 and *Nfe2l2*/Nrf2 (*Mills et al., 2018*). In the injured nerve, *Atf3* and *Nfe2l2* are low at 1dpc and upregulated at 3dpc (*Figure 5E*). The mitochondrial enzyme *Gatm* (glycine amidinotransferase) is important for the biosynthesis of creatine, a molecule that facilitates ATP production from ADP. *Gatm* is transiently downregulated in Mo/Mac at 1dpc and increases in Mac at 3dpc and 7dpc (*Figure 5E*).

Taken together, application of iSNAT provides multiple lines of evidence that during the early injury response Mo/Mac undergo rapid metabolic reprogramming to increase glycolytic flux and acquire a proinflammatory state. The proinflammatory state is short-lived as Mac rewire their metabolism toward OXPHOS, and this is paralleled by a transition toward a pro-resolving phenotype.

## Myeloid cells in peripheral blood are not programed for glycolytic energy production

The large number of hematogenous immune cells entering the injured nerve (*Kalinski et al., 2020*; *Ydens et al., 2020*) prompted us to carry out a deeper analysis of PBMC. One important goal was to determine which PBMC populations enter the nerve upon injury and to determine how this impacts gene expression and metabolic profiles. PBMC obtained from naïve mice were subjected to scRNAseq. The UMAP plot of 34,386 high-quality PBMC shows 20 Louvain determined clusters with a resolution cutoff of 0.5 (*Figure 6A*). We identified four clusters with B cells (BC1-BC4) expressing *Cd79a* (40.2% of all PBMC); plasma blasts (PB, 0.83%) expressing *Jchain*/joining chain of multimeric IgA and IgM, and platelets/megakaryocytes (P/M, 1.08%) expressing, *Pf4*/platelet factor 4, *Gp5*/glycoprotein V platelet, and *Plxna4*/PlexinA4. In addition, there are five clusters of *Cd3e*+ T cells (TC1-TC5, 26.41%), including *Cd8b1* (TC2) and *Cd4* (TC5) cells. NK cells (7.4%) strongly express *Nkg7*/natural killer cell granule protein 7. Cells in cluster 8 express markers for NK (*Nkg7*) and TC (*Cd3e*), suggesting that they represent NKT cells (*Figure 6B–H*).

Two clusters with granulocytes (GC) were identified, harboring mature (mGC: *Ly6g*$^{hi}$, *Ngp*$^{hi}$, *Lcn2*$^{hi}$, *Il1b*$^{neg}$) and immature (iGC: *Cxcr2*$^{hi}$, *Csf3r*$^{hi}$, *Ly6g*$^{low}$, *Il1b*$^{hi}$) cells and make up 9.6% of all PBMC (*Figure 6B, I and J*). In addition, there are two Mo subpopulations, Mo1 (5.95%) and Mo2 (2.61%), representing patrolling (*Cx3cr1*$^{hi}$, *Csf1r*$^{hi}$, *Ear2*$^{hi}$, *Cd36*$^{hi}$, *Ccr2*$^{neg}$, *Ly6c2*$^{neg}$, and *Nr4a1*$^{hi}$) and classical (*Ly6c2*$^{hi}$, *Chil3*$^{hi}$, *Ccr2*$^{hi}$, and *Ear2*$^{neg}$) Mo (*Figure 6B and K–O*). To underscore differences between Mo1 and Mo2, a heatmap with the top DEGs was prepared (*Figure 6—figure supplement 1*). There is a small cluster with Mac (Mac0, 1.15%) expressing *Prg4*/proteoglycan4$^{hi}$, *Ccl24*$^{hi}$, *Ltc4s*$^{hi}$, and *Fn1*$^{hi}$ (*Figure 6B and P*). DC (*Cd209a*, *Clec10a*) make up 1.77% and Mast cells (*Fcer1a*, *Cpa3*) 0.55% of all PBMC (*Figure 6B and Q*).

Next, we used computational methods to extract blood myeloid cell (iGC, mGC, Mo1, Mo2, Mac0, and DC) transcriptomes for comparison with myeloid cells in the injured sciatic nerve at 1dpc, 3dpc, and 7dpc. Of the two populations of hematogenous Mo (Mo1 and Mo2), classically activated Mo2 (*Ly6c2*$^{hi}$, *Ccr2*$^{hi}$, *Chil3*$^{hi}$) are transcriptionally more similar to Mo in the 1dpc nerve, indicating they are the primary source of Mo entering the nerve upon injury. GC in the 1dpc nerve are *Csf3r*$^{hi}$ and *Ngp*$^{low}$, and more closely resemble iGC than mGC in blood, suggesting preferential entry of iGC (data not shown).

Because Mo/Mac in the 1dpc nerve are metabolic programmed for glycolytic energy production (*Figure 5*), we wondered whether their precursors in blood, Mo2, exhibit a similar metabolic profile. Strikingly, Mo2 show low levels *Hif1a*, *Pfkl*, *Pgk1*, *Ldha*, and *Slc16a3* (*Figure 6R–V*). This suggests that upon nerve entry, classical Mo become activated and undergo rapid metabolic reprogramming toward glycolytic energy production. Some immature GC in blood, identified by their high *Cxcr2* expression (*Figure 6J*), appear to be glycolytic and may produce lactate, based on elevated levels of *Hif1a*, *Pfkl*, *Pgk1*, *Ldha*, and *Slc16a3* (*Figure 6R–V*). GC further increase expression of these gene products upon nerve entry (*Figure 6—figure supplement 2A–J*).

*Arg1,* an important regulator of innate and adaptive immune responses, is not expressed in peripheral blood Mo; however, upon nerve entry, maturing Mo strongly upregulate *Arg1* expression

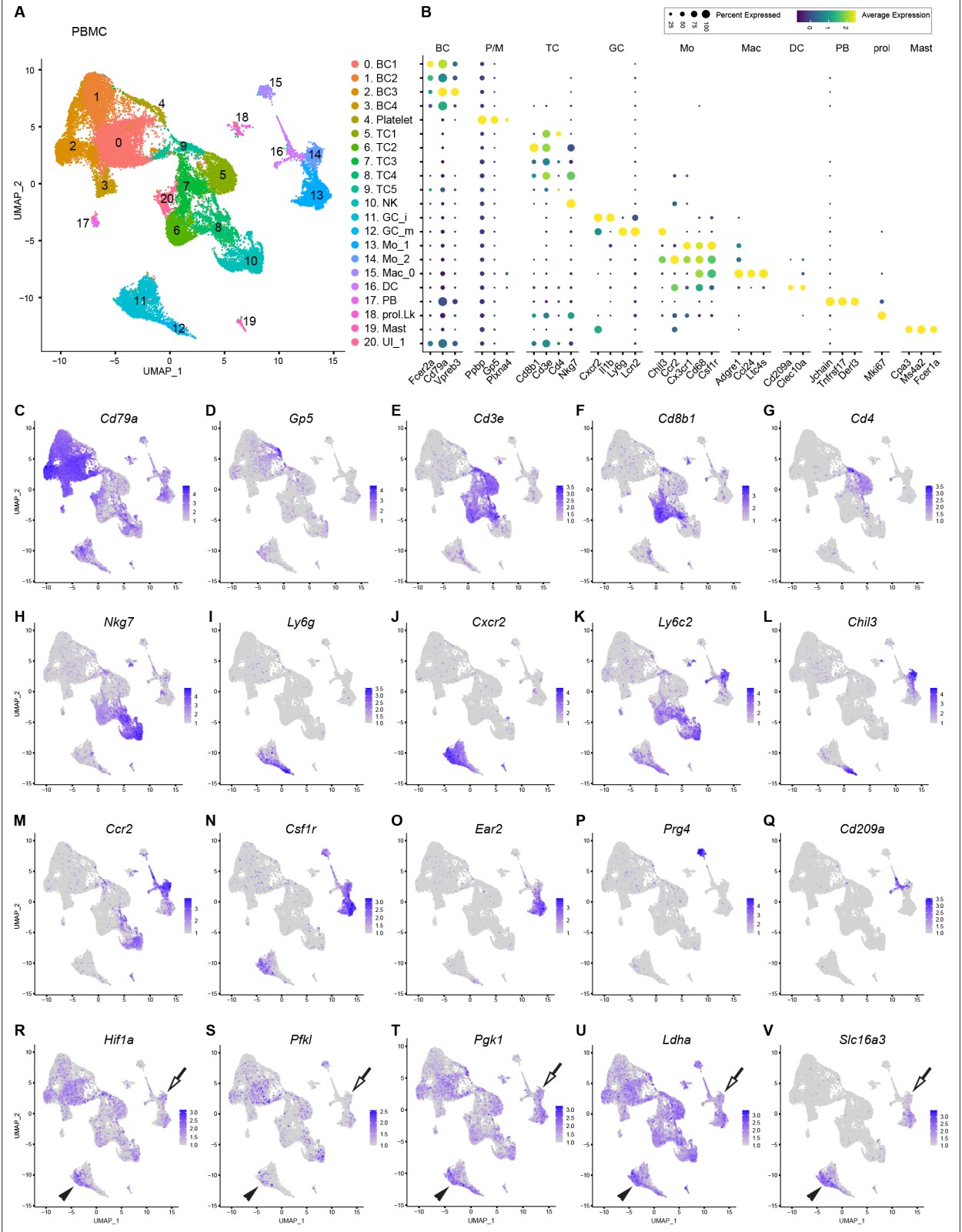

**Figure 6.** Catalog of peripheral blood mononuclear cells (PBMCs). (**A**) Blood was collected by cardiac puncture of naïve mice. UMAP plot embedding of PBMC revealed 20 cell clusters, including B cells (BC1-BC4), Platelets/megakaryocytes (P/M), T cells (TC1-TC5), natural killer cells (NK), immature and mature granulocytes (iGC and mGC), monocytes (Mo_1 and Mo_2), macrophages (Mac_0), dendritic cells (DC), plasma blasts (PB), proliferating leukocytes (prol.Lk), mast cells (MAST), and one cluster with unidentified cells (UI1). A total of 34,386 high-quality PBMC were analyzed (n = 2 replicates).

*Figure 6 continued on next page*

Figure 6 continued

(**B**) Dotplot with marker genes enriched in PBMC subpopulations. Color-coded expression levels are shown. The dot size reflects the percentage of cells that express the gene. (**C–Q**) Feature plots of marker gene expression in PBMC used for cell-type identification. *Cd79a*, CD79A antigen (immunoglobulin-associated alpha); *Gp5*, glycoprotein V platelet; *Cd3e*, T cell receptor complex CD3 epsilon subunit; *Cd8b1*, CD8 antigen beta chain 1; *Cd4*, CD4 antigen; *Nkg7*, natural killer cell group 7; *Ly6g*, lymphocyte antigen 6 complex locus G; *Cxcr2*, chemokine (C-X-C motif) receptor 2; *Ly6c2*, lymphocyte antigen 6 complex locus 2, *Chil3*, chitinase-like 3 (Ym1); *Ccr2*, chemokine (C-C motif) receptor 2; *Csf1r*, colony stimulating factor 1 receptor; *Ear2*, eosinophil-associated ribonuclease A family member 2; *Prg4*, proteoglycan 4; *Cd209a*, DC-SIGN (C-type lectin). (**R–V**) Assessment of PBMC metabolic state. In feature plots classical monocytes (Mo) are labeled with an arrow and immature granulocytes (iGC) with a black arrowhead. Feature plots for *Hif1a* (hypoxia inducible factor 1 alpha), *Pfkl* (phosphofructokinase liver type), *Pgk1* (phosphoglycerate kinase 1), *Ldha* (lactate dehydrogenase alpha), and *Slc16a3* (lactate exporter) are shown. Expression levels are projected onto the UMAP with a minimum expression cutoff of 1.

The online version of this article includes the following figure supplement(s) for figure 6:

**Figure supplement 1.** Differential gene expression between patrolling and classical Mo in blood.

**Figure supplement 2.** Activation of circulating immune cells upon nerve entry.

(*Figure 6—figure supplement 2K–O*). Upon nerve entry, Mo/Mac do not express the inflammation enhancing gene product nitric oxide synthase 2 (*Nos2*) (*Figure 6—figure supplement 2P–T*). At 1dpc, gene expression analysis of $Arg1^{hi}$ Mac identified high glycolytic activity, indicative of a proinflammatory character. Commensurate with this, $Arg1^{hi}$ cells strongly express *Cxcl3, Ccl2, Ccl6, Ccl7, Ccl12*, chemotactic molecules that augment nerve inflammation. At 3dpc, $Arg1^{hi}$ Mac exhibit a more resolving phenotype, with reduced levels of *Hif1a* and proinflammatory chemokines. At 7dpc, $Arg1^+$ are no longer found in the injured nerve (*Figure 6—figure supplement 2*). Taken together, our longitudinal studies show that upon nerve entry, circulating Mo undergo extensive transcriptional changes and metabolic reprogramming. Most notable is the sharp increase in glycolytic activity in classical Mo (Mo2) during the early injury response.

## Tracking myeloid cell transcriptional states and maturation in the injured nerve

Mo/Mac are often described as highly 'plastic' cells, educated by the local tissue environment. While Mac subpopulations in the injured PNS have been cataloged (*Kalinski et al., 2020*; *Ydens et al., 2020*), tracking them over time, as they mature, has not yet been attempted. To better understand Mo/Mac maturation in the injured nerve, we took advantage of our longitudinal scRNAseq datasets. Computational methods were used to extract myeloid cells, TC and NK from PBMC (*Figure 6*), naïve nerve (*Figure 1*), and three post-injury time points (*Figure 2*) for an integrated data analysis (*Figure 7A–E*). Comparing the UMAP plots of naïve sciatic nerve and PBMC shows little overlap, indicative of largely distinct immune compartments (*Figure 7A and B*). When comparing myeloid cells in the naïve and 1dpc nerve, a massive increase in GC is observed. The increase is transient in nature as GC are largely absent from 7dpc nerve (*Figure 7—figure supplement 1F–J*). GC in the 1dpc nerve appear more similar to $Il1b^+$iGC than $Il1b^-$ mGC in blood (*Figure 7—figure supplement 1K–O*). Mo in the 1dpc nerve are transcriptionally more similar to Mo2 than Mo1 in blood. This suggests that following SNC, iGC and classically activated Mo (Mo2) preferentially enter the injured nerve. The integrated data analysis clearly shows the rapid increase of $Chil3^+$ in the 1dpc nerve, followed by a decline over the next six days (*Figure 7F–J*). In a similar vein, *Ccl2* and *Ccl7* show a rapid but transient increase during the first week (*Figure 7—figure supplement 1*). Conversely, few $Gpnmb^+$ Mac are detected in blood or naïve nerve; however, $Gpnmb^+$ Mac gradually increase following SNC. A small number of $Gpnmb^+$ Mac is detected at 1dpc, more at 3dpc, and still more at 7dpc, suggesting that *Gpnmb* labels mature Mac (*Figure 7K–O*).

To predict trajectories of Mo maturation into their descendants, Mac and MoDC, we carried out SlingShot, pseudotime analysis of integrated myeloid cells. Using Mo as starter cells revealed a bifurcated differentiation trajectory and indicates that Mo give rise to Mac and MoDC in the injured nerve (*Figure 7*, *Figure 7—figure supplements 1–4*). The high degree of Mo/Mac plasticity is evident when cell cluster identified at 1dpc (Mac-I to Mac-V) (*Figure 2B*), 3dpc (Mac1-Mac4) (*Figure 2J*), and 7dpc (Mac-a to Mac-d) (*Figure 2R*) are visualized in the integrated dataset (*Figure 7—figure supplement 1*). To facilitate access to integrated immune cell datasets, we added these data to iSNAT.

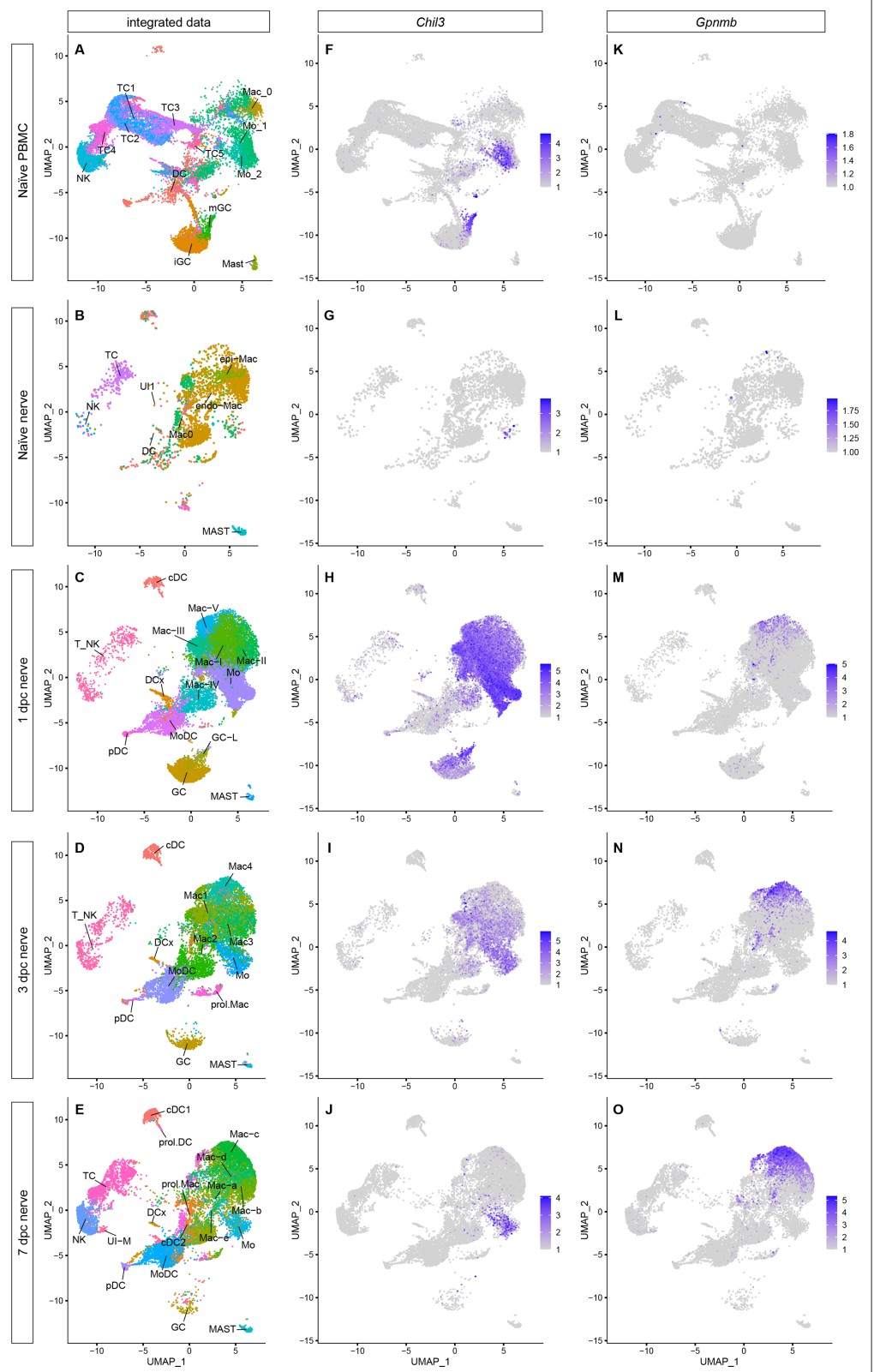

**Figure 7.** Integrated analysis of single-cell transcriptomes generated from peripheral blood mononuclear cells (PBMC), naïve nerve, and injured nerve. (**A–E**) UMAP plots of integrated myeloid cells split into (**A**) PBMC, (**B**) naïve sciatic nerve trunk, (**C**) 1dpc nerve, (**D**) 3dpc nerve, and (**E**) 7dpc nerve. (**F–J**) Integrated analysis of *Chil3*⁺ in (**F**) PBMC, (**G**) naïve nerve, (**H**) 1dpc nerve, (**I**) 3dpc nerve, and (**J**) 7dpc nerve. (**K–D**) Integrated analysis

*Figure 7 continued on next page*

*Figure 7 continued*

of *Gpnmb⁺* Mac in (**K**) PBMC, (**L**) naïve nerve, (**M**) 1dpc nerve, (**N**) 3dpc nerve, and (**O**) 7dpc nerve. For feature plots (**F–O**), expression values are projected onto the integrated UMAP with a minimum expression cutoff of 1. Abbreviations: iGC, immature granulocytes; mGC, mature granulocytes; Mo, monocytes; Mac, macrophages: DC, dendritic cells (MoDC, monocyte-derived DC; cDC, conventional DC; prol.DC, proliferating DC; pDC, plasmocytoid DC; and DCx, homing DC); MAST, mast cells; T_NK, T cells and natural killer cells; TC, T cells; NK, natural killer cells; UI, unidentified cells.

The online version of this article includes the following figure supplement(s) for figure 7:

**Figure supplement 1.** Transient upregulation of proinflammatory gene products in the injured sciatic nerve.

**Figure supplement 2.** Tracking of myeloid cells before and after entering the injured nerve.

**Figure supplement 3.** Heatmap showing the top 30 genes by importance in predicting pseudotime (*Figure 7— figure supplement 2E*) as determined by random forest analysis.

**Figure supplement 4.** Heatmap showing the top 30 genes by importance in predicting pseudotime (*Figure 7— figure supplement 2F*) as determined by random forest analysis.

## Nerve trauma generates spatial differences in immune cell composition

A crush injury divides a nerve into three distinct compartments: the proximal nerve, the injury site, and the distal nerve (*Figure 8A*). While specialized immune cells have been identified at the site of nerve injury (*Cattin et al., 2015*; *Kalinski et al., 2020*; *Shin et al., 2018*), a comparative analysis of cells at the injury site versus the distal nerve has not yet been carried out. Here, we harvested sciatic nerves at 3dpc and microdissected ~3 mm segments that either harbor the injury site or distal nerve. Innate immune cells were then captured with anti-CD11b and further analyzed by scRNAseq (*Figure 8A*). UMAP embedding of myeloid cells at the injury site versus distal nerve revealed location-specific distribution (*Figure 8B*). GC (cluster 10) are more abundant at the nerve injury site than in the distal nerve, and conversely, Mo (cluster 0) are more abundant in the distal nerve (*Figure 8C*). Most notably is the location-specific enrichment of select Mac subpopulations (*Figure 8C*). For example, *Arg1* expressing Mac4 (cluster 2) and Mac1 (cluster 3) are enriched at the injury site (*Figure 8D and E*), while *Cd38⁺* 3 (cluster 1) are more abundant in the distal nerve (*Figure 8F and G*). The number of proliferating Mac is comparable between the injury site and distal nerve (*Figure 8C*). In a similar vein, the distribution of MoDC, DCx, cDC, and T/NK is comparable between the injury site and the distal nerve (*Figure 8C*). NK are more abundant in the distal nerve (*Figure 8C*).

For an unbiased cell cluster identification at the injury site and in the distal nerve, we compared the scRNAseq datasets generated from the 3dpc injury site and distal nerve to the 'whole nerve' 3dpc reference scRNAseq data (*Figure 2J*). We projected the whole nerve principal component analysis (PCA) structure onto 'query' scRNA-seq data generated from the injury site or the distal nerve, implemented through Seurat v4 (*Stuart et al., 2019*). Our 'TransferData' pipeline finds anchor cells between the 3dpc whole nerve reference data and the query dataset, then uses a weighted vote classifier based on the known reference cell labels to yield a quantitative score for each cell's predicted label in the query dataset. A prediction score of 1 means maximal confidence, all votes, for the predicted label, and a score of 0 means no votes for that label (*Figure 8H and I*). Most notable is the strong enrichment of Mac4 at the injury site (*Figure 8H*) compared to distal nerve (*Figure 8I*). Similarly, distribution of the Mac1 population is skewed toward the injury site, however, to a lesser extent than Mac4. Conversely, Mac3 cells are enriched in the distal nerve (*Figure 8H and I*).

To confirm the location-specific distribution of different Mac subpopulations at 3dpc, we analyzed the injury site and distal nerve for Mac subpopulation-enriched transcripts using qRT-PCR. Because *Gpnmb*, *Syngr1*/synaptogyrin 1, *Fabp5/*fatty binding protein 5, and *Spp1* are highly enriched in Mac4, we assessed their expression by qRT-PCR and found significantly higher expression at the injury site (*Figure 8J*). Conversely, scRNAseq reveled preferential expression of *Cd38*/ADP-ribosyl cyclase 1 in Mac3 in the distal nerve and this was independently confirmed by qRT-PCR (*Figure 8K*). Expression of the Mo marker *Chil3* is not significantly different between the injury site and distal nerve. Compared to naïve nerve, *Gatm* (creatine biosynthesis) is reduced in the nerve at 3dpc, both at the injury site and the distal nerve (*Figure 8K*).

To assess the spatial distribution of Mac4 in the 3dpc nerve, we used in situ hybridization with an RNAscope probe specific for *Gpnmb*. Very few *Gpnmb⁺* are detected in naïve sciatic nerve (*Figure 8L*)

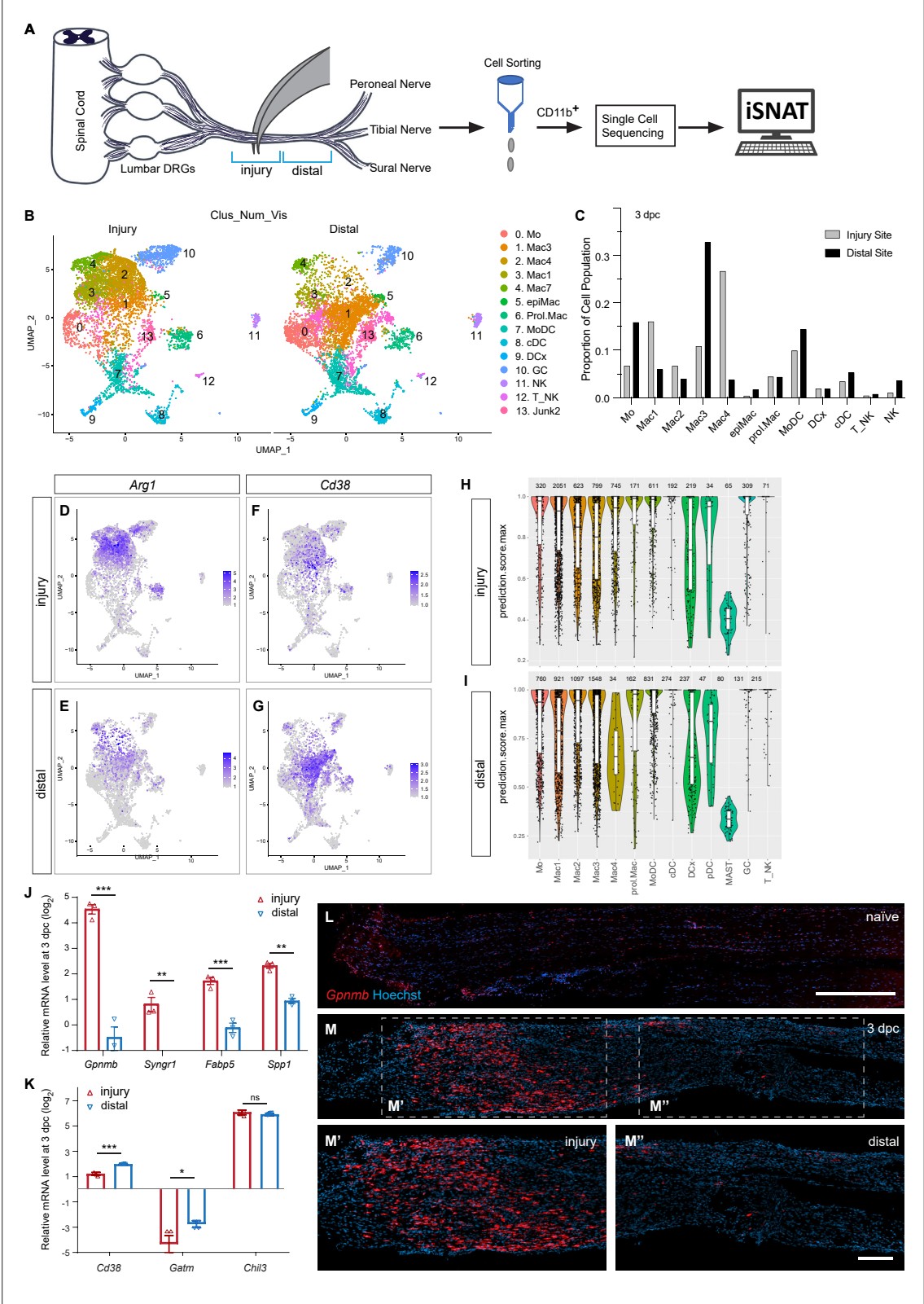

**Figure 8.** Spatial differences in the immune landscape of the injured sciatic nerve. (**A**) Schematic of workflow for analysis of wound tissue at the site of nerve injury and distal nerve tissue. Cartoon of a mouse lumbar spinal cord, dorsal root ganglia (DRGs), and major branches of the sciatic nerve. Nerve segments ~3 mm in length (injury or distal), marked with blue brackets, were harvested separately. Innate immune cells were captured with anti-CD11b magnetic beads and analyzed by scRNAseq. (**B**) UMAP plot of sciatic nerve myeloid cells captured at the injury site (left) and the distal

*Figure 8 continued on next page*

Figure 8 continued

nerve (right) at 3dpc. A total of 17,404 high-quality cells were subjected to unsupervised Seurat-based clustering, resulting in 13 cell clusters. (**C**) Bar graph of population size at the injury site versus distal nerve for 3d injured nerve immune cells. (**D, E**) Feature plots for *Arg1* at 3dpc showing injury site and distal nerve (**F, G**). Feature plots for *Cd38* at 3dpc showing injury site and distal nerve. Expression levels are color coded and calibrated to average gene expression. (**H, I**) Projection of the 3dpc 'whole nerve' reference data onto cells at the 3dpc injury site (**H**) and 3dpc distal nerve (**I**) onto 3dpc 'whole nerve' reference data. The y-axis shows the prediction score for each cell's top predicted cell population. The number of cells assigned to each population is shown on top. (**J, K**) Quantification of gene expression by qRT-PCR in the 3d injured nerve injury site versus distal nerve (n = 3). p-values, *<0.05, **<0.001, ***<0.0001, Student's *t* test. ns, not significant. (**L**) Longitudinal sections of naïve sciatic nerve stained for *Gpnmb* expression by RNAscope. (**M**) Longitudinal sections of 3d injured nerve stained for *Gpnmb* expression by RNAscope, proximal is to the left. Sections were counterstained with Hoechst. High power images of injury site (**M'**) and distal nerve (**M''**) are shown. Scale bar: 200 µm (**L, M''**).

The online version of this article includes the following figure supplement(s) for figure 8:

**Figure supplement 1.** Location-specific distribution of Mac subpopulations in the injured sciatic nerve.

and at 3dpc, *Gpnmb*+ are enriched in the endoneurium at the site of nerve injury site. Far fewer *Gpnmb*+ are detected in the distal nerve (*Figure 8M–M''*). When 3dpc nerves from *Arg1-YFP* reporter mice were stained for *Gpnmb* mRNA, largely overlapping labeling patterns were detected (*Figure 8—figure supplement 1A–G*).

MoDC show a more uniform distribution in the injured nerve, cells are present at the injured site and distal nerves, as assessed by RNAscope for *Cd209a* (*Figure 8—figure supplement 1H–K*). Together, these studies reveal spatial differences in Mac subpopulation distribution within the injured nerve.

## Nerve trauma causes a strong inflammatory response independently of Wallerian degeneration

To separate the immune response to mechanical nerve wounding from the immune response to WD, we employed *Sarm1-/-* mice (*Osterloh et al., 2012*). WT and *Sarm1-/-* mice were subjected to SNC and nerves harvested at 1, 3, and 7dpc (*Figure 9A–O*). Longitudinal sections were prepared and assessed for the presence of degenerating myelin and abundance of Mac in the proximal nerve, the injury site, and distal nerve. At 7dpc, fluoromyelin staining reveals intact myelin profiles proximal to the crush site, both in WT and *Sarm1-/-* nerves (*Figure 9J and K*), myelin ovoids emanating from disintegrated axons are observed in an ~3 mm nerve segment centered around the injury site in WT and *Sarm1-/-* mice (*Figure 9L and M*). In the distal nerve, however, myelin ovoids are observed only in WT (*Figure 9N*), but not *Sarm1-/-* mice (*Figure 9O*), confirming previous reports of delayed axon degeneration (*Osterloh et al., 2012*). In naive WT and *Sarm1-/-* mice, few F4/80+ Mac are detected (*Figure 9B and C*). Following nerve crush injury, WT and *Sarm1-/-* mice show a rapid increase in F4/80+ Mac at the nerve injury site at 1dpc (*Figure 9D and E*), 3dpc (*Figure 9F and G*), and 7dpc (*Figure 9H and I*). In the distal nerve, however, *Sarm1-/-* mice show far fewer F4/80+ cells following injury (*Figure 9F–I*). For an independent assessment of WD-elicited nerve inflammation, nerve trunks were isolated from naïve and injured WT and *Sarm1-/-* mice, divided into proximal nerve, the injury site, or distal nerve, and analyzed by Western blotting (*Figure 9P*). Independently of *Sarm1* genotype, the injury sites show elevated levels of CD11b/integrin alpha-M compared to proximal nerve. In the distal nerve, CD11b was more abundant in WT than in *Sarm1-/-* mice.

To quantify different immune cell profiles in WT and *Sarm1-/-* mice, we used flow cytometry (*Figure 9Q–W*, gating strategy is illustrated in *Figure 9—figure supplement 1*). In sham-operated mice, no differences in Ly6C-high (Ly6C$^{hi}$), Ly6C-intermediate (Ly6C$^{int}$), or Ly6C-negative (Ly6C$^-$) Mo/Mac are observed (*Figure 9Q and T*). For quantification of immune cell profiles that respond to traumatic nerve wounding versus WD, we separately harvested the site of nerve injury and distal nerve for analysis by flow cytometry. At the 3dpc injury site, the total number of Mo/Mac is comparable between WT and *Sarm1-/-* mice (*Figure 9R, U and W*). However, within the distal nerve, significantly more Mo/Mac cells are present in WT mice than in *Sarm1-/-* mice (*Figure 9S, V, and W*). Taken together, these studies show that nerve trauma causes a highly inflamed wound microenvironment, independently of WD, and a distinct inflammatory response in the distal nerve, that is WD dependent.

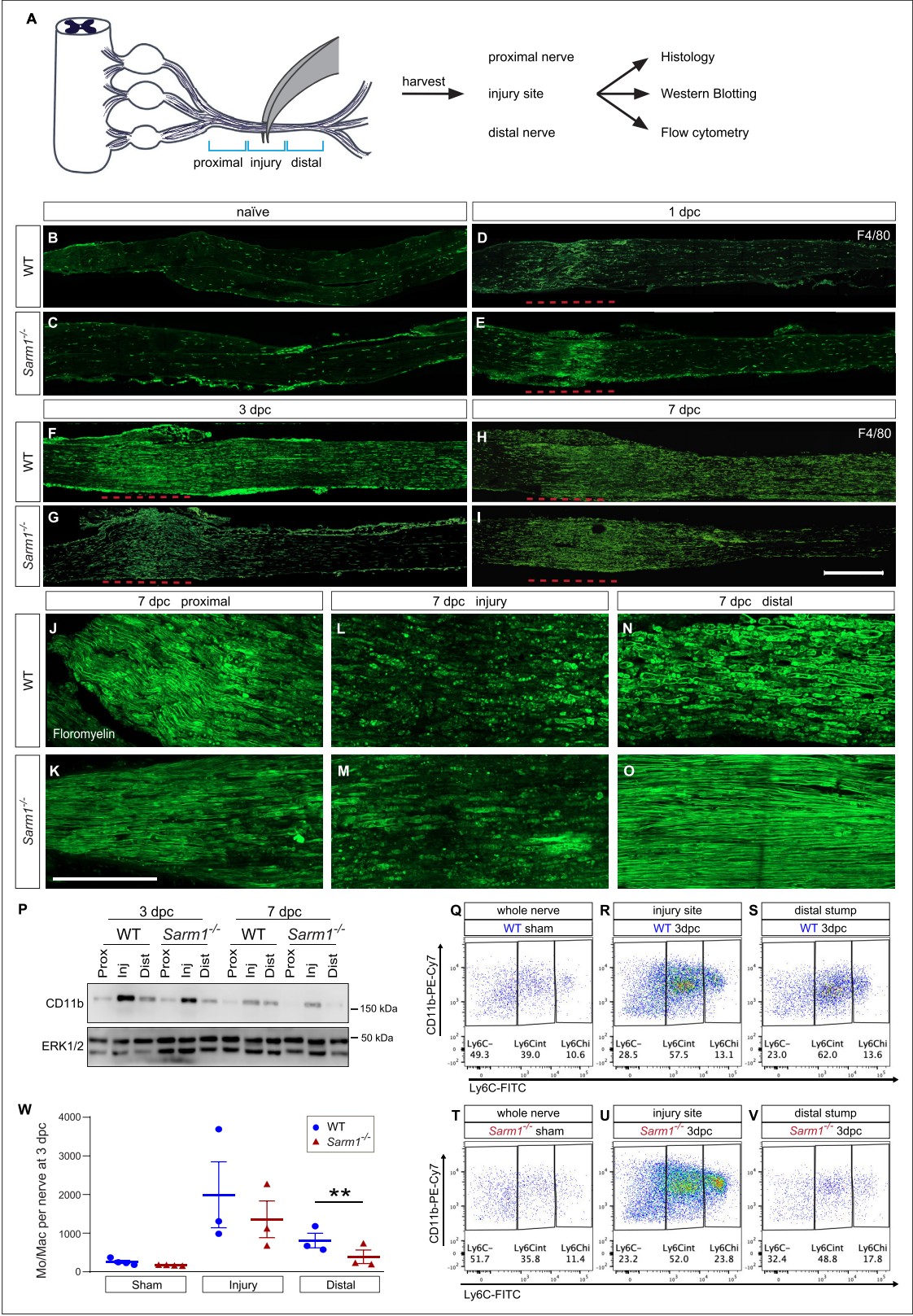

**Figure 9.** Nerve trauma causes Wallerian degeneration (WD)-independent nerve inflammation. (**A**) Cartoon of a mouse lumbar spinal cord, dorsal root ganglia (DRGs), and major branches of the sciatic nerve. A nerve injury divides the nerve trunk into a proximal segment, the injury site, and distal segment, each marked with blue brackets. Nerve segments were harvested and subjected to analysis. (**B–I**) Longitudinal sciatic nerve sections from WT and *Sarm1*−/− mice, stained with anti-F4/80 for identification of Mac. Representative examples of (**B, C**) naïve nerve, (**D, E**), 1dpc (**F, G**), 3dpc, and at

*Figure 9 continued on next page*

*Figure 9 continued*

(**H, I**) 7dpc. Injury site is marked with a dashed red line, proximal is to the left. Scale bar, 500 µm. (**J–O**) Longitudinal sciatic nerve sections from WT and *Sarm1*-/- mice at 7dpc, stained with fluoromyelin. Representative images of proximal nerve, the injury site and distal nerve are shown. Scale bar, 200 µm. (**P**) Western blots of sciatic nerve segments collected at 3dpc and 7dpc from WT and *Sarm1*-/- mice. Nerves were divided into proximal, injury site, and distal segments and blots probed with anti-CD11b, and anti-ERK1/2. (**Q–V**) Flow cytometry dotplots for Mo/Mac of sham-operated WT and *Sarm1*-/- sciatic nerve trunks, the 3dpc nerve injury site and distal nerve. (**W**) Quantification of Mo/Mac (Ly6C$^{hi}$ + Ly6C$^{int}$ + Ly6C$^-$) in sham-operated mice, the 3dpc injury site, and 3dpc distal nerve of WT and *Sarm1*-/- mice. N = 3, with 3–5 mice per genotype per replica. Flow data are represented as mean ± SEM. Statistical analysis was performed in GraphPad Prism (v9) using two-way, paired *t*-test. **$p < 0.01$.

The online version of this article includes the following source data and figure supplement(s) for figure 9:

**Source data 1.** Western blots (LiCOR) of sciatic nerve segments probed with anti-CD11b and anti-Erk1/2.

**Figure supplement 1.** Gating strategy for flow cytometry.

## In *Sarm1-/-* mice, monocytes enter the distal nerve stump but fail to mature into macrophages

Because we used *Sarm1* global knock-out mice for our studies and *Sarm1* has been shown to function in Mac (*Gürtler et al., 2014*), a potential confounding effect is *Sarm1* deficiency in Mac. To assess nerve entry of circulating WT immune cells, in a *Sarm1-/-* background, we employed parabiosis (*Figure 10A*). *Sarm1-/-* host mice were surgically fused to tdTomato (tdT) donor mice and allowed to recover for 3 weeks. For comparison, WT/tdT parabiosis complexes were generated and processed in parallel. Flow cytometry was used to analyze blood samples of host parabionts for tdT$^+$ leukocytes and revealed ~30% chimerism (*Figure 10B and C*). In each complex, both parabionts were subjected to bilateral nerve crush. At 7dpc, analysis of longitudinal nerve sections of the *Sarm1-/-* parabiont revealed that many tdT$^+$ leukocytes entered the site of nerve injury, comparable to injured WT parabionts (*Figure 10D–F*). In the distal nerve of the *Sarm1-/-* parabiont, at 1 mm, 2 mm, and 3 mm from to the injury site, some tdT$^+$ leukocytes are present, however, at significantly reduced numbers when compared to WT parabionts (*Figure 10F*). Interestingly, only a few tdT$^+$ cells in the *Sarm1-/-* distal nerve stained for F4/80, a marker for Mac (*Figure 10G and H*), or Ly6G, a marker for neutrophils (*Figure 10—figure supplement 1*).

To further investigate blood-borne immune cells that enter the 7dpc distal nerve of *Sarm1-/-* mice, we separately harvested and analyzed the 7dpc injury site and distal nerve segments from WT and *Sarm1-/-* single mice using flow cytometry. The abundance of Mo and Mac, identified as Ly6C$^{hi}$, Ly6C$^{int}$, and Ly6C$^-$ cells, at the nerve injury site is comparable between WT and *Sarm1-/-* mice (*Figure 10I and J*). In the distal nerve however, Ly6C$^{hi}$ cells in *Sarm1-/-* nerves are significantly elevated compared to Ly6C$^{int}$ and Ly6C$^-$ populations (*Figure 10I and J*). This stands in contrast to WT distal nerves, where Ly6C$^-$ and Ly6C$^{int}$ cells outnumber the Ly6C$^{hi}$ population (*Figure 10K*). Because WT and *Sarm1-/-* mice show a similar baseline of Mac in the naïve nerve (*Figure 9Q and T*), this shows that WD is not required for the immune response to mechanical nerve wounding but is required for a full-blown immune response in the distal nerve stump. Furthermore, out data show that in *Sarm1-/-* mice Ly6C$^{hi}$ Mo enter the distal nerve prior to WD but fail to differentiate into Mac. This suggests that chemoattractive signals for Mo are released from severed fibers prior to WD and that WD degeneration is required for Mo maturation.

## Discussion

Upon injury, the adult murine PNS exhibits a remarkable degree of spontaneous regeneration of motor and sensory axons and near complete functional recovery. To better understand the cellular and molecular events associated with PNS degeneration and regeneration, we carried out a longitudinal scRNAseq study. Analysis of the immune response to nerve crush injury, during the first week, revealed a highly dynamic microenvironment. The early immune response is pro-inflammatory and dominated by GC and Mo/Mac, metabolically programmed for glycolytic energy production. The elevated expression of nearly all glycolytic enzymes, lactate dehydrogenase, and the lactate export channel MCT4 indicates that a Warburg-like effect is at play, coupling glycolytic energy production with a proinflammatory Mo/Mac phenotype. This stands in marked contrast to the low glycolytic activity of circulating Mo in blood and suggests that upon nerve entry Mo become activated and

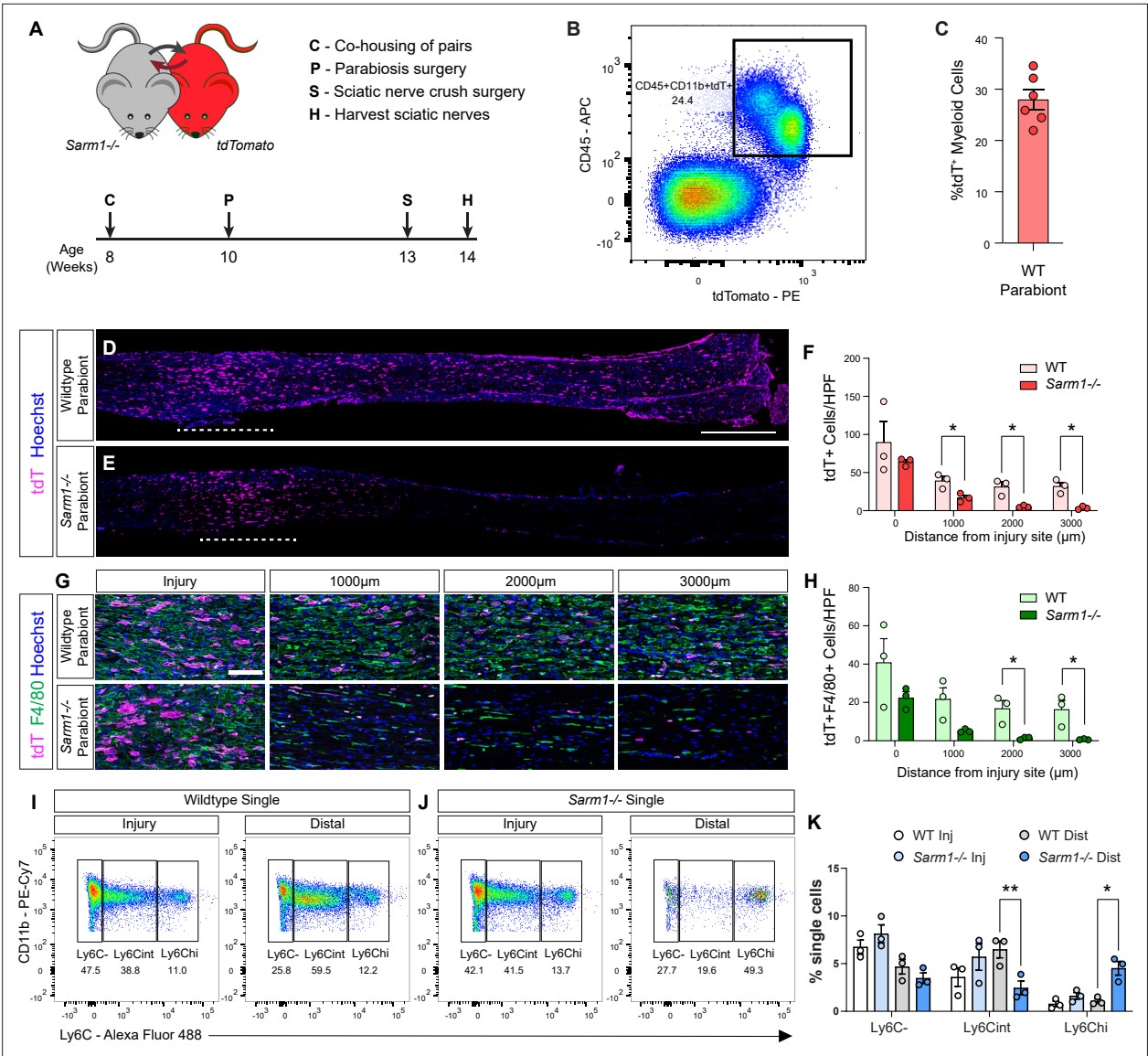

**Figure 10.** Evidence for Wallerian degeneration (WD)-dependent and WD-independent nerve inflammation. (**A**) Timeline for parabiosis experiments. After a 2-week co-housing period, 10-week-old WT or *Sarm1-/-* and *tdTomato* mice were surgically paired. (**B**) To assess chimerism, blood was harvested and analyzed by flow cytometry. Dotplot of tdT+ myeloid cells (CD45+CD11b+) is shown. (**C**) Quantification of tdT+ myeloid cells in host parabionts (n = 6), revealed chimerism of 28 ± 2%. (**D, E**) Bilateral SNC was performed 3 weeks after pairing and tissue harvested at 7dpc. Longitudinal sciatic nerve sections from (**D**) WT and (**E**) *Sarm1-/-* parabionts showing infiltrating tdT+ leukocytes (magenta). Nuclei (blue) were labeled with Hoechst dye. The nerve crush site is marked by the white dashed line, proximal is to the left. Scale bar, 500 µm. (**F**) Quantification of tdT+ cells per high power field (HPF, 500 µm × 250 µm) at the injury site (0 µm) and at 1000, 2000, and 3000 µm distal to the injury site. The average cell number ± SEM is shown, n = 3 mice per genotype, average of four HPF per two nerves. Student's *t* test, *p<0.05. (**G**) HPF of sciatic nerves from WT and *Sarm1-/-* parabionts 7dpc taken from the injury site, 1000, 2000, and 3000 µm distal to the injury site showing infiltrating tdT+ leukocytes (magenta), F4/80+ macrophages (green), and nuclei (blue). Scale bar, 100 µm. (**H**) Quantification of tdT+F4/80+ cells per HPF ± SEM at indicated distances distal to the injury site, n = 3 mice, average of four HPF per two nerves. Student's *t* test, *p<0.05. (**I, J**) Flow cytometric analysis of sciatic nerves from single (not part of parabiosis complex) WT and *Sarm1-/-* mice 7dpc. Sciatic nerve trunks were microdissected and separated in 3 mm injury site and distal nerve segments. Dotplots showing Mo/Mac maturation assessed by Ly6C surface staining, Mo (Ly6Chi), Mo/Mac (Ly6Cint), Mac (Ly6C-), previously gated as CD45+CD11b+Ly6G-CD11c- cells. (**K**) Quantification of Mo/Mac shown in panels (**I**) and (**J**) as a percentage of single cells ± SEM, n = 3, injury and distal sites were pooled from 5 mice per genotype per biological replicate. Two-way ANOVA with Tukey's post-hoc test for multiple comparisons, *p<0.05, **p<0.01.

The online version of this article includes the following figure supplement(s) for figure 10:

**Figure supplement 1.** Neutrophils are reduced in the *Sarm1-/-* distal nerve.

undergo a rapid metabolic shift. The glycolytic burst is short-lived, however, since at 3dpc expression of glycolytic enzymes begins to decline, and at 7dpc levels have dropped to what is found in naive nerves. As glycolytic activity declines, there is evidence for a metabolic shift toward OXPHOS, and this coincides with the appearance of Mac with a resolving phenotype. Separation of the nerve injury site from distal nerve revealed that mechanical nerve injury creates two separate immune microenvironments: a wound repair response at the crush site, and an inflammatory response to WD in the distal nerve. This finding was independently corroborated by analysis of *Sarm1*[-/-] mice. At the injury site of *Sarm1*[-/-] and WT mice, nerve crush results in a strong immune response, dominated by hematogenous leukocytes. In the distal nerve of *Sarm1*[-/-] mice, full-blown nerve inflammation is delayed, and thus, WD dependent. Hematogenous immune cells are largely missing from the *Sarm1*[-/-] distal nerve, except for Mo, suggesting that chemotactic signals are released from severed fibers prior to physical disintegration. Taken together, we describe a framework of cell types and single-cell transcriptomes for a neural tissue with a high degree of spontaneous morphological and functional regeneration. The datasets reported provide an essential step toward understanding the dynamic nature of complex biological processes such as neural tissue degeneration and regeneration.

## iSNAT

To facilitate mining of scRNAseq datasets of naïve nerve and at different post-SNC time points, we generated theiSNAT. The *Expression Analysis* tool can be used to identify cells in the nerve that express a gene of interest and to determine relative abundance to other cells. The output is four feature pots (naïve, 1, 3, and 7dpc nerve) displayed side-by-side, showing which cell types express a gene of interest and if gene expression is regulated by nerve injury. In addition, any cell type in the nerve, at each of the four time points during the first week, can be selected to identify the top 50 enriched gene products. The *two Genes* function quantifies co-expression of two genes of interest in the same cell. Higher resolution UMAP plots of select cell types, such as *immune cells* and *structural cells* (stromal cells), can be accessed and mined separately for analysis of subcluster specific gene expression. We added single-cell transcriptomes of PBMC and used dataset integration to show how immune cell clusters change during the first week following nerve injury. Embedded in iSNAT is the *CellChat* function, designed for the identification of intercellular signaling networks. Families of surface or secreted molecules, e.g., CXCL family members, can be searched in the naïve and injured nerve to identify cells that express the corresponding receptor(s); the probability for a specific ligand–receptor interaction to occur is calculated by CellChat. This provides a powerful tool for understanding interactions among different cell types in the nerve. The *Spatial Distribution* function shows gene expression in immune cells at the injury site versus distal nerve at 3dpc. We have validated many gene products identified by scRNAseq using a combination of qRT-PCR, RNAscope, immunofluorescence staining, reporter gene expression, and ELISA. While iSNAT is expected to facilitate data analysis and functional studies in the injured PNS, there are some notable limitations. The reading depth at the single-cell level is still limited. On average, we detect ~2000 unique features per cell, well below the estimated 5000–10,000 per cell. Because of the stochastic nature of mRNA capture and the large number of cells sequenced (~157,000 high-quality cells), the coverage of the transcriptome of cell populations (>500 cells) is expected to be substantially higher, but still incomplete. Thus, if a gene of interest is not found in iSNAT, it may either not be expressed or be expressed below the detection sensitivity of the scRNAseq methods employed. We acknowledge that enzymatic tissue digestion combined with mechanical tissue dissociation may lead to cell loss or variable capturing efficiency for different cell types. Most notably, mSC are sparse in our naïve nerve dataset. Additional cell types that may be lost include B cells and adipocytes, both of which have previously been detected in the rodent PNS (*Chen et al., 2021*). Our gene expression atlas is work in progress, and we anticipate that future studies will overcome these limitations, allowing us to build on existing data and generate more advanced next generations of iSNAT.

## Structural cells in the injured nerve shape the immune microenvironment

A nerve crush injury triggers proliferation of stromal cells, including epineurial Fb, identified as *Mki67*[+]*Pdgfra*[+] cells that mature into Fb. The most noticeable expansion of stromal cells in the injured nerve is observed for dMES. In addition, pMES and eMES proliferate following SNC. In the UMAP plot

of 3dpc nerve, clusters representing proliferating structural cells are connected and give rise to pMES, eMES, and dMES. This suggests that in addition to damaging the epineurium, SNC damages protective cell layers within the nerve, including the perineurium, a thin cell layer of epithelioid myofibroblasts that surrounds nerve fascicles, and the endoneurium, a delicate layer of connective tissue that covers individual myelinated nerve fibers and contains the endoneurial fluid. Proliferation of eMES and pMES likely reflects damage to the BNB and is supported by the accumulation of some serum proteins in the crushed nerve. In addition to IgGs (*Vargas et al., 2010*), we detected other serum proteins that function in opsonization. These include complement components, soluble C1qR1/CD93, adiponectin (Adipoq), and pentraxins (Nptx2, Crp), molecules that aid in the clearance of cellular debris and apoptotic cells and push Mac toward an anti-inflammatory resolving phenotype (*Blackburn et al., 2019*; *Casals et al., 2019*; *Guo et al., 2012*).

Of interest, structural cells in the injured nerve consistently show the highest number of unique transcripts, indicative of a strong stromal cells response to injury response. MES are a major source of immune modulatory factors, shaping the injured nerve microenvironment in a paracrine manner through release of soluble factors. CellChat identified important roles in chemotaxis, angiogenesis, ECM deposition and remodeling, suggestive of extensive stroma-immune cell communication. In particular, eMES are a major signaling hub and show strong interactions with Mo, Mac, SC, pMES, and EC. The presence of several chemokines, growth factors, and immune proteins identified at the transcriptional level was independently validated by ELISA, providing confidence in the quality of our scRNAseq datasets.

## Cellular metabolism and macrophage functional polarization in the injured PNS

Evidence from injured non-neural tissues shows that immune cell metabolism is directly linked to cell plasticity and function, thereby affecting tissue repair and scarring (*Eming et al., 2021*; *Eming et al., 2017*). Little is known about the metabolic adaptions associated with neural tissue repair. Here, we compared immune cell metabolism of bone-marrow-derived circulating myeloid cells before and after nerve entry. Once in the injured nerve, neutrophils, Mo, and Mac undergo rapid metabolic reprogramming, greatly increasing gene products that drive glycolysis. This is similar to the metabolic shift observed in non-neural tissues with high regenerative capacities, such as skeletal muscle (*Eming et al., 2021*). Interestingly, rapid upregulation of glycolytic activity in myeloid cells in the injured nerve is reminiscent of the injury-regulated glycolytic shift in SC, involving the mTORC/Hif1α/c-Myc axis (*Babetto et al., 2020*). The increased glycolytic flux and lactate extrusion from SC is axoprotective (*Babetto et al., 2020*), and lactate released by innate immune cells may further augment the protective effects.

Many cells use aerobic glycolysis during rapid proliferation since glycolysis provides key metabolites for the biosynthesis of nucleotides and lipids (*Lunt and Vander Heiden, 2011*). In the 1dpc nerve, myeloid cells show highest expression of glycolytic enzymes; however, only a few *Mki67* (encoding Ki67+) proliferating cells are detected, suggesting elevated glycolysis serves other functions. Interestingly, in immune cells, alterations in metabolic pathways couple to immune cell effector function, most notably the production of different cytokines (*Mancebo et al., 2022*; *O'Neill et al., 2016*). Glucose is a main source for cellular energy (ATP) production through two linked biochemical pathways, glycolysis and the mitochondrial TCA cycle (*O'Neill et al., 2016*; *Voss et al., 2021*). Glycolysis converts glucose into pyruvate and pyruvate is converted into acetyl-CoA to enter the TCA and fuel OXPHOS in mitochondria as an efficient means of ATP production. Alternatively, pyruvate can be converted into lactate and NAD+, creating a favorable redox environment for subsequent rounds of glycolysis. The upregulation of *Ldha* and *Slc16a3*/MCT4 in myeloid cells of the injured nerve is striking and resembles the Warburg effect described for cancer cells (*Schuster et al., 2021*; *Zhu et al., 2015*). The transient increase in extracellular lactate may not only be axon protective and pro-regenerative (*Babetto et al., 2020*; *Fünfschilling et al., 2012*; *Li et al., 2020*), but additionally regulate immune cell reprogramming (*Morioka et al., 2018*) and trigger pain (*Rahman et al., 2016*). The injury-induced increase of *Hif1α* suggests that hypoxia is a main driver of metabolic reprogramming; however, the Hif1α pathway can also be activated by pattern recognition receptors recognizing DAMPs released by injured cells following trauma (*Corcoran and O'Neill, 2016*).

The molecular basis for Mac reprogramming into an anti-inflammatory state remains incompletely understood. A resolving Mac phenotype may be initiated by Mac-mediated engulfment and digestion of apoptotic cell corpses (*Boada-Romero et al., 2020*; *Greenlee-Wacker, 2016*). Mac in the injured sciatic nerve are fully equipped with the molecular machinery for efferocytosis, including phagocytic receptors, enzymes, and transporters to cope with elevated cholesterol load and other metabolic challenges (*Kalinski et al., 2020*). Parabiosis experiments, combined with SNC, revealed that clearing of apoptotic leukocytes, through efferocytosis, does take place in the injured PNS (*Kalinski et al., 2020*). Professional phagocytes that undergo multiple rounds of efferocytosis experience metabolic stress such as accumulation of intracellular lipids (*Schif-Zuck et al., 2011*). Growing evidence suggests that Mac leverage efferocytotic metabolites for anti-inflammatory reprogramming to promote tissue repair (*Zhang et al., 2019*). During the resolution phase, Mac are equipped with the machinery for fatty acid oxidation and OXPHOS as a means of energy production. Metabolic reprogramming of Mac is likely key for wound healing, axon regeneration, and restoration of neural function. Timely resolution of inflammation protects from excessive tissue damage and fibrosis. Interestingly, pMES may function as a lactate sink in the injured nerve since they express high levels of *Slc16a1*/MCT1 for cellular import, as well as *Ldhb* for conversion of lactate into pyruvate. This suggests that different cell types in the injured nerve employ different strategies to cover their bioenergetic needs. It will be interesting to examine whether Mo/Mac metabolic reprogramming, efferocytosis, and inflammation resolution are altered under conditions where nerve health and axon regeneration are compromised (*Sango et al., 2017*). Because nerve inflammation has been linked to the development of neuropathic pain (*Davies et al., 2020*), prolonged and poorly resolving nerve inflammation, due to impaired metabolic reprogramming, may contribute to pain syndromes.

## Identification of distinct immune compartments in the injured PNS

Mo/Mac are highly plastic cells that are educated by the microenvironment. Because a nerve crush injury causes location-specific changes in the microenvironments, it is perhaps not surprising that different Mac subpopulations exist and that they are not uniformly distributed along the injured nerve. At the site of injury, nerve trauma is caused by tissue compression, resulting in mechanical cell destruction, release of damage-associated molecular patterns (DAMPs), vascular damage, nerve bleeding, and disruption of tissue homeostasis. In the distal nerve, where physical trauma is not directly experienced, severed fibers undergo WD. Thus, mechanical nerve injury results in temporally and spatially distinct microenvironments. This was demonstrated by single-cell RNAseq of immune cells captured at the nerve injury site versus the distal nerve. We identified distinct, yet overlapping immune compartments, suggesting the existence of Mac populations that function in wound healing and Mac populations associated with WD. In the 3dpc nerve, Mac4 express the highest levels of efferocytotic receptors (*Kalinski et al., 2020*) and their enrichment at their nerve injury site suggests that phosphatidylserine-mediated clearance of apoptotic cells is primarily a response to tissue wounding.

Experiments with *Sarm1-/-* mice demonstrate that SNC triggers a spatially confined inflammatory response and accumulation of blood-borne immune cells independently of WD. Thus, physical nerve wounding and the resulting disruption of tissue homeostasis are sufficient to trigger robust local nerve inflammation. Our findings are reminiscent of a study in zebrafish larvae where laser transection of motor nerves resulted in Mac accumulation at the lesion site prior to axon fragmentation. Moreover, delayed fragmentation of severed zebrafish motor axons expressing the *Wld(s)* transgene did not alter Mac recruitment (*Rosenberg et al., 2012*). In injured *Sarm1-/-* mice, the distal nerve is much less inflamed when compared to parallel processed WT mice. Our observation is consistent with studies in *Wld(s)* mice, where reduced nerve inflammation has been reported (*Chen et al., 2015*; *Coleman and Höke, 2020*; *Lindborg et al., 2017*; *Perry and Brown, 1992*). Parabiosis experiments show that hematogenous WT immune cells readily enter the *Sarm1-/-* injury site, and to a much lesser extent, the distal nerve. Interestingly, Mo enter the *Sarm1-/-* distal nerve prior to fiber disintegration, but fail to mature into Mac. This suggests that severed, but physically intact PNS fibers in the *Sarm1-/-* distal nerve release chemotactic signals for Mo. Our studies show that SNC is sufficient to trigger Mo recruitment to the distal nerve, but WD is required for Mo differentiation and full-blown nerve inflammation.

## What drives WD-associated nerve inflammation?

Studies with injured *Sarm1-/-* mice show that full-blown inflammation of the distal nerve requires WD; the underlying molecular signals, however, remain incompletely understood. Because WD results in axon disintegration and simultaneous breakdown of myelin sheaths into ovoids, it is not clear whether myelin debris, SC activation, or axon fragmentation is the main trigger for WD-associated nerve inflammation. While Sarm1 is best known for its vital role in axon degeneration, evidence suggests that Sarm1 functions in myeloid cells to regulate innate immunity (*DiAntonio et al., 2021*; *Waller and Collins, 2022*). We used parabiosis to show that similar to *Sarm1-/-* Mac, wildtype Mac fail to accumulate in the distal nerve of injured *Sarm1-/-* mice, indicating that reduced inflammation is not due to loss of *Sarm1* in Mac.

In the PNS, axon fragmentation results in rapid SC activation and dedifferentiation into a p75-postive progenitor-like cell state. Of interest, transgenic expression of Raf-kinase in mSC in adult mice is sufficient to drive SC dedifferentiation without compromising axon integrity. SC dedifferentiation resulted in cytokine expression and nerve inflammation (*Napoli et al., 2012*). This suggests that dedifferentiated SC are sufficient to drive WD-like nerve inflammation; however, a detailed analysis of the immune cell composition, and comparison of Raf kinase transgenic mice to wildtype mice subjected to SNC, has not yet been carried out.

In the healthy PNS, the endoneurial milieu is protected by the BNB, a selectively permeable barrier formed by specialized EC along with the perineurial barrier. The BNB creates an immunologically and biochemically privileged space harboring nerve fibers and endoneurial fluid. ELISA of injured nerve tissue revealed that the BNB is at least partially compromised following SNC, resulting in local disturbances of vascular permeability, allowing access of serum proteins that function in opsonization and phagocytosis. This suggests that in addition to degenerating axons, myelin debris, and activated SC, disruption of the BNB may contribute to trauma-inflicted nerve inflammation. Additional studies are needed to fully define the mechanisms that underlie WD-associated nerve inflammation and its contribution to tissue repair.

Taken together, we carried out a longitudinal analysis of injured mouse PNS, naïve nerve, and PBMC transcriptomes at single-cell resolution. The study provides unprecedented insights into the dynamic landscape of cell states, cell–cell interaction networks, and immune cell metabolic reprograming during the first week following nerve crush injury. To facilitate dataset mining, we developed the iSNAT, a novel tool to navigate the cellular and molecular landscape of a neural tissue endowed with a high regenerative capacity.

# Materials and methods

## Mice and genotyping

All procedures involving mice were approved by the Institutional Animal Care and Use Committees (IACUC) of the University of Michigan (PRO 00009851) and performed in accordance with guidelines developed by the National Institutes of Health. Young adult male and female mice (8–16 weeks) on a C57BL/6 background were used throughout the study. Transgenic mice included *Sarm1/Myd88-5-/-* (Jackson Laboratories, Stock No: 018069), *ROSA26-mTdt/mGFP* (Jackson Laboratories, Stock No: 007576), and *Arg1-eYFP* reporter mice (Jackson Laboratories, Stock No: 015857). Mice were housed under a 12 hr light/dark cycle with regular chow and water ad libitum. For genotyping, ear biopsies were collected, and genomic DNA extracted by boiling in 100 µl alkaline lysis buffer (25 mM NaOH and 0.2 mM EDTA in ddH$_2$O) for 30 min. The pH was neutralized with 100 µl of 40 mM Tris-HCl (pH 5.5). For PCR, 1–5 µl of gDNA was mixed with 0.5 µl of 10 mM dNTP mix (Promega, C1141, Madison, WI), 10 µl of 25 mM MgCl$_2$, 5 µl of 5X Green GoTaq Buffer (Promega, M791A), 0.2 µl of GoTaq DNA polymerase (Promega, M3005), 1 µl of each PCR primer stock (100 µM each), and ddH$_2$O was added to a total volume of 25 µl. The following PCR primers, purchased from *Integrated DNA Technologies*, were used: *Sarm1* WT Fwd: 5′GGG AGA GCC TTC CTC ATA CC 3′; *Sarm1* WT Rev: 5′TAA GAA TGA GCA GGG CCA AG 3′; *Sarm1* KO Fwd: 5′CTT GGG TGG AGA GGC TAT TC 3′; *Sarm1* KO Rev: 5′AGG TGA GAT GAC AGG AGA TC 3′; *Rosa26* WT Fwd: 5′-CGT GAT CTG CAA CTC CAG TC-3′; *Rosa26* WT Rev: 5′-GGA GCG GGA GAA ATG GAT ATG-3′. PCR conditions were as follows: hot start 94°C 3 min; DNA denaturing at 94°C 30 s; annealing 60°C 1 min; extension 72°C 1 min, total cycles 34. Final extension for 6 min at 72°C.

## Surgical procedures

Mice were deeply anesthetized with a mixture of ketamine (100 mg/kg) and xylazine (10 mg/kg) or with isoflurane (5% induction, 2–3% maintenance, SomnoSuite Kent Scientific). Buprenorphine (0.1 mg/kg) was given as an analgesic. For SNC, thighs were shaved and disinfected with 70% isopropyl alcohol and iodine (PDI Healthcare). A small incision was made on the skin, underlying muscles separated, and the sciatic nerve trunk exposed. For sham-operated mice, the nerve was exposed but not touched. For SNC, the nerve was crushed for 15 s, using a fine forceps (Dumont #55). The wound was closed using two 7 mm reflex clips (Cell Point Scientific). Parabiosis surgery was performed as described (*Kalinski et al., 2020*). Briefly, before parabiosis surgery, similar-aged, same-sex mice were housed in the same cage for 1–2 weeks. Mice were anesthetized and their left side (host) or right side (donor) shaved and cleaned with iodine pads. A unilateral skin-deep incision was made from below the elbow to below the knee on the host and donor mouse. Mice were joined at the knee and elbow joints with non-absorbable nylon sutures. Absorbable sutures were used to join the skin around the shoulders and hindlimbs. Reflex wound clips (7 mm) were used to join the remainder of the skin between the two mice. Mice were allowed to recover for 3–4 weeks before use for SNC surgery.

## Histological procedures

Mice were euthanized with an overdose of xylazine/ketamine and transracially perfused for 5 min with ice-cold PBS followed by 5 min with freshly prepared ice-cold 4% paraformaldehyde in PBS. Sciatic nerve trunks were harvested and postfixed for 2 hr in ice-cold perfusion solution, followed by incubation in 30% sucrose in PBS solution at 4°C overnight. Nerves were covered with tissue Tek (Electron Microscopy Sciences, 62550-01) and stored at −80°C. Nerves were cryo-sectioned at 14 µm thickness and mounted on Superfrost+ microscope slides, air-dried overnight, and stored in a sealed slide box at −80°C. For antibody staining, slides were brought to RT and rinsed in 1× PBS three times, 5 min each. Slides were incubated in 0.3% PBST (1× PBS plus 0.3% Tween 20) for 10 min, followed by incubation for 1 hr in 5% donkey serum solution in 0.1% PBST (blocking buffer). Primary antibodies at appropriate dilutions were prepared in blocking solution, added to microscope slides, and incubated at 4°C overnight. The next day, sections were rinsed three times in PBS, 5 min each. Appropriate secondary antibodies in blocking buffer were added for 1 hr at a dilution of 1:1000 at room temperature (RT). Slides were rinsed three times in PBS, 5 min each, and briefly with MilliQ water. Sections were mounted in DAPI containing mounting medium (Southern Biotec [Cat# 0100-20]), and air-dried. Images were acquired with a Zeiss Apotome2 microscope equipped with an Axiocam 503 mono camera and ZEN software. Image processing and analysis were conducted using the ZEN software.

For in situ mRNA detection by RNAscope, mice were perfused for 5 min with ice-cold PBS, followed by RT 10% Neutral Buffered Formalin (NBF, Fisher Chemical, SF-100). Tissues were harvested and postfixed in 10% NBF overnight at RT. The RNAscope Multiplex Fluorescent Reagent Kit v2 (ACD, 323100) was used. Microscope slides with serially cut nerves were rinsed in 1× PBS for 5 min (repeated twice) and air-dried by incubation at 70°C for 10 min in an oven (VWR Scientific, Model 1525 incubator). Next, tissue sections were post-fixed in ice-cold 10% NBF for 15 min and dehydrated by incubation in a graded series of 50%, 70%, and 100% ethanol for 5 min each. Sections were air-dried for 5 min at RT and one drop of hydrogen peroxide solution (ACD Cat# PN 322381) was added to each nerve section on each slide and incubated at RT for 10 min. Sections were then submerged in 99°C RNase-free water for 15 s, followed by incubation in 99°C 1× antigen retrieval solution (ACD Cat# 322000) for 11 min. The slides were washed in MilliQ water twice, followed by a transfer to 100% ethanol for 3 min at RT. The slides were taken out and air-dried. Protease Plus solution (ACD Cat# PN 3223311) was applied to tissue sections followed by incubation at 40°C in an ACD hybridization oven (ACD Cat# 321710) for 10 min followed by MilliQ H$_2$O washes. RNA probes were mixed at appropriate ratios and volumes (typically 50:1 for C1:C2) for complex hybridization. For single RNA probe hybridization, RNA probes were diluted with probe dilutant at 1:50–1:100 (ACD Cat# 300041). Appropriate probes or the probe mixtures were applied to tissue sections and incubated for 2 hr in the hybridization oven at 40°C. A 1× wash buffer solution was prepared from a 50× stock (ACD Cat# PN 310091) and sections rinsed for 2 min. Amplification probe 1 was applied and slides incubated in a hybridization oven for 30 min 40°C and rinsed twice with 1× wash solution. Next, the A2 and A3 probes were applied. For development, the TSA system (AKOYA, Cy3: NEL744001KT; Cy5: NEL745001KT; Fluorescein: MEL741001KT) was used. Once the color for probe C1 was selected, HRPC1 solution (ACD Cat# 323120), it was applied

to the appropriate sections and incubated for 15 min in the hybridization oven at 40°C. The sections were then rinsed in 1× wash solution. Designated TSA color for probe C1, diluted in the TSA dilutant (ACD Cat# 322809) at 1:2000 was applied to the respective sections and incubated for 30 min in the ACD hybridization oven at 40°C. Sections were rinsed in 1× wash solution and then HRP blocker (ACD Cat# 323120) was applied and incubated for 15 min in the ACD hybridization oven at 40°C. This procedure was repeated for probes C2 and C3 as needed using HRPC2 and HRPC3, respectively. Sections were mounted in DAPI Southern Biotech mounting media (Cat# 0100-20), air-dried, and imaged and stored at 4°C in the dark. For quantification of labeled cells in nerve tissue sections, an FoV was defined, 200 μm × 500 μm at the nerve injury site and in the distal nerve. The FoV in the distal nerve was 2000 μm away from the injury site. The number of labeled cells per FoV was counted. Only labeled cells with a clearly identifiable nucleus were included in the analysis. The number of cells counted per FoV was from n = 3 mice with n = 2 technical replicates per mouse.

## Preparation of single-cell suspensions for flow cytometry and scRNAseq

Mice were euthanized and transcardially perfused with ice-cold PBS to reduce sample contamination with circulating leukocytes. Sciatic nerve trunks from naïve and injured mice were harvested. For injured mice, a segment was collected that includes the injury site and distal nerve just before the trifurcation of the tibial, sural, and peroneal nerves. Nerves were placed in ice-cold PBS containing actinomycin D (45 μM, Sigma-Aldrich, A1410). Some nerves were further dissected into 3 mm segments, either encompassing the site of nerve injury, ~1.5 mm proximal to ~1.5 mm distal of the crush site, or distal nerve, located between +1.5 to +4.5 mm away from the crush site. Nerves from three mice (6 mm segments) or five to six mice (3 mm segments) were pooled for each biological replicate. Nerves were minced with fine scissors and incubated in 1 ml PBS supplemented with collagenase (4 mg/ml Worthington Biochemical, LS004176) and dispase (2 mg/ml, Sigma-Aldrich, D4693) and incubated for 30–45 min at 37°C in a 15 ml conical tube. For scRNAseq, the digestion buffer also contained actinomycin D (45 μM). Nerves were triturated 20× with a 1000 μl pipette every 10 min and gently agitated every 5 min. Next, nerves were rinsed in DMEM with 10% FBS, spun down at 650 × $g$ for 5 min, the resulting pellet resuspended, and fractionated in a 30% Percoll gradient. For flow cytometry, the cell fraction was collected and filtered through a pre-washed 40 μm Falcon filter (Corning, 352340) and cells were pelleted at 650 × $g$ for 5 min at 4°C. Immune cell populations were identified with established antibody panels as described (*Kalinski et al., 2020*). For scRNAseq, the cell suspension was cleared of myelin debris with myelin removal beads (Miltenyi, 130-096-733), and cells resuspended in Hanks balanced salt solution (Gibo, 14025092) supplemented with 0.04% BSA (Fisher Scientific, BP1600). To enrich for immune cells, some samples were run over an anti-CD45 or anti-CD11b column (Miltenyi, 130-052-301). Cells were counted and live/dead ratio determined using propidium iodine staining and a hemocytometer (*Kalinski et al., 2020*). Blood was collected from adult naïve mice by cardiac puncture and collected into K2 EDTA-coated tubes (BD 365974) to prevent coagulation. Approximately 500 μl of blood was passed through a 70 μm cell strainer in 5 ml ACK (ammonium-chloride-potassium) lysis buffer. Blood was incubated at RT for 5 min and erythrocyte lysis stopped by addition of 15 ml FACS buffer, followed by leukocyte spin down in a clinical centrifuge. This process was repeated three times for complete erythrocyte lysis.

## Barcoding and library preparation

The Chromium Next GEM Single Cell 3' Reagent kit v3.1 (Dual Index) was used. Barcoding and library preparation was performed following the manufacturer's protocols. Briefly, to generate single-cell gel-bead-in-emulsion (GEMs) solution, approximately 15,000 cells, in a final volume of 43 μl, were loaded on a Next GEM Chip G (10X Genomics) and processed with the 10X Genomics Chromium Controller. Reverse transcription was performed as follows: 53°C for 45 min and 85°C for 5 min in a Veriti Therml Cycler (Applied Biosystems). Next, first-strand cDNA was cleaned with DynaBeads MyOne SILANE (10 X Genomics, 2000048). The amplified cDNA, intermedium products, and final libraries were prepared and cleaned with SPRIselect Regent kit (Beckman Coulter, B23318). A small aliquot of each library was used for quality control to determine fragment size distribution and DNA concentration using a bioanalyzer. Libraries were pooled for sequencing with a NovaSeq 6000 (Illumina) at an estimated depth of 50,000 reads per cell, yielding 11.3 billion reads. NovaSeq control

software version 1.6 and Real Time Analysis (RTA) software 3.4.4 were used to generate binary base call (BCL) formatted files.

## scRNAseq data analysis

Raw scRNAseq data were processed using the 10X Genomics CellRanger software version 3.1.0. The CellRanger 'mkfastq' function was used for de-multiplexing and generating FASTQ files from raw BCL. The CellRanger 'count' function with default settings was used with the mm10 reference supplied by 10X Genomics, to align reads and generate single-cell feature counts. CellRanger filtered cells and counts were used for downstream analysis in Seurat version 4.0.5 implemented in R version 4.1.2. Cells were excluded if they had fewer than 500 features, more than 7500, or the mitochondrial content was more than 15%. For each post-injury time point, reads from multiple samples were integrated and normalized following a standard Seurat SCTransform+ CCA integration pipeline (*Hafemeister and Satija, 2019*). The mitochondrial mapping percentage was regressed out during the SCTransform normalization step. PCA was performed on the top 3000 variable genes and the top 30 principal components were used for downstream analysis. A K-nearest-neighbor graph was produced using Euclidean distances. The Louvain algorithm was used with resolution set to 0.5 to group cells together. Nonlinear dimensional reduction was done using UMAP. The top 100 genes for each cluster, determined by Seurat's FindAllMarkers function and the Wilcoxon rank-sum test, were submitted to QIAGEN's Ingenuity Pathway Analysis (IPA) software version 70750971 (QIAGEN Inc, https://digitalinsights.qiagen.com/IPA) using core analysis of up- and downregulated expressed genes. Top-scoring enriched pathways, functions, upstream regulators, and networks for these genes were identified utilizing the algorithms developed for QIAGEN IPA software (*Krämer et al., 2014*), based on QIAGEN's IPA database of differentially expressed genes.

## Comparative analysis of cell identities at different post-injury time points

Comparison of cell identities between time points was done using the Seurat technique for classifying Cell Types from an integrated reference. This technique projects the PCA structure of the reference time point onto the query time point. This is similar to Seurat's implementation of canonical correlation analysis (CCA) in that it creates anchors between the two datasets, but it stops short of modifying the expression values of the query. The output of this technique is a matrix with predicted IDs and a prediction score between 0 and 1. For each reference cell type, we used the geometric mean of prediction scores for cells predicted with that type in the query set. This single prediction score was used as a surrogate for confidence in the same cell state existing in the query cells. Alternatively, using the overlapping, top 3000, highly variable genes from each dataset, we computed the Pearson correlation between the genes' average expression (log2 of uncorrelated 'RNA' assay counts) from each reference cell type and its predicted query cells.

## CellChat

CellChat version 1.1.3, with its native database, was used to analyze cell–cell communication (*Jin et al., 2021*; https://doi.org/10.1038/s41467-021-21246-9). A truncated mean of 25% was used for calculating average mean expression, meaning a gene was considered to have no expression in a cell type if fewer than 25% of cells in that cell type expressed the gene.

## Differential gene expression

Based on recent recommendations and updates to Seurat https://doi.org/10.1186/s13059-021-02584-9 we used GitHub Seurat version 4.1.1.9006 for downstream differential expression. The same integration pipeline was used but with SCTransform 'V2.' Because PBMC lack stromal cells, two independent integrations were performed. One with all cells from naïve, 1dpc, 3dpc, and 7dpc nerves, and the other limited to only immune cells from 1dpc, 3dpc, and 7dpc nerves and PBMC, except for B cells since no B cells were found in naïve or injured nerve tissue. Gamma-Poisson Generalized Linear Models were fit using glmGamPoi package from Bioconductor (https://doi.org/10.1093/bioinformatics/btaa1009). Genes not expressed in at least 25% of cells being compared were removed.

## SlingShot

SlingShot version 2.2.1 was used to model trajectories in the 'V2' integrated myeloid dataset. The PCA embeddings from the first four principal components were used as input and the beginning of

the trajectory anchored at the Monocytes. The Mo to Mac and Mo to MoDC trajectories were selected as interesting. To predict the genes that contribute most to pseudotime, we used the tidymodels R package (Hadley Wickham and Max Kuhn; https://www.tidymodels.org/); specifically, regression models in random forests with 200 predictors (genes) randomly samples at each split and 1400 trees. The Impurity method was used to calculate a genes importance in predicting the pseudotime. Heatmaps of the top 30 genes by importance were used to visually examine how the gene changes in cells ordered by their pseudotime.

## Code availability

iSNAT, an interactive web application, was generated with the RStudio's shiny package (https://github.com/rstudio/shiny; *Chang et al., 2022*). The Dashboard format was generously supplied by the RStudio group (https://rstudio.github.io/shinydashboard/; *R Core Team, 2018*). The code for all analysis and Rshiny server is available from GitHub (https://github.com/GigerLab/iSNAT, (*Giger Lab, 2022*; copy archived at swh:1:rev:2cdf1b41d0e22efac91ce1e2272a696e69d86104)).

## Protein analysis

For protein analysis, mice (naïve, and at 1, 3, and 7dpc) were euthanized and transcardially perfused with ice-cold PBS for 5 min. Naïve and injured sciatic nerves were dissected, with nerves from three mice pooled per time point in ice-cold PBS with 1% protease inhibitor cocktail (Sigma-Aldrich, P8340). Samples were minced and homogenized in 1% Triton X-100 (Sigma-Aldrich, T8787). Samples were frozen at –80°C, thawed, centrifuged at 10,000 × $g$ for 5 min to pellet cellular debris. The protein concentrations of the nerve supernatants and serum were determined using a BCA protein concentration assay. Western blot analysis of sciatic nerve tissue was carried out as described previously (*Kalinski et al., 2020*). For some WT and *Sarm1-/-* mice, injured nerves were divided into proximal, injury site, and distal segments. For the proteome Profiler Mouse XL Cytokine array, 200 µg of protein was applied to each ELISA membrane and developed according to the manufacturer's instructions (Proteome Profiler Mouse XL Cytokine Kit, ARY028, R&D Systems, Minneapolis, MN). Cytokine array signals were detected by X-ray film, scanned, and quantified using LI-COR Image Studio software version 5.2.5. Cytokine signals (pixel-density value) from duplicate spots were averaged, then normalized to the reference spots in the upper right and left corners and the lower-left corner on the membrane. Representative images of array membranes are shown (n = 1–2 biological replicates per condition).

## qRT-PCR

Quantitative PCR (qPCR) was carried out in triplicate with SYBR Green Fluorescein Master Mix (Thermo Scientific Cat# 4364344) on a QuantStudio 3 real-time PCR system (Applied Biosystems Cat# A28567). The ΔΔCt method was used to determine the relative expression of mRNAs, which was normalized to *Rlp13a* mRNA levels. Mice were transcardially perfused with ice-cold PBS, sciatic nerve trunks harvested, and collected in an RNase-free 1.5 ml Eppendorf tube in 1 ml TRIzol solution (Thermo Fisher Cat# 15596026). Nerves were minced into small pieces with RNase-free spring scissors, homogenized using a pestle motor mixer (RPI, 299200), and frozen at –80°C overnight. The next day, specimens were thawed, placed on ice, and 0.2 ml of chloroform was added to the TRIzol mix and shaken thoroughly for 5 min. Samples were centrifuged at 12,000 × $g$, 4°C for 15 min in a tabletop centrifuge. The aqueous phase was removed and placed in a new Eppendorf tube. Glycogen (2 µl) and isopropanol (0.5 ml) were added to the aqueous solution and mildly shaken with a shaker at 4°C for 10 min. Samples were centrifuged at 12,000 × $g$ at 4°C for 10 min to precipitate total RNA. The supernatant was discarded, and the pellet resuspended in 1 ml 75% ethanol, vortexed briefly, and then centrifuged 4°C for 5 mins at 7500 × $g$. The supernatant was discarded, the pellet airdried for 1–2 hr, and resuspended in 20 µl of RNase-free water. RNA yield was quantified with a nanodrop (Thermo Scientific, Nanodrop One), RNA aliquoted, and stored at –80°C.

RNA was reverse transcribed into first-strand cDNA using the Invitrogen SuperScript III First-Strand Synthesis System kit (Cat# 18080051). Briefly, to 250 ng of total RNA, 1 µl of 50 µM oligo(dT)20 primer, 1 µl of 10 mM dNTP mix were added and the final volume adjusted to 10 µl using RNase-free water. The RNA mix was incubated at 65°C for 5 min and quickly placed on ice for at least 5 min. In a separate tube, a master mix was prepared containing 2 µl of 10× RT buffer, 4 µl of 25 mM MgCl$_2$, and 2 µl of

0.1 M DTT. 8 µl of the master mix was added to 10 µl of RNA mix with 1 µl of SuperScript III reverse transcriptase and 1 µl of RNaseOut enzyme. The reverse transcription reaction (final volume 20 µl) was carried out at 50°C for 50 min, followed by a 5 min incubation at 85°C for 5 min. RNase H (1 µl) was added to the first-strand cDNA and incubated at 37°C for 20 min. First-strand cDNA was quantified using a Nanodrop One and stored in aliquots at –20°C. The cDNA was diluted to 50 ng/ul and used for qPCR in a 96-well plate format. SYBR Green Fluorescein Master Mix 10 µl (Thermo Scientific Cat# 4364344), 0.4 µl of forward primer, 0.4 µl of reverse primer, first-strand cDNA (50 ng) was mixed, and the final volume adjured to 20 µl per well with MilliQ water. The plate was sealed, centrifuged at 2500 × $g$ for 1 min, and then placed in QuantStudio 3 real-time PCR system (Applied Biosystems Cat# A28567) to run the qPCR. The protocol for running was as follows. Data was exported in Excel format and analyzed in Excel using the ΔΔCt method where the gene control was *Rlp13a* while the reference control was the sciatic nerve in naïve condition.

## Statistical methods

Data are presented as mean ± SEM. Statistical analysis was performed in GraphPad Prism (v7) using paired or unpaired two-tailed Student's $t$ test, or one-way or two-way ANOVA with correction for multiple comparisons with Tukey's post-hoc test, as indicated in the figure legends. A *p-value$<0.05$ was considered significant. **p$<0.01$, ***p$<0.001$, and ****p$<0.0001$. Data acquisition and analysis were carried out by individuals blinded to experimental groups.

## Acknowledgements

We thank the members of the Giger lab for critical reading of the manuscript and Qing Wang for excellent technical support. Richard Locksley (UCSF) for providing Arg1-YFP reporter mice. This work was supported by NIH T32 NS07222, Training in Clinical and Basic Neuroscience (AK), NIH T32 GM113900, Training Program in Translational Research (LH). The National Institutes of Health, MH119346 (RG), R01DC018500 (GC), the University of Michigan MICHR seed funds (RG and GC), and the Dr. Miriam and Sheldon G Adelson Medical Research Foundation (RK, JT, DG, RG). *This paper is dedicated to the memory of Dr. Lynda Jun-San Yang.*

## Additional information

### Competing interests

Gabriel Corfas: Except for Gabriel Corfas, the authors declare no competing financial or non-financial interests. Gabriel Corfas is a scientific founder of Decibel Therapeutics; he has an equity interest in and has received compensation for consulting. The company was not involved in this study. The other authors declare that no competing interests exist.

### Funding

| Funder | Grant reference number | Author |
| --- | --- | --- |
| NIH Blueprint for Neuroscience Research | MH119346 | Roman J Giger |
| NIH Blueprint for Neuroscience Research | R01DC018500 | Gabriel Corfas |
| NIH Blueprint for Neuroscience Research | T32 NS07222 | Ashley L Kalinski |
| National Institutes of Health | T32 GM113900 | Lucas D Huffman |
| Dr Miriam and Sheldon G Adelson Medical Research Foundation | | Riki Kawaguchi Jeffery L Twiss Daniel H Geschwind Roman J Giger |

| Funder | Grant reference number | Author |
|---|---|---|

The funders had no role in study design, data collection and interpretation, or the decision to submit the work for publication.

## Author contributions

Xiao-Feng Zhao, Formal analysis, Supervision, Investigation, Methodology; Lucas D Huffman, Matthew C Finneran, Formal analysis, Validation, Investigation, Methodology; Hannah Hafner, Formal analysis, Validation, Investigation, Visualization, Methodology; Mitre Athaiya, Formal analysis, Validation, Investigation, Visualization; Ashley L Kalinski, Formal analysis, Investigation, Visualization, Methodology; Rafi Kohen, Formal analysis, Investigation, Methodology; Corey Flynn, David Kohrman, Methodology; Ryan Passino, Investigation, Methodology; Craig N Johnson, Data curation, Software, Visualization, Methodology; Riki Kawaguchi, Data curation, Software, Methodology; Lynda JS Yang, Supervision, Methodology; Jeffery L Twiss, Supervision, Funding acquisition; Daniel H Geschwind, Software, Formal analysis, Funding acquisition; Gabriel Corfas, Supervision, Funding acquisition, Methodology; Roman J Giger, Conceptualization, Resources, Data curation, Formal analysis, Supervision, Funding acquisition, Validation, Investigation, Visualization, Methodology, Writing – original draft, Project administration, Writing – review and editing

## Author ORCIDs

Xiao-Feng Zhao ⓘ http://orcid.org/0000-0002-7574-7163
Lucas D Huffman ⓘ http://orcid.org/0000-0002-7779-086X
Ashley L Kalinski ⓘ http://orcid.org/0000-0001-7611-0810
Corey Flynn ⓘ http://orcid.org/0000-0001-5811-7269
Craig N Johnson ⓘ http://orcid.org/0000-0003-3955-7394
Jeffery L Twiss ⓘ http://orcid.org/0000-0001-7875-6682
Daniel H Geschwind ⓘ http://orcid.org/0000-0003-2896-3450
Gabriel Corfas ⓘ http://orcid.org/0000-0001-5412-9473
Roman J Giger ⓘ http://orcid.org/0000-0002-2926-3336

## Ethics

All procedures involving mice were approved by the Institutional Animal Care and Use Committees (IACUC) of the University of Michigan (PRO 00009851) and performed in accordance with guidelines developed by the National Institutes of Health.

## Decision letter and Author response

Decision letter https://doi.org/10.7554/eLife.80881.sa1
Author response https://doi.org/10.7554/eLife.80881.sa2

# Additional files

## Supplementary files

• Supplementary file 1. Datasets included in injured sciatic nerve atlas (iSNAT). The table shows newly generated and existing scRNAseq datasets used in this study. Columns show cell numbers before and after applying exclusion criteria, replicates, statistics on reading depth and sequence saturation. A total of 157,409 high-quality single-cell transcriptomes were analyzed from naïve mouse sciatic nerve, injured sciatic nerves, and peripheral blood mononuclear cells (PBMC). Some of the 3 day (3d) injured nerves were divided into injury site and distal nerve and sequenced separately. SN, sciatic nerve; UMI, unique molecular identifier.

• MDAR checklist

## Data availability

All scRNA-seq datasets (fastq files and Seurat objects) are available online through the Gene Expression Omnibus (GEO) database https://www.ncbi.nlm.nih.gov/geo, accession number GSE198582. All code for iSNAT is available at https://github.com/GigerLab/iSNAT, (copy archived at swh:1:rev:2cdf1b41d0e22efac91ce1e2272a696e69d86104).

The following dataset was generated:

| Author(s) | Year | Dataset title | Dataset URL | Database and Identifier |
| --- | --- | --- | --- | --- |
| Giger RJ, Zhao X-F, Johnson CN | 2022 | Injured Sciatic Nerve Atlas (iSNAT) | https://www.ncbi.nlm.nih.gov/geo/query/acc.cgi?acc=GSE198582 | NCBI Gene Expression Omnibus, GSE198582 |

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

# Appendix 1

## Appendix 1—key resources table

| Reagent type (species) or resource | Designation | Source or reference | Identifiers | Additional information |
|---|---|---|---|---|
| Antibody | Neurofilament heavy chain (chicken polyclonal) | Aves Lab | NFH | 1:750 |
| Antibody | Anti-chicken Cy3 (donkey polyclonal) | Jackson Immunoresearch | 703-165-155 | 1:200 |
| Antibody | Iba1 (rabbit polyclonal) | Wako Chemicals | 019-19741 | 1:500 |
| Antibody | F4/80 (rat IgG2b monoclonal) | Thermo Fisher Scientific | ma1-91124 | 1:500–1:1000 |
| Antibody | CD68 (rabbit polyclonal) | Abcam | ab125212 | 1:500 |
| Antibody | SCG10 (rabbit polyclonal) | Novus Biologicals | NBP149461 | 1:500–1:1000 |
| Antibody | CD11b (rabbit monoclonal) | Abcam | ab133357 | 1:200–1:1000 |
| Antibody | ERK1/2 (rabbit polyclonal) | Cell Signaling | 9102 | 1:5000 |
| Antibody | Anti-rabbit HRP (donkey polyclonal) | EMD Millipore | AP182P | 1:2000–1:10,000 |
| Antibody | CD16/32 (rat IgG2a monoclonal) | BD Pharmingen | 553141 | 1 µg/1 million cells/25 µl |
| Antibody | CD11b-PE-Cy7 (rat IgG2b monoclonal) | Thermo Fisher Scientific | 25-0112-82 | 1:200 |
| Antibody | Isotype Control-PE-Cy7 (rat-IgG2b monoclonal) | Thermo Fisher Scientific | 25-4031-82 | 1:100 |
| Antibody | CD45-e450 (rat-IgG2b monoclonal) | Thermo Fisher Scientific | 48-0451-82 | 1:100 |
| Antibody | CD45.1-e450 (mouse-IgG2a monoclonal) | BioLegend | 110721 | 1:100 |
| Antibody | Isotype Control-e450 (mouse-IgG2a monoclonal) | BioLegend | 400235 | 1:100 |
| Antibody | CD45.2-APC (mouse-IgG2a monoclonal) | BioLegend | 109813 | 1:100 |
| Antibody | Isotype Control-APC (mouse-IgG2a monoclonal) | BioLegend | 400221 | 1:100 |
| Antibody | Ly6G-APC-Cy7 (rat-IgG2a monoclonal) | BD Biosciences | 560600 | 1:100 |
| Antibody | CD11c-PerCP-Cy5.5 (ArmHam-IgG monoclonal) | Thermo Fisher Scientific | 45-0114-82 | 1:100 |
| Antibody | Isotype Control-PerCP-Cy5.5 (ArmHam-IgG monoclonal) | Thermo Fisher Scientific | 45-4888-80 | 1:100 |
| Antibody | Ly6C-FITC (rat-IgM monoclonal) | BD Biosciences | 553104 | 1:100 |
| Antibody | Isotype Control-FITC (rat-IgM monoclonal) | BD Biosciences | 553942 | 1:100 |
| Antibody | Iba1 (goat polyclonal) | Novus Biologicals | NB100-1028 | 1:200 |
| Antibody | Anti-goat Alexa Fluor 488 (donkey polyclonal) | Jackson Immunoresearch | 705-545-147 | 1:200 |

*Appendix 1 Continued on next page*

*Appendix 1 Continued*

| Reagent type (species) or resource | Designation | Source or reference | Identifiers | Additional information |
|---|---|---|---|---|
| Antibody | LDHA (rabbit polyclonal) | Cell Signaling Technology | 2558 | 1:300 |
| Chemical compound, drug | TOPRO pan-nuclear stain | Thermo Fisher Scientific | T3605 | 1:2000 |
| Chemical compound, drug | Fixable Viability Dye | Thermo Fisher Scientific | 65086614 | 1:500 |
| Chemical compound, drug | Proteinase K | New England Biolabs | P8107S | |
| Chemical compound, drug | 10 mM dNTP mix | Promega | C1141 | |
| Chemical compound, drug | 5X Green GoTaq Buffer | Promega | M791A | |
| Chemical compound, drug | GoTaq DNA polymerase | Promega | M3005 | |
| Chemical compound, drug | Buprenorphine | Par Pharmaceutical | NDC12496-0757-1 | |
| Chemical compound, drug | Ketamine | Par Pharmaceutical | NDC42023-115-10 | |
| Chemical compound, drug | Xylazine | Akorn | NDC59399-110-20 | |
| Chemical compound, drug | Fluriso (Isoflurane, USP) | Vet One | 501017 | |
| Chemical compound, drug | Rhodamine-conjugated dextran MW 3000 (Microruby) | Life Technologies | D-7162 | |
| Chemical compound, drug | Cholera toxin B (CTB) | Life Technologies | C34775 | |
| Chemical compound, drug | Puralube Eye ointment | Dechra | NDC-17033-211-38 | |
| Chemical compound, drug | N2 media supplement | Gibco | 17502048 | Cell culture |
| Chemical compound, drug | N1 media supplement | Sigma | N6530 | Cell culture |
| Chemical compound, drug | Leibovitz-15 (L-15) | Gibco | 21083-027 | Cell culture |
| Chemical compound, drug | Penicillin/Streptomycin | Life Technologies | 15140-122 | Cell culture |
| Chemical compound, drug | DMEM Ham's F-12 | Gibco | 10565-018 | Cell culture |
| Chemical compound, drug | Fetal bovine serum | Atlanta Biologicals | S11550 | Cell culture |
| Chemical compound, drug | Cytosine arabinoside | Sigma-Aldrich | C1768 | Cell culture |

*Appendix 1 Continued on next page*

*Appendix 1 Continued*

| Reagent type (species) or resource | Designation | Source or reference | Identifiers | Additional information |
| --- | --- | --- | --- | --- |
| Chemical compound, drug | Collagenase type 2 | Worthington Biochemical | LS004176 | Tissue digestion |
| Chemical compound, drug | PBS without calcium, magnesium | Gibco | 10010023 | Cell culture |
| Chemical compound, drug | Poly-L-lysine MW 70,000–150,000 | Sigma-Aldrich | P4707 | Cell culture |
| Chemical compound, drug | Laminin | Sigma-Aldrich | L2020 | Cell culture |
| Chemical compound, drug | Paraformaldehyde | Sigma-Aldrich | 158127-500G | |
| Chemical compound, drug | Triton-X100 | Sigma-Aldrich | T8787 | |
| Chemical compound, drug | Bovine serum albumin (BSA) heat shock fraction V | Fisher Scientific | BP1600 | |
| Chemical compound, drug | Hoechst 33342 | Invitrogen | H3570 | Nuclear dye |
| Chemical compound, drug | Tissue-Tek O.C.T. Compound | Electron Microscopy Sciences | 62550-01 | |
| Chemical compound, drug | β-Glycerophosphate | Sigma-Aldrich | G9422-100G | |
| Chemical compound, drug | Sodium orthovanadate (Na3VO4) | Sigma-Aldrich | S6508-10G | |
| Chemical compound, drug | Protease inhibitor cocktail | Sigma-Aldrich | P8340-5ML | |
| Chemical compound, drug | DC Protein Assay Kit | Bio-Rad | 5000111 | |
| Chemical compound, drug | 2× Laemmli sample buffer | Bio-Rad | 1610737 | |
| Chemical compound, drug | β-Mercaptoethanol | EMD Millipore | 6010 | |
| Chemical compound, drug | Blotting-grade blocker | Bio-Rad | 1706404 | |
| Chemical compound, drug | SuperSignal West Pico PLUS Chemiluminescent Substrate | Thermo Fisher Scientific | 34580 | |
| Chemical compound, drug | SuperSignal West Femto Maximum Sensitivity Substrate | Thermo Fisher Scientific | 34095 | |
| Chemical compound, drug | WesternSure PREMIUM Chemi Substrate | LI-COR Biosciences | 926-95000 | |
| Chemical compound, drug | Fixable Viability Dye eF506 | Thermo Fisher Scientific | 65-0866-14 | |

*Appendix 1 Continued on next page*

*Appendix 1 Continued*

| Reagent type (species) or resource | Designation | Source or reference | Identifiers | Additional information |
|---|---|---|---|---|
| Chemical compound, drug | TRIzol | Thermo Fisher Scientific | 15596026 | |
| Chemical compound, drug | Dispase | Sigma-Aldrich | D4693 | |
| Chemical compound, drug | Actinomycin D | Sigma-Aldrich | A1410 | |
| Chemical compound, drug | Percoll | Sigma-Aldrich | P4937 | |
| Chemical compound, drug | MACS buffer | Miltenyi | 130-091-376 | |
| Chemical compound, drug | CD45 MicroBeads | Miltenyi | 130-052-301 | |
| Chemical compound, drug | CD11b MicroBeads | Miltenyi | 130-049-601 | |
| Chemical compound, drug | Myelin removal Beads | Miltenyi | 130-096-733 | |
| Chemical compound, drug | LS Columns | Miltenyi | 130-042-401 | |
| Chemical compound, drug | Hanks balanced salt solution | Gibco | 14025092 | |
| Chemical compound, drug | Sucrose | Fisher Scientific | S5-500 | |
| Other | Superfrost Plus Microscope Slides | Fisher Scientific | 12-550-15 | For histology |
| Other | Zeiss Axio Observer Z1 | Zeiss | 491912-0049-000 | Microscope |
| Other | Zeiss Axiocam 503 mono camera | Zeiss | 426559-0000-000 | Microscope camera |
| Other | EC PlnN ×10 objective | Zeiss | 420341-9911-000 | Objective for microscope |
| Other | Motorized tissue homogenizer | RPI | 299200 | Homogenization of nerve tissue |
| Other | Fisher Scientific Sonic Dismembrator | Fisher Scientific | Model 500 | Western blot equipment |
| Other | photo spectrometer | Molecular Devices | SpectraMax M5e | Measurement of protein concentration |
| Other | LI-COR C-Digit | LI-COR Biosciences | CDG-001313 | Scanning of Western blot membranes |
| Other | 40 µm filter | BD Falcon | 352340 | Cell isolation |
| Other | 70 µm cell strainer | Corning | 352350 | Cell isolation |
| Chemical compound, drug | PVDF membrane | EMD Millipore | IPVH00010 | |
| Chemical compound, drug | Ammonium-chloride-potassium (ACK) Lysing Buffer | Gibco | A1049201 | Removal of erythrocytes |
| Commercial assay or kit | Myelin Removal Beads | Miltenyi | 130-096-731 | |
| Commercial assay or kit | MidiMACS separator | Miltenyi | 130-042-302 | |

*Appendix 1 Continued on next page*

*Appendix 1 Continued*

| Reagent type (species) or resource | Designation | Source or reference | Identifiers | Additional information |
|---|---|---|---|---|
| Other | LS Columns | Miltenyi | 130-042-401 | Cell sorting |
| Other | Hemacytometer | MilliporeSigma | Z359629 | Cell counting |
| Commercial assay or kit | Chromium Next GEM Chip G | 10X Genomics, Inc | NC1000127 | |
| Other | 10X Genomic Chromium Controller | 10X Genomics, Inc | GCG-SR-1 | Barcoding of cells for scRNA-sequencing |
| Commercial assay or kit | Chromium Next GEM Single Cell 3' Kit v3.1 | 10X Genomics, Inc | 1000268 | |
| Commercial assay or kit | Chromium Next GEM Chip G Single Cell Kit | 10X Genomics, Inc | 1000127 | |
| Commercial assay or kit | Dual Index Kit TT Set A | 10X Genomics, Inc | 1000125 | |
| Chemical compound, drug | Dynabeads | 10X Genomics, Inc | 2000048 | |
| Chemical compound, drug | SPRIselect | Beckman Coulter | B23318 | |
| Other | NovaSeq Illumina 6000 | Illumina | N/A | DNA library sequencing |
| Other | Cryostat | Leica Biosystems | CM3050S | Tissue sectioning |
| Other | Confocal Microscope | Nikon | C1 | Imaging of tissue sections |
| Other | Confocal Microscope | Leica Biosystems | SP8 | Imaging of tissue sections |
| Commercial assay or kit | Proteome Profiler, Mouse XL Cytokine membranes (ELISA) | R&D Systems, Minneapolis, MN, USA | ARY028 | |
| Strain, strain background (*Mus musculus*) | *Sarm1-/-* C57BL/6 | Jackson Laboratories | Stock# 018069 | PMID:22678360 |
| Strain, strain background (*M. musculus*) | *ROSA26-mTdt/mGFP* C57BL/6 | Jackson Laboratories | Stock# 007576 | PMID:17868096; MGI: J:124702 |
| Strain, strain background (*M. musculus*) | *Arg1-eYFP* C57BL/6 | Jackson Laboratories | Stock# 015857 | PMID:17450126; MGI: J:122735; PMID:33263277 |
| Strain, strain background (*M. musculus*) | *CD45.1* C57BL/6 | Jackson Laboratories | Stock# 002014 | PMID:11698303; MGI: J:109863; PMID:11994430; MGI: J:109854; PMID:12004082; MGI: J:109853 |
| Strain, strain background (*M. musculus*) | Wildtype, WT C57BL/6 | Taconics | B6NTac | |
| Sequence-based reagent | Neomycin Forward | Integrated DNA Technologies | N/A | 5'-CTTGGGTGGAGAGGCTATTC-3' |
| Sequence-based reagent | Neomycin Reverse | Integrated DNA Technologies | N/A | 5'-AGGTGAGATGACAGGAGATC-3' |
| Software, algorithm | WIS-Neuromath | Weizmann Institute of Science | Version 3.4.8 | PMID:23055261 |
| Software, algorithm | Image Studio Software | LI-COR Biosciences | Version 5.2.5 | |
| Software, algorithm | NovaSeq control software | Illumina | Version 1.6 | |
| Software, algorithm | Real Time Analysis (RTA) software | Illumina | Version 3.4.4 | |
| Software, algorithm | CellRanger | 10X Genomics, Inc | Version 3.1.0 | |

*Appendix 1 Continued on next page*

*Appendix 1 Continued*

| Reagent type (species) or resource | Designation | Source or reference | Identifiers | Additional information |
|---|---|---|---|---|
| Software, algorithm | Seurat | Satija Lab–New York Genome Center | Version 4.0.5 | |
| Software, algorithm | Seurat | https://github.com/satijalab/seurat; *Srivastava and Hoffman, 2022; Hao et al., 2021* | Version 4.1.1.9006 | |
| Software, algorithm | R | r-project.org | Version 4.1.2 | |
| Software, algorithm | SlingShot | bioconductor.org | Version 2.2.1 | |
| Software, algorithm | Ranger | Comprehensive R Archive Network | Version 0.13.1 | |
| Software, algorithm | CellChat | https://github.com/sqjin/CellChat; *Jin and CaoWei-UM, 2022; Jin et al., 2021* | Version 1.1.3 | |
| Software, algorithm | shiny | Rstudio.com | Version 1.7.1 | |
| Software, algorithm | Prism | GraphPad | Versions 7 and 8 | |
| Software, algorithm | Ingenuity pathway analysis | QIAGEN | Version 81348237 | |
| Software, algorithm | Imaris | Bitplane | | |
| Software, algorithm | Leica Application Suite (LAS X) | Leica | | |
| Software, algorithm | Zen Application Software | Zeiss | Pro 3.8 | |
| Other | SomnoSuite | Kent Scientific | SS-01 | Anesthesia |
| Other | Povidone-Iodine Prep Pad | PDI Healthcare | B40600 | Disinfection |
| Other | Alcohol Prep, Sterile, Md, 2 Ply | Covidien | 6818 | Disinfection |
| Other | Fine Forceps Dumont #55 Dumoxel | Roboz Surgical Instrument | RS-5063 | Surgical tool |
| Other | 7 mm Reflex Wound Clips | Cell Point Scientific | 203-1000 | Surgical tool |
| Other | Micro Friedman Rongeur | Roboz Surgical Instrument | RS-8306 | Surgical tool |
| Other | McPherson-Vannas Micro Dissecting Spring scissors | Roboz Surgical Instrument | RS-5600 | Surgical tool |
| Other | COATED VICRYL (polyglactin 910) Suture | Ethicon | J463G | Surgical suture |
| Other | Dumont #7 curved forceps | Fine Science Tools | 11271-30 | Surgical tool |
| Other | Miltex Halsted mosquito forceps | Integra LifeSciences | 724 | Surgical tool |
| Other | Nanofil 10 µl syringe | World Precision Instruments | NANOFIL | Small syringe |
| Other | 36g beveled nanofil needle | World Precision Instruments | NF36BV-2 | Perfusion |
| Other | Non-absorbable sutures | Ethicon | 640G | Surgical sutures for parabiosis |
| Other | Absorbable sutures | Ethicon | J463G | Surgical sutures |

*Appendix 1 Continued on next page*

*Appendix 1 Continued*

| Reagent type (species) or resource | Designation | Source or reference | Identifiers | Additional information |
|---|---|---|---|---|
| Sequence-based reagent | RNAscope Probe- Mm-Gpnmb-C3-*Mus musculus* glycoprotein (transmembrane) Gpnmb (Gpnmb) mRNA | ACD Bio | 489511-C3 | 1:50 |
| Sequence-based reagent | RNAscope Probe- Mm-Ccl8-C2-*Mus musculus* chemokine (C-C motif) ligand 8 (Ccl8) mRNA | ACD Bio | 546211-C2 | 1:50 |
| Sequence-based reagent | RNAscope Probe- Mm-Cd209a-C2- musculus CD209a antigen (Cd209a) mRNA | ACD Bio | 480311-C2 | 1:50 |
| Commercial assay or kit | RNAscope Multiplex Fluorescent Reagent Kit v2 | ACD Bio | 323100 | |
| Other | Model 1525 incubator | VWR Scientific | 1525 | RNAscope equipment |
| Other | ACD hybridization oven | ACD Bio | 321710 | RNAscope equipment |
| Chemical compound, drug | Hydrogen peroxide solution | ACD Bio | PN 322381 | |
| Chemical compound, drug | 10× antigen retrieval solution | ACD Bo | 322000 | |
| Chemical compound, drug | Protease Plus solution | ACD Bio | 322331 | |
| Chemical compound, drug | Cy3 | AKOYA Biosciences | NEL744001KT | 1:2000 |
| Chemical compound, drug | Cy5 | AKOYA Biosciences | NEL745001KT | 1:2000 |
| Chemical compound, drug | DAPI mounting media | Southern Biotech | 0100-20 | |
| Chemical compound, drug | RNAscope wash buffer | ACD Bio | 310091 | |
| Chemical compound, drug | Formalin (1:10) | Fisherbrand | 427-098 | |
| Antibody | PFKfb3 (rabbit monoclonal) | Cell Signaling Technology | 13123 | 1:300 |
| Antibody | PKM (rabbit polyclonal) | Abcam | ab137791 | 1:300 |
| Antibody | Sox10 (goat polyclonal) | R&D Systems | AF2864 | 1:300 |
| Sequence-based reagent | Spp1 Forward | This paper | PCR primer | AAGTCTAGGAGTTTCCAGGTTTC |
| Sequence-based reagent | Spp1 Reverse | This paper | PCR primer | GCTCTTCATGTGAGAGGTGAG |
| Sequence-based reagent | Gapdh Forward | This paper | PCR primer | AACTTTGGCATTGTGGAAGG |
| Sequence-based reagent | Gapdh Reverse | This paper | PCR primer | GGATGCAGGGATGATGTTCT |
| Sequence-based reagent | Chil3 Forward | This paper | PCR primer | AGCCCTCCTAAGGACAAAC |

*Appendix 1 Continued on next page*

*Appendix 1 Continued*

| Reagent type (species) or resource | Designation | Source or reference | Identifiers | Additional information |
|---|---|---|---|---|
| Sequence-based reagent | Chil3 Reverse | This paper | PCR primer | GGAATGTCTTTCTCCACAGATTC |
| Sequence-based reagent | Rlp13a Forward | This paper | PCR primer | GCTGCTCTCAAGGTTGTTC |
| Sequence-based reagent | Rlp13a Reverse | This paper | PCR primer | GTACTTCCACCCGACCTC |
| Sequence-based reagent | Hif1a Forward | This paper | PCR primer | CTGATGGAAGCACTAGACAAAG |
| Sequence-based reagent | Hif1a Reverse | This paper | PCR primer | CAATATTCACTGGGACTGTTAGG |
| Sequence-based reagent | Acod1 Forward | This paper | PCR primer | GGCACAGAAGTGTTCCATAAAG |
| Sequence-based reagent | Acod1 Reverse | This paper | PCR primer | GTGGGAGCCTGAAGTCTG |
| Sequence-based reagent | Il1b Forward | This paper | PCR primer | CTTCCAGGATGAGGACATGAG |
| Sequence-based reagent | Il1b Reverse | This paper | PCR primer | TCACACACCAGCAGGTTATC |
| Sequence-based reagent | Slc16a3 Forward | This paper | PCR primer | GCAGAAGCATTATCCAGATCTAC |
| Sequence-based reagent | Slc16a3 Reverse | This paper | PCR primer | GATTGAGCATGATGAGGGAAG |
| Sequence-based reagent | Ldha Forward | This paper | PCR primer | CATTGTCAAGTACAGTCCACAC |
| Sequence-based reagent | Ldha Reverse | This paper | PCR primer | TTCCAAGCCACGTAGGTC |
| Sequence-based reagent | Pkm Forward | This paper | PCR primer | CTGGATACAAAGGGACCTGAG |
| Sequence-based reagent | Pkm Reverse | This paper | PCR primer | CAGAGTGGCTCCCTTCTTC |
| Sequence-based reagent | Pgk1 Forward | This paper | PCR primer | GTGGAATGGCCTTTACCTTC |
| Sequence-based reagent | Pgk1 Reverse | This paper | PCR primer | GACAATCTTGGCTCCTTCTTC |
| Sequence-based reagent | Cd38 Forward | This paper | PCR primer | ACTGTCCCAACAACCCTATTAC |
| Sequence-based reagent | Cd38 Reverse | This paper | PCR primer | ATCACTTGGACCACACCAC |
| Sequence-based reagent | Pfkl Forward | This paper | PCR primer | CTGCTGAGCTACACAGAGG |
| Sequence-based reagent | Pfkl Reverse | This paper | PCR primer | CGTGTCCCTTGGTGAGAAG |
| Sequence-based reagent | Gatm Forward | This paper | PCR primer | TTGCTTTGATGCTGCTGAC |
| Sequence-based reagent | Gatm Reverse | This paper | PCR primer | CACTCGATGCCCAGGTAG |
| Chemical compound, drug | SYBR Green Fluorescein Master Mix | Thermo Scientific | 4364344 | |
| Other | QuantStudio 3 real-time PCR system | Applied Biosystems | A28567 | Genotyping |
| Other | Pestle motor mixer | RPI | 299200 | Tissue homogenization |
| Other | Nanodrop | Thermo Scientific | | Measurement of nucleic acid concentration |

*Appendix 1 Continued on next page*

*Appendix 1 Continued*

| Reagent type (species) or resource | Designation | Source or reference | Identifiers | Additional information |
|---|---|---|---|---|
| Commercial assay or kit | SuperScript III First-Strand Synthesis System kit | Invitrogen | 18080051 | |
| Chemical compound, drug | RIPA buffer | Sigma | R0278 | Supplemented with 50 mM BGP, 1 mM $Na_3VO_4$, and 1:100 PIC |
| Chemical compound, drug | DC protein assay | Bio-Rad | 5000111 | |
| Chemical compound, drug | β-Mercaptoethanol | Sigma | 60242 | |
| Chemical compound, drug | 2X Laemmli buffer | Bio-Rad | 1610737 | |
| Other | Immune-Blot PVDF membrane | BioRad | 1620260 | Western blotting |
| Chemical compound, drug | TBST blocking buffer | This paper | Buffer | 48.4 g Tris Base, 351.2 g NaCl, ddH$_2$O 2 L, pH 7.4, 0.5% Tween-20 with BSA 200 mg |
| Chemical compound, drug | Bovine serum albumin | Fisher | BP1600-100 | |
| Chemical compound, drug | SuperSignal West Pico PLUS Chemiluminescent Substrate | Thermo Fisher Scientific | 34580 | |

