## [Editor Report]

Zhao et al. provide an extensive transcriptomic analysis, using single-cell RNAseq, of the injured sciatic nerve, assessing both temporal and spatial differences as well as active communication networks between cells. They focus particularly on immune cells in the nerve and demonstrate shifts in the various populations and metabolic state over time and position relative to the injury. Collectively their findings suggest that Wallerian degeneration is critical for recruiting and maturation of immune cells distal to the injury but not at the injury site itself. These results provide a valuable resource and important insights for understanding inflammation, regeneration, and pain responses following nerve injury.

---

## [Decision Letter]

**Decision letter after peer review:**

Thank you for submitting your article "The Injured Sciatic Nerve Atlas (iSNAT), Insights into the Cellular and Molecular Basis of Neural Tissue Degeneration and Regeneration" for consideration by *eLife*. Your article has been reviewed by 3 peer reviewers, one of whom is a member of our Board of Reviewing Editors, and the evaluation has been overseen Carla Rothlin as the Senior Editor. The following individuals involved in the review of your submission have agreed to reveal their identity: Mikael Simons (Reviewer #1); Bruce D Carter (Reviewer #2).

Essential revisions:

1. More analyses and comparisons of the macrophage subpopulations identified in Figure 1 and Figure 8. For example: Are the endoMacs similar to the groups identified in Figure 8, especially the Mac 3 population in the distal nerve? Also- would be helpful to relate the Mac subtypes identified here to those found in their previous study (Kalinski et al., 2020). See Rev 2 comments for more details.

2. Please define the metabolic phenotype of the macrophages specifically in the distal nerve (Figure 8) as noted by Rev 2.

3. Issue with Statistics and Analyses: Several of the timepoints are n=1. Although this is not the case in all of the scRNAseq datasets (eg. Figure 2). This must be addressed to include biological replicates as was done for the other datasets. If this is not possible- the authors must emphasize and be clear on the major caveats for those specific data sets and which of the data sets and temper claims.

4. Integrative analyses on some data sets:- Please See Comments 2 reviewer 4 re: Figure 2, Blood and lesion in Figure 4.

5. Include an analysis of differential abundance between conditions – see point 3 by reviewer 4. Please address.

6. For Figure 5S2, please quantify the dot blot, as in Figure 1S3.

*Reviewer #1 (Recommendations for the authors):*

I recommend publication in its current form.

*Reviewer #2 (Recommendations for the authors):*

It would be helpful to compare some of the Mac subpopulations identified in Figure 1 and Figure 8. Are the endoMacs similar to the groups identified in Figure 8, especially the Mac 3 population in the distal nerve? Similarly, the authors should relate the Mac subtypes identified here to those found in their previous study (Kalinski et al., 2020), profiling the nerve after an axotomy injury. For example, do the distal Macs in the present study express phagocytic receptors and PS-bridging molecules, similar to what they observed after axotomy?

The authors should define the metabolic phenotype of the macrophages specifically in the distal nerve (Figure 8). Are these initially pro-inflammatory and then inflammation resolving or are they always largely anti-inflammatory? They could also immunostain for markers such as PKM2 and LDHA, as in Figure 3. In fact, the distal portion of the nerves in Figure 3N-U may even be in those sections, but they are not shown.

For Figure 5S2, please quantify the dot blot, as in Figure 1S3.

*Reviewer #3 (Recommendations for the authors):*

1. While it is commendable to present summary metadata that is so often hidden, several of the timepoints are n=1. Although this is not the case in all of the scRNAseq datasets (eg. Figure 2). This must be addressed and increased for the other datasets.

2. At several points in the paper, the authors analyze and display datasets separately (eg. Different time points in Figure 2, Blood and lesion in Figure 4), and at other points, they display the data together (injured and distal in Figure 8).

It is not clear to me why some of these datasets cannot be analyzed together and a proper differential gene expression analysis performed at the cell type level. This would be a critical addition that would be of immense use to the field. (Note this is distinct from the dataset integration performed in figure 6). In general, there is a total lack of differential expression analysis throughout all the datasets which massively reduces the impact of this study. The authors have all the data and could easily implement these analyses.

3. With the exception of the injury v distal analysis there is no analysis of differential abundance between conditions and unfortunately it doesn't leverage the longitudinal nature of their core dataset.

---

## [Author Response]

Essential revisions:1. More analyses and comparisons of the macrophage subpopulations identified in Figure 1 and Figure 8. For example: Are the endoMacs similar to the groups identified in Figure 8, especially the Mac 3 population in the distal nerve? Also- would be helpful to relate the Mac subtypes identified here to those found in their previous study (Kalinski et al., 2020). See Rev 2 comments for more details.

It would be helpful to compare some of the Mac subpopulations identified in Figure 1 and Figure 8. Are the endoMacs similar to the groups identified in Figure 8, especially the Mac 3 population in the distal nerve?

We agree with the reviewer. In the injured sciatic nerve atlas (https://cdb-rshiny.med.umich.edu/Giger_iSNAT/) we included scRNAseq datasets of 3day (3d) injury site and 3d distal nerve. The vast majority of immune cells at both locations of the 3d injured nerve, are blood-borne, and thus not endoMac. This was demonstrated by parabiosis (Kalinski et al., 2020, PMID: 33263277) and Figure 10D of the current manuscript. While Mac3 in the 3d distal nerve express some genes enriched in endoMac, these two populations are distinct. As examples we show the distribution of three endoMac markers Cd163, Lilra5, and Unc93b1 (Figure 1) and show their distribution at the 3d injury site and 3d distal nerve (Figure 2).

The feature plots show *Cd163* us expressed by a subset of Mac in the Mac3 cluster and epiMac. Other gene products that more endoMac in the naïve nerve are broadly upregulated in the 3d injured nerve (e.g. *Unc93b1*) or downregulated by injury (e.g. *Lilra5).*

Similarly, the authors should relate the Mac subtypes identified here to those found in their previous study (Kalinski et al., 2020), profiling the nerve after an axotomy injury. For example, do the distal Macs in the present study express phagocytic receptors and PS-bridging molecules, similar to what they observed after axotomy?

The Mac subpopulations in the 3d injured nerve (Figure 2J-2Q) are highly similar to the ones previously described by Kalinski et al., 2020. We generated new scRNAseq datasets for the current manuscript, and in addition, included existing datasets originally described by Kalinski et al. 2020. In both datasets we identify 5 Mac subpopulations (Mac1 – Mac5). For a detailed description of scRNAseq datasets included in the current,study, see table 1.

We quantified the distribution of Mac subpopulations at the 3d injury site and 3d distal nerve (Figure 8H). Mac4, and to a lesser extent Mac1, are enriched at the injury site. Analysis of phagocytic receptor expression by different Mac subpopulations was previously reported by Kalinski et al., 2020. Highest levels of phagocytic receptors were found in the Mac4 subpopulation (see Figure 7 in Kalinski et al., 2020). In the current study, strong enrichment of Mac4 at the injury site was shown by scRNAseq of micro-dissected nerve injury site and distal nerve segments (Figure 8C). To validate the strong enrichment of Mac4 at the injury site we used a combination of qRT-PCR (Figure 8J), histological analysis of injured Arg1-YFP reporter mice (Figure 8, Suppl 1B), and RNAscope for the Mac4 marker gene Gpnmb in longitudinal sections of 3d injured nerves (Figure 8, Suppl 1C). Together we provide independent lines of evidence that Mac4 are strongly enriched at the nerve injury site. Our previous findings show that Mac4 express the highest levels of phagocytic receptors (Kalinski et al., 2020). Please note, other Mac populations (including Mac3) also express phagocytic receptors, though at lower levels, so it seems very likely that PS dependent phagocytosis takes place at both locations. Regarding the location specific distribution of PS-binding molecules, we previously showed they are expressed by immune and non-immune cells in the injured nerve. In the present study, for our comparison of cells at the nerve injury site vs distal nerve, we only focused on myeloid cells isolated cells by anti-CD11b immunopanning followed by scRNAseq.

2. Please define the metabolic phenotype of the macrophages specifically in the distal nerve (Figure 8) as noted by Rev 2.

We carried out additional experiments and now show in the revised manuscript that macrophages in distal nerve (as well as macrophages at the injury site) acquire a glycolytic phenotype. This is demonstrated by double-immunofluorescence labeling of longitudinal sciatic nerve sections with anti-LDHA and anti-F4/80 (Figure 5F-5L). The number of LDHA^+^ macrophages per field of view is quantified in Figure 5N.

3. Issue with Statistics and Analyses: Several of the timepoints are n=1. Although this is not the case in all of the scRNAseq datasets (eg. Figure 2). This must be addressed to include biological replicates as was done for the other datasets. If this is not possible- the authors must emphasize and be clear on the major caveats for those specific data sets and which of the data sets and temper claims.

Agreed. We carried out additional scRNAseq experiments to increase the number of biological replicates for PBMC (now n = 2 with a total of 34,386 cells, see Table S1, Suppl file 1 for details). The second round of scRNAseq confirmed findings from the first round (see revised Figure 6).

Because naïve peripheral nerve immune cells have been sequenced previously (Ydens et al., 2020 isolated CD45+CD64+F4/80+ cells and Wang et al., 2020 isolated CD45intCD64+ cells), we used these published datasets for comparison and validation of our own naïve nerve scRNAseq datasets (Figure 1). We sequenced a total of 21,973 high quality cells and report on the transcriptomes of naïve nerve cells, including subpopulations of immune cells, stromal cells, vascular cells, and Schwann cells. Given the similarity between our datasets and the ones by Ydens et al., 2020 and Wang et a., 2020, we think an n = 1 is prudent and sufficient.

For all other sc datasets the total number of replicates is n = 2-8 (see Table S1, Suppl file 1). For comparison of myeloid cells at the 3dpc injury site versus distal nerve, n = 1. We think this is sufficient since we find the same immune cell types (though with location specific distributions) as in whole 3dpc nerve (see Figure 8H and 8I). Location specific gene expression was validated by RNAscope and Arg1-YFP reporter gene expression (Figure 8L-8M) and Figure 8 Suppl. 1. The data shown confirms the quality of our location specific scRNAseq datasets.

4. Integrative analyses on some data sets:- Please See Comments 2 reviewer 4 re: Figure 2, Blood and lesion in Figure 4.At several points in the paper, the authors analyze and display datasets separately (eg. Different time points in Figure 2, Blood and lesion in Figure 4), and at other points, they display the data together (injured and distal in Figure 8).It is not clear to me why some of these datasets cannot be analyzed together and a proper differential gene expression analysis performed at the cell type level. This would be a critical addition that would be of immense use to the field. (Note this is distinct from the dataset integration performed in figure 6). In general, there is a total lack of differential expression analysis throughout all the datasets which massively reduces the impact of this study. The authors have all the data and could easily implement these analyses.

We sympathize with the reviewer and the desire to integrate all the sc datasets together and cleanly run differential expression analyses. We found the SCTransform+CCA as well as other integration techniques did a poor job integrating our data across different time points. This was apparent when cells with clear markers at one time point were being clustered with different cell types at another time point; often, the UMAPs low dimension representation and the Louvain clustering would likewise not agree on cell communities. Further, correlations inherent to the integration technique, were introduced into the corrected UMI counts which prevent their use for differential expression by introducing false positives. Simple library size normalization and log transformation lead to similar results we could not routinely validate. Candidly, this may have resulted from technical differences from differing depths of sequencing over the many sequencing runs.

However, these issues were beautifully laid out by the Satija lab shortly after our initial submission of this manuscript https://doi.org/10.1186/s13059-021-02584-9. We are very impressed at how well the new corrected UMI counts from the SCTransform “V2” normalization method performed alongside the glmGamPoi Bioconductor package. The DEG lists shown in Figure 3 and Figure 3 Suppl. 1-4 of the revised manuscript resemble the painstaking work we did with the non-integrated time specific datasets. It is our hope that the richness of this dataset and its availability to the public through iSNAT (https://cdb-rshiny.med.umich.edu/Giger_iSNAT/) will allow for it to be continuously reanalyzed as new methods emerge.

Specifically, we carried out additional analysis, including differential gene expression for major cell types at different time points in the injured nerve (Figure 3 in the revised manuscript). These include stromal/structural cells because of their immune regulatory function [e.g. eMES, pMES, dMES/Fb (Figure 3, Suppl 1-4)], as well as cell types in the myeloid lineage (Figure 3). Pathway analysis of the top injury upregulated gene products in macrophages identified “glycolysis” (Figure 3H), supporting data shown in Figure 5.

In Figure 6 and Figure 7, we show “V2” dataset integration of circulating leukocytes (PBMC), naïve nerve, 1dpc, 3dpc, and 7dpc immune cells and show how proinflammatory immune markers peak early, while inflammation resolving molecules increase more gradually and are most abundant at 7 days after injury. We included these integrated datasets in iSNAT to facilitate differential gene expression analysis.

5. Include an analysis of differential abundance between conditions – see point 3 by reviewer 4. Please address.

We now include a table showing the relative abundance of all major cell types in the naïve and at different post-injury timepoints (Figure 2, Suppl. 3)

6. For Figure 5S2, please quantify the dot blot, as in Figure 1S3.

The quantification (bar graph) of proteins detected in the injured sciatic nerve, at different post-injury time points, is now shown in the revised manuscript, please see the revised Figure 4, Suppl 1.